# ANALYZING AND INTERNALIZING COMPLEX POLICY DOCUMENTS FOR LLM AGENTS

## ABSTRACT

Large Language Model (LLM) based agentic systems rely heavily on in-context policy documents that encode diverse business rules. As business requirements expand, these documents grow substantially, creating significant computational overhead. This motivates the need for internalization methods that embed policy documents into model priors while preserving performance. While prior prompt compression research primarily targets generic prompts, we find that agentic policy documents span multiple levels of complexity and demand more intensive reasoning, presenting greater internalization challenges. We first introduce *CC-Gen*, an agentic benchmark generator with *Controllable Complexity* defined across four levels, enabling systematic benchmarking of how well agents handle complexities and provides a framework for comprehensive evaluation of policy internalization algorithms. Our initial analysis reveals that complex policy specifications governing agent workflows may pose the most significant reasoning challenges. When supporting internalization with gold user–agent interaction trajectories containing chain-of-thought (CoT) annotations through supervised fine-tuning (SFT), we find that this baseline is highly data-intensive and its effectiveness deteriorates markedly as policy document complexity increases. To mitigate data burden and reasoning challenges, we propose Category-Aware Policy Continued Pretraining(CAP-CPT). Our automated pipeline analyzes policy documents to extract key specifications, grouping them into factual, behavioral, and conditional types. We further isolate complex conditions, which introduce high workflow complexity and drive core reasoning difficulty. This categorization guides a targeted therapy, synthesizing specialized training data for each specification type and enabling agents to internalize policy information more effectively through an autoregressive pretraining loss. Our extensive experiments demonstrate the effectiveness of the curated data and training objective. Combined with SFT, our approach improves baseline across all data scenarios. It is especially effective in data-sparse settings and under high policy complexity, yielding gains of up to 41% and 22% on Qwen-3-32B. Overall, we achieve up to 97.3% prompt length reduction in our benchmark. Applied to $\tau$-Bench, our approach further improves performance and reduces input length with very limited SFT data. [1]

## 1 INTRODUCTION

While Large Language Models (LLMs) exhibit strong instruction-following abilities (Ouyang et al., 2022; Zhou et al., 2023; Zeng et al., 2023), LLM-based agents still depend heavily on in-context policy documents to function as effective user assistants. For instance, as illustrated in Figure 1, an airline policy document must be provided in context for the agent to perform its duties. However, these documents, which often encode extensive business rules and behavioral guidelines, can consume a large portion of the input prompt. Even in simplified simulated environments such as $\tau$-Bench (Yao et al., 2024), they account for roughly 35% of the input tokens. In real-world applications, policy prompts expand with business growth and can already reach $\sim$50K tokens [2], dominating the prompt relative to user inputs and in some cases exceeding the available context length. This cre-

---

[1] All data and code will be publicly released.
[2] Exact numbers are not disclosed due to the proprietary nature of system prompts.

ates substantial computational overhead and highlights the need for efficient internalization methods that embed policy documents into a model's prior knowledge while preserving agent performance.

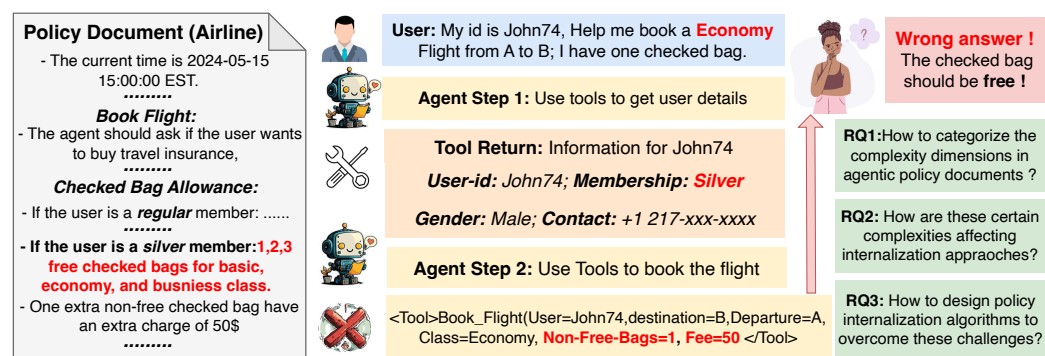

Figure 1: Even state-of-the-art LLM-based agents fail to reliably follow policy documents, and our analysis shows that certain policy specifications are inherently complex, imposing substantial reasoning demands. These observations motivate the central research questions we investigate in this paper. A more detailed illustration of this failure case is provided in Appendix J.

While prior token-compression approaches typically treat all inputs as generic prompts (Zou et al., 2024; Li et al., 2024), our observations show that models often struggle to follow specific policy specifications, suggesting that internalizing policy documents poses distinct challenges. As shown in Figure 1, evaluation on $\tau$-bench reveals that even Claude-4-Sonnet (Bubeck et al., 2024) based tool-using agents suffer severe performance degradation with policy documents as short as 1K tokens. To the best of our knowledge, no prior work has systematically examined what makes a policy document easy or difficult to follow. To investigate the cause, we manually analyzed user–agent interaction trajectories and found that certain policy specifications are inherently more complex, imposing substantial reasoning demands that degrade performance (see concrete examples in Appendix J). These insights motivate us to categorize policy complexities, measure their impact on internalization methods, and design algorithms to mitigate these challenges.

To address these challenges, we introduce *CC-Gen*, a benchmark generator that synthesizes policy documents and paired agentic tasks with predefined *Controllable Complexity*. It specifies four levels of complexity: environmental, task level, workflow, and user query (see Appendix A for definitions), allowing each to be independently manipulated to isolate its impact on agent performance. *CC-Gen* further supports fine-grained synthesis of policy modifications and policy-centric QAs, enabling systematic evaluation of both prompting-based and internalization approaches. Our initial results reveal that workflow complexity induces the most severe performance degradation for tool-using agents, followed by task-level complexity, highlighting the key challenges for effective policy internalization. Building on these findings, we construct benchmarks with varied workflow and task-level complexities to evaluate internalization methods across both standard task-oriented queries and broader capabilities such as policy substitution, override, referral, and general instruction following. As a baseline, we curate 1K–30K gold chain-of-thought trajectories for supervised fine-tuning (SFT). Our results show that SFT remains highly data-intensive and suffers from substantial performance gaps under high complexities, underscoring the need for more effective internalization approaches to improve agent robustness and generalization.

To overcome the limitations of baseline approaches, we propose Category-Aware Policy Continued Pretraining(CAP-CPT). Central to our method is an automated pipeline for policy complexity analysis. We leverage an LLM to categorize policy specifications into three types: factual, behavioral, and conditional, further subdividing conditional specifications into simple and complex cases. Each type presents distinct learning challenges, prompting us to generate tailored data for each category. Across all policy specification categories, we construct policy paraphrases and question–answer pairs to seed a compact understanding and durable recall of the documents. Since conditional specifications frequently govern agent workflows, we simulate diverse scenarios in which agents must solve subproblems that hinge on these complex conditions. For behavioral specifications, we add role-model agent demonstrations. We then combine all generated data with existing SFT trajectories, producing a dataset of five complementary data types. Finally, we apply continual pretraining

with an autoregressive loss over all tokens, enabling the model to broadly acquire policy knowledge and generalize across complexity levels.

Combining our approach with SFT, we improve baseline performance by over 10% across all scenarios on Qwen-3-32B. Notably, our method boosts performance by 44% in data-sparse settings and reduces performance disparities between workflow complexity level (1) and level (3) by up to 37%. Ablation studies confirm that our curated scenario-simulation data is crucial for handling complexity and that our CPT-based training outperforms using the same data for SFT alone. Beyond task-oriented evaluations, our method achieves superior results on policy referral, substitution, and override tasks (Comprehensive evaluation framework in Appendix E), while maintaining strong general instruction-following ability (Zhou et al., 2023). Overall, our approach achieves up to 97.3% input token compression on our synthetic benchmark and remains broadly applicable with minimal assumptions about the policy document. Applied to $\tau$-Bench, it further improves performance and reduces input length even with very limited SFT data.

Overall, our contributions are: (1) We characterize complexity types in agentic policy documents and construct benchmarks with controllable complexity, enabling systematic evaluation of internalization methods and laying a foundation for future research. (2) Using these benchmarks, we analyze what makes policy internalization challenging and identify complex workflows as the primary driver of performance degradation. (3) We propose Category-Aware Policy Continued Pretraining, which categorizes policy specifications, and curates targeted data for continual pretraining. Experiments show that our approach delivers substantial performance gains across diverse scenarios and remains broadly applicable with minimal assumptions about the policy document.

## 2 COMPLEXITY CHARACTERIZATION OF LLM-BASED AGENTIC TASKS

### 2.1 LLM-BASED AGENTIC TASK SETTING

To isolate the effect of policy complexity from confounding factors such as multimodal inputs (Xie et al., 2024) or unstable user simulators in multi-turn dialogues (Wang et al., 2024; Zhu et al., 2024), we focus on text-only, single-turn, LLM-based agentic tasks. The user provides a query $q \in \mathcal{Q}$ that specifies potentially complex requirements and a target task. The agent receives a general instruction $\mathcal{I}$ and a policy document $\mathcal{P}$, a long text corpus defining tasks, completion rules, tool usage instructions, few-shot demonstrations, and general prompts that guide the LLM as an agent. At each step $t$, the agent maintains a history $h_t = (q, \mathcal{I}, \mathcal{P}, r_{<t}, a_{<t}, o_{<t})$ and applies a recursive mapping $(r_t, a_t) = LLM(h_t)$, where $r_t$ is the reasoning trace and $a_t$ is an action from the tool set defined in $\mathcal{P}$. The action is executed by a tool function $g \in G$, producing an observation $o_t = g(a_t)$, after which the history is updated. The external environment is restricted to database access to ensure controlled workflows. The full trajectory is $\tau = \{q, \mathcal{I}, \mathcal{P}, r_1, a_1, o_1, \ldots, r_T, a_T, o_T\}$ and terminates when $(r_T, a_T, o_T)$ resolves $q$ under $\mathcal{P}$ or fails after reaching the iteration limit. We leave multimodal and multi-turn extensions to future work (Appendix M).

### 2.2 *CC-GEN*: AGENTIC BENCHMARK GENERATOR WITH CONTROLLABLE COMPLEXITIES

Based on the above setting, we categorize policy-governed agentic tasks along four complexity dimensions: **task-level complexity**, reflecting the intricacy of predefined tasks determined by their number and required arguments; **workflow-level complexity**, arising from the logical rules in policy documents, such as nested *if–else* structures, their depth, and branching factors; **environmental-level complexity**, depending on the richness and scale of external databases accessible through tool functions; and **query-level complexity**, originating from user queries that may impose special requirements or additional reasoning constraints. Each dimension is quantified by a Complexity-Type $K$, where larger $K$ indicates higher complexity, with formal definitions and quantization provided in Appendix C. Building on these complexity dimensions, we propose *CC-Gen*, a benchmark generator with fine-grained control over complexity. Given user-specified parameters and sample size, *CC-Gen* produces benchmarks comprising a policy document $\mathcal{P}$ defining global attributes, rules, interaction environment, tool usage instructions, and task specifications; a set of databases with initialized data and executable tools for agent-environment interaction; and a collection of user queries mapped to one or more tasks, optionally with gold trajectories. As summarized in Table 1, the benchmarks generated by *CC-Gen* offer three key advantages: (1) they provide sufficiently complex

policy documents to serve as rich conditioning context for completing target tasks; (2) they expose controllable complexity across all characterized dimensions, enabling systematic studies of their individual and joint effects; and (3) they form a comprehensive testbed for evaluating policy internalization methods, supporting abundant training data as well as policy-referral and policy-override tasks. These evaluation tasks are described in Section §4 and Appendix D. The generator workflow is illustrated in Figure 4, with further implementation details in Appendix A and concrete data examples in Appendix B.

Table 1: Comparison of existing agentic benchmarks and those produced by our *CC-Gen*. *CC-Gen* distinguishes itself by (1) supporting long, complex policy documents, (2) allowing for controllable complexity to systematically study its effects, and (3) supporting more comprehensive internalization training and evaluation, including policy-referral and policy-override tasks.

| Agent Benchmark | Data Instances | Tool Usage | Long Policy Document | Complexity Study | | Internalization Evaluation | |
| --- | --- | --- | --- | --- | --- | --- | --- |
| | | | | Characterization | Control | Policy-Referral | Policy-Override |
| AgentIF Qi et al. (2025) | 707 | ✓ | ✗ | ✓ | ✗ | ✗ | ✗ |
| IFEval Zeng et al. (2023) | 541 | ✗ | ✗ | ✗ | ✗ | ✗ | ✗ |
| Tau-Bench Yao et al. (2024) | 165 | ✓ | ✓ | ✗ | ✗ | ✗ | ✗ |
| Follow-Bench Jiang et al. (2024) | 820 | ✗ | ✗ | ✗ | ✗ | ✗ | ✗ |
| AgentOrca Li et al. (2025) | 663 | ✓ | ✗ | ✗ | ✗ | ✗ | ✗ |
| Multi-IF He et al. (2024) | 4501 | ✗ | ✗ | ✗ | ✗ | ✗ | ✗ |
| ComplexBench Wen et al. (2024) | 1150 | ✓ | ✗ | ✓ | ✗ | ✗ | ✗ |
| Sys-Bench Qin et al. (2024) | 500 | ✗ | ✗ | ✗ | ✗ | ✗ | ✗ |
| **Ours (CC-Gen)** | **Unlimited** | ✓ | ✓ | ✓ | ✓ | ✓ | ✓ |

Table 2: Tool-using agent performance under varying complexity levels. For each setting, evaluation data are randomly sampled from *CC-Gen*. Workflow(K) and Task(K) denote the respective complexity levels, with formal definitions in Section § 2.3. Model performance consistently declines as task-level and workflow complexity increase, with some models dropping to zero under the most challenging workflow settings.

| Performance of Tool Using Agents under Different Complexities. *Evaluation Metric: Success Rate* | | | | | | | | | | | | |
| --- | --- | --- | --- | --- | --- | --- | --- | --- | --- | --- | --- | --- |
| Model / Complexity | Workflow (1) | | | | Workflow (2) | | | | Workflow (3) | | | |
| | Task (3) | Task (5) | Task (8) | Task (12) | Task (3) | Task (5) | Task (8) | Task (12) | Task (3) | Task (5) | Task (8) | Task (12) |
| **Gemma-3-27B** | 0.28 | 0.30 | 0.17 | 0.11 | 0.20 | 0.17 | 0.03 | 0.00 | 0.07 | 0.03 | 0.02 | 0.00 |
| **Qwen2.5-32B** | 0.26 | 0.07 | 0.02 | 0.01 | 0.03 | 0.04 | 0.00 | 0.00 | 0.01 | 0.01 | 0.00 | 0.00 |
| **Qwen-3-8B** | 0.62 | 0.59 | 0.52 | 0.44 | 0.54 | 0.36 | 0.16 | 0.13 | 0.40 | 0.33 | 0.10 | 0.07 |
| **Qwen-3-32B** | 0.83 | **0.82** | **0.75** | **0.71** | **0.79** | **0.62** | **0.47** | **0.25** | **0.68** | **0.53** | **0.42** | **0.11** |
| **Claude-3-5-Sonnet** | **0.84** | 0.75 | 0.71 | 0.47 | 0.58 | 0.35 | 0.13 | 0.03 | 0.64 | 0.06 | 0.08 | 0.00 |

## 2.3 Benchmarking Agent Performance with Controlled Complexity

We conduct experiments (see Appendix A) to examine how complexity dimensions impact agent performance and reasoning, motivated by the hypothesis that they likewise obstruct internalization. Our experiments yields three main observations: (1) environmental complexity has minimal effect, as it is not directly exposed to agents and only indirectly affects the number of required tools, causing slight performance variation; (2) task-level complexity causes a gradual performance decline, whereas workflow-level complexity leads to a much sharper drop, underscoring their influence on reasoning and internalization and motivating us to benchmark their effects; and (3) while query-level complexity is crucial in practice, we leave it unconstrained to preserve user input flexibility; accordingly, we randomly sample queries from the task space defined by $\mathcal{P}$ for benchmarking and follow-up evaluation. Guided by these observations, we construct 12 benchmark settings with controlled task-level and workflow-level complexities (as they appear to pose the greatest reasoning challenges and most strongly degrade in-context and internalization performance). As shown in Table 2, Task($N$) denotes a benchmark where the policy specifies $N$ predefined tasks, each requiring $N$ correct arguments computed according to the policy rules, and Workflow($K$) denotes a benchmark where computing a task argument involves an *if–else* structure of depth $K$ (see complexity quantification in Appendix A and examples in Appendix B). Model performance consistently declines as both dimensions increase. All models are sensitive to rising workflow complexity, but some degrade sharply, even to zero in the most challenging settings, while others remain more robust. Notably, the

Qwen-3 series shows significantly greater resilience, consistently outperforming Claude-3.5 under high-complexity conditions.

# 3 INTERNALIZING COMPLEX AGENTIC POLICY DOCUMENTS

Based on the agent setting defined in Section §2.1, the goal of internalization is to partially or fully remove the policy document $\mathcal{P}$ from the input. Viewing the agent as $M_\theta$, full internalization corresponds to enforcing the alignment $\mathcal{M}_\theta(q, \mathcal{I}, \mathcal{P}) \sim \mathcal{M}_\theta(q, \mathcal{I})$, meaning the model should produce equivalent outputs without explicitly receiving $\mathcal{P}$. In practice, a policy $\mathcal{P}$ may have multiple versions across domains or situational requirements. To efficiently manage these and provide a recall anchor, we assign each policy a unique identifier (e.g., `<#Policy-1356X>`), encouraging the model to treat identifiers as retrieval cues that strengthen its ability to recall and apply the correct rules at inference time. In deployment, such identifiers would be supplied by a routing or RAG system that selects the relevant policy based on the user query. Let $pid$ denote the identifier for policy $\mathcal{P}$; our objective becomes aligning $\mathcal{M}_\theta(q, \mathcal{I}, \mathcal{P})$ with $\mathcal{M}_\theta(q, \mathcal{I}, pid)$. We adopt this formulation throughout training, with concrete examples of prompt formats and token usage provided in Appendix B.

## 3.1 BASELINE: SFT WITH GOLD COT-ENHANCED INTERACTION TRAJECTORIES

To capture the complex reasoning dynamics required by policy documents and to align model outputs with the desired behavior, we curate 1K–30K full interaction trajectories augmented with manually constructed gold Chain-of-Thought (CoT). As described in Section §2.1, each trajectory is formulated as $\tau = \{q, \mathcal{I}, \mathcal{P}, r_1, a_1, o_1, r_2, a_2, o_2, \ldots, r_T, a_T, o_T\}$. To match the inference format, the policy $\mathcal{P}$ is replaced with an identifier $pid$, which in practice would be obtained by a routing or RAG system. The reasoning steps $\{r_1, \ldots, r_T\}$ are manually curated to ensure interpretability and logical consistency. The action sequence $\{a_1, \ldots, a_T\}$ corresponds to ground-truth actions provided by our benchmark generator, while the observation sequence $\{o_1, \ldots, o_T\}$ is deterministically produced through the tool set. This yields training data of the form $\tau = \{q, \mathcal{I}, pid, r_1, a_1, o_1, r_2, a_2, o_2, \ldots, r_T, a_T, o_T\}$. We perform supervised fine-tuning (SFT) on these trajectories by minimizing the standard autoregressive loss over reasoning and action tokens: $\mathcal{L}_{\text{SFT}} = -\sum_t \log p_\theta(y_t \mid y_{<t}), \ y_t \in \{r_t, a_t\}$. To study data sparsity, we train on datasets of size 1K, 5K, 10K, 20K, and 30K independently.

## 3.2 OUR APPROACH: CATEGORY-AWARE POLICY CONTINUED PRETRAINING

While training with Gold CoT-Enhanced Interaction Trajectories yields reasonable internalization performance, our experiments reveal two major limitations. First, like other SFT methods, it is highly data-intensive and fails in data-sparse settings, a critical issue in real-world scenarios where collecting full interaction trajectories with exemplar Chain-of-Thought annotations is difficult. Second, the approach struggles with the intensive reasoning demands of complex policy documents, with performance dropping by up to 46% as workflow complexity increases from level (1) to level (3) on Qwen-2.5-32B models (see Section§ 4). To address these challenges, we propose Category-Aware Policy Continued Pretraining, which implements an automatic pipeline that analyzes policies, categorizes their specifications into four types, and generates tailored data for continued pretraining.

**Policy Document Analysis and Categorization**  Our core insight, drawn from the analysis in Section §2.3, is that different policy specifications pose distinct challenges for reasoning and internalization. To address this, we categorize elements of policy documents by how they are applied in the agent reasoning process and how they affect internalization algorithms. Based on our observation for real-world policies, we define four categories of specifications: Factual Policy Specifications, Behavioral Policy Specifications, Simple Conditional Specifications, and Complex Conditional Specifications. Detailed definitions are provided in Appendix C. As shown in the upper part of Figure 2, our pipeline begins with an LLM-based preprocessing step: the LLM is prompted to identify task types in the policy, extract the corresponding specifications, and classify them into these four categories. In parallel, the LLM determines the valid scope of each specification to construct a complete representation of the policy. For more complex cases in practice, this process may be enhanced by an optional manual check to ensure the categorization is accurate.

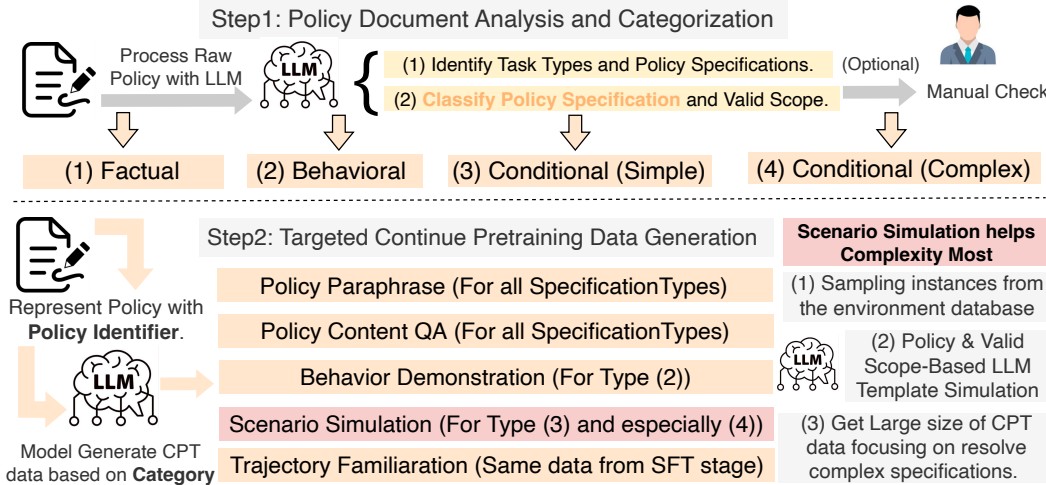

Figure 2: Pipeline for our Category-Aware Policy Continued Pretraining (CAP-CPT).**Top:** An LLM-centric pipeline analyzes policy documents and categorizes policy specifications into four major types. **Bottom:** Based on this categorization, we generate targeted training data for each specification type. In particular, scenario-simulation examples address conditional rules that require complex reasoning, helping the model internalize and apply the most challenging policy knowledge.

**Targeted Continued Pretraining Data Generation** After policy analysis and categorization, our pipeline leverages an LLM to generate targeted data for each specification type. In all cases, direct references to the policy are replaced with the policy identifier $pid$. As illustrated in Figure 2, we adopt a "targeted therapy" perspective: the data generation process is tailored to the distinct complexity of each specification category. For factual specifications, the primary challenge is memorization and accurate recall. To address this, we construct policy paraphrases and QA-style content that strengthen the model's ability to store and retrieve policy details. For behavioral specifications, the challenge shifts from simple recall to demonstrating compliant behaviors under defined circumstances. Accordingly, we curate data where ground-truth responses act as role models: the LLM generates scenarios requiring the application of behavioral rules, queries the agent, and produces responses that consistently reflect satisfactory and policy-aligned behavior. Conditional specifications govern the workflow of the LLM and their influence increases with complexity. To support this, we curate large volumes of scenario-simulation data that go beyond memorization, emphasizing the practical application of policy rules and enabling the model to fully exercise its reasoning capabilities. Unlike standard CPT data focused on rote recall, this simulation data operationalizes the policy document, transforming abstract rules into executable workflows. An intuitive explanation of why such data better facilitate model learning is provided in Appendix F. During this process, the LLM synthesizes scenarios and samples concrete instances from the environment database. For example, given the complex policy specification in Figure 1, the LLM can generate numerous queries by sampling user and reservation details, then compute the correct number of non-free checked bags and the corresponding total fee. Finally, we incorporate SFT trajectory data as an auxiliary source to better prepare the model for downstream task solving. Although all curated data are structured in QA format, they are employed within a continued d pretraining (CPT) paradigm, where the objective is to minimize the standard language modeling loss $\mathcal{L}_{\text{CPT}} = -\sum_{t=1}^{T} \log P_\theta(x_t \mid x_{<t})$, with $\theta$ denoting model parameters and $x_t$ the target token at position $t$. The CPT stage enhances the model's ability to internalize and reason over policy content, rather than merely memorizing query answer pairs. We validate the effectiveness of our curated data and training objective in Section § 4.

## 4 EVALUATION OF POLICY DOCUMENT INTERNALIZATION

### 4.1 EXPERIMENT SETTINGS

**Model and Data Settings** We use Qwen-2.5-32B and Qwen-3-32B for policy document internalization, chosen for their strong prior knowledge and distinct performance when complex policy

documents are provided in context. To evaluate complexity effects, we sample datasets that control other dimensions while varying workflow complexity from level (1) to (3), as well as datasets that vary task-level complexity with level (3), (5), (8), and (12) tasks. For SFT, we provide between 1K, 5K, and up to 30K training samples. We also apply our approach to $\tau$-Bench, which offers only 500 training samples with no CoT based reasoning. Using Qwen-3-32B, we self-generate CoT trajectories and yield 282 SFT samples. More details are in Appendix D.

**Evaluation Framework and Metrics** The primary focus of our evaluation is task completion after policy internalization, where agents must follow the internalized policy document to execute predefined tasks. To provide a more comprehensive assessment, we also consider scenarios involving policy substitution or override, policy-referral QA grounded in the document, and general instruction-following tests using IFeval (Zhou et al., 2023). Detailed settings are in Appendix E. Task completion is measured by success rate (SR), policy QAs are scored on a 0–5 scale by a language model and rescaled to 0–100, and instruction following is evaluated by average accuracy.

Table 3: Task-completion performance after policy internalization under varying workflow complexities, with SFT trajectory sizes from 1K to 30K. Our CAP-CPT + SFT consistently outperforms strong baselines, alleviates data sparsity, and reduces the gap between high- and low-complexity scenarios. On Qwen-2.5-32B, it even surpasses agent performance with the full policy in context.

| Model | Complexity | Prompting | Internalization Approach | Internalization Training Data Size | | | | |
|---|---|---|---|---|---|---|---|---|
| | | | | 1K | 5K | 10K | 20K | 30K |
| **Qwen2.5-32B** | Task (5) Workflow (1) | 0.07 | Gold CoT SFT | 0.04 | 0.80 | 0.95 | 0.97 | 0.98 |
| | | | CAP-CPT + Gold CoT SFT | 0.57 | 0.94 | 0.98 | 0.98 | **0.99** |
| | Task (5) Workflow (2) | 0.04 | Gold CoT SFT | 0.03 | 0.23 | 0.31 | 0.47 | 0.59 |
| | | | CAP-CPT + Gold CoT SFT | 0.43 | 0.66 | 0.74 | 0.88 | **0.90** |
| | Task (5) Workflow (3) | 0.01 | Gold CoT SFT | 0.00 | 0.14 | 0.26 | 0.32 | 0.52 |
| | | | CAP-CPT + Gold CoT SFT | 0.36 | 0.63 | 0.72 | 0.85 | **0.85** |
| **Qwen3-32B** | Task (5) Workflow (1) | **0.82** | Gold CoT SFT | 0.03 | 0.41 | 0.55 | 0.71 | 0.78 |
| | | | CAP-CPT + Gold CoT SFT | 0.44 | 0.67 | 0.72 | 0.74 | 0.80 |
| | Task (5) Workflow (2) | **0.62** | Gold CoT SFT | 0.02 | 0.18 | 0.23 | 0.35 | 0.42 |
| | | | CAP-CPT + Gold CoT SFT | 0.27 | 0.35 | 0.46 | 0.53 | 0.57 |
| | Task (5) Workflow (3) | **0.53** | Gold CoT SFT | 0.01 | 0.13 | 0.17 | 0.31 | 0.36 |
| | | | CAP-CPT + Gold CoT SFT | 0.16 | 0.27 | 0.39 | 0.41 | 0.47 |

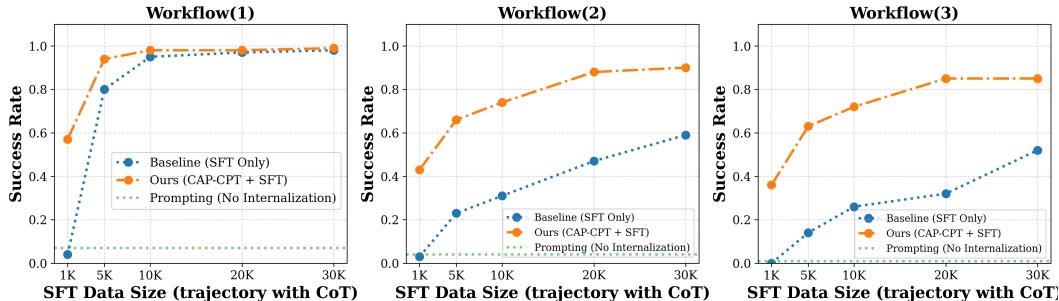

Figure 3: Performance curves for internalizing policy documents with varying workflow complexities on Qwen-2.5-32B, comparing the baseline with our method. Our approach consistently outperforms the baseline across all settings and substantially narrows the performance gap in high-complexity and data-sparse scenarios.

## 4.2 MAIN RESULTS

**CAP-CPT Significantly Boosts Performance** We evaluate agent task-completion performance under varying workflow complexities in Table 3, with corresponding performance curves in Figure 7. Relying solely on Gold CoT–enhanced trajectory data for SFT is highly data-intensive and

results in large disparities across complexity levels. In contrast, our CAP-CPT approach consistently improves performance across all data splits, with particularly strong gains under data-sparse conditions. Although the curated data is not explicitly optimized for task completion, it substantially strengthens policy internalization and narrows performance gaps: CAP-CPT reduces the disparity between high- and low-complexity scenarios by 37% on Qwen-2.5-32B and 21% on Qwen-3-32B, even with abundant SFT data. This yields more robust and generalizable policy understanding. Similar trends are observed under varying task-level complexities (Appendix D). Overall, our internalization achieves input token compression of up to 97.3%. Notably, internalization training on the strongest base models does not yield gains over the prompting baseline or over training on originally weaker models. We analyze this in Appendix F.

**CAP-CPT Helps Under Broader Evaluation Settings** We evaluate agent post-internalization performance on policy-referral, policy-substitute, and policy-override tasks, as well as general instruction following. Results on Qwen-3-32B are shown in Table 4, with more comprehensive results in Appendix D. Across all policy-related tasks, our method substantially outperforms SFT baselines but does not surpass the prompting baseline, indicating that these out-of-domain tasks remain challenging and warrant further study. For policy-substitute and policy-override, both require balancing internalized rules with newly introduced ones, with full substitution proving more difficult than partial override. Improving performance in these settings will likely require additional training data. For policy-referral, the model immediately after continued pretraining achieves the highest score, but its performance steadily declines as SFT data size increases, suggesting that SFT tends to hard-code task solutions rather than really helps to understand policy rules and learn how to put them into practice. Finally, general instruction-following ability is largely preserved, likely because policy-focused training is orthogonal to generic instruction following.

Table 4: Comprehensive evaluation results on post-trained Qwen-3-32B across supportive tasks—including Policy-Substitute, Policy-Override, Policy-Referral, and instruction following, with further details in Appendix D. While our approach consistently outperforms SFT baselines after internalization, performance on most tasks still lags behind in-context prompting, suggesting that additional task-specific training data is needed to fully retain these specialized capabilities.

| Model | Complexity | Prompting | Internalization Approach | Internalization Training Data Size | | | | |
|---|---|---|---|---|---|---|---|---|
| | | | | 1K | 5K | 10K | 20K | 30K |
| **Qwen-3-32B** (Substitute) | Task (5) Workflow (3) | **0.53** | Gold CoT SFT | 0.01 | 0.00 | 0.02 | 0.00 | 0.00 |
| | | | CAP-CPT + Gold CoT SFT | 0.07 | 0.06 | 0.08 | 0.06 | 0.05 |
| **Qwen-3-32B** (Override) | Task (5) Workflow (3) | **0.53** | Gold CoT SFT | 0.00 | 0.00 | 0.00 | 0.00 | 0.00 |
| | | | CAP-CPT + Gold CoT SFT | 0.09 | 0.12 | 0.17 | 0.22 | 0.25 |
| **Qwen-3-32B** (Referral) | Task (5) Workflow (3) | **0.76** | Gold CoT SFT | 0.00 | 0.00 | 0.00 | 0.00 | 0.00 |
| | | | CAP-CPT + Gold CoT SFT | 0.59 | 0.31 | 0.23 | 0.20 | 0.13 |
| **Qwen-3-32B** (Ifeval) | Task (5) Workflow (3) | 0.44 | Gold CoT SFT | 0.45 | 0.43 | 0.46 | 0.42 | 0.45 |
| | | | CAP-CPT + Gold CoT SFT | 0.44 | 0.45 | 0.44 | **0.47** | 0.43 |

## 4.3 ABATION STUDY

We assess the effectiveness of our approach by evaluating two variants of the complete method. The first variant uses all generated Category-Aware QA-format data for SFT, while the second excludes the scenario-simulation data designed for complexity handling. As shown in Table 5, both variants outperform the SFT baselines, but the full approach consistently achieves the strongest results across all data settings. This underscores the importance of jointly leveraging targeted data and the CAP-CPT training objective. Additional analyses of the benefits and limitations of these two variants are provided in Appendix H. Notably, both variants still yield substantial gains over SFT-only baselines, further validating the effectiveness of our curated data. We also test our method under multi-policy internalization; results indicate that internalization performance remains consistent when applied across a number of distinct policies with different complexity levels. Details are provided in Appendix G.

Table 5: Demonstration of the effectiveness of our CAP-CPT approach. We validate the CPT training objective by applying the generated data for SFT and assess the scenario-simulation data's ability to handle complexity by selectively removing portions of it. Both variants yield suboptimal performance compared to our full approach.

| Model | Complexity | Prompting | Internalization Approach | Internalization Training Data Size | | | | |
|---|---|---|---|---|---|---|---|---|
| | | | | 1K | 5K | 10K | 20K | 30K |
| **Qwen-3-32B** | Task (5) Workflow (3) | 0.53 | Gold CoT SFT | 0.01 | 0.13 | 0.17 | 0.31 | 0.36 |
| | | | **CAP-CPT + Gold CoT SFT** | **0.16** | **0.27** | **0.39** | **0.41** | **0.47** |
| | | | (CAP-CPT data + Gold CoT) for SFT | 0.08 | 0.21 | 0.28 | 0.34 | 0.42 |
| | | | Remove Scenario Simulation Data | 0.09 | 0.23 | 0.32 | 0.36 | 0.44 |

## 4.4 APPLICATION ON $\tau$-BENCH

Finally, we evaluate our approach on $\tau$-bench. Following the setup described in Section§ 2.1, we mitigate potential user-simulator bias by modifying the protocol so that agents solve complete queries directly rather than through multi-turn interaction. We prompt Qwen-3-32B to self-generate responses for the 500 training samples provided by $\tau$-bench, yielding 282 successful trajectories with Self-CoT used for SFT. We subsequently perform policy analysis and synthesize CAP-CPT data. As summarized in Table 14, the original Qwen-3-32B model with in-context policy achieves a 26.96% success rate. After internalization using only SFT, performance slightly drops to 23.48%, underperforming the prompting baseline. In contrast, our full approach surpasses the prompting baseline, achieving a 28.70% success rate while reducing the overall input length by 34.8%. We further evaluate the policy categorization stage of our pipeline and verify that these gains persist in real-world settings without manual intervention. Notably, the policy analysis and data generation steps are executed entirely by Qwen-3-32B, eliminating the need for any external LLM APIs. Detailed precision, recall, and F1 results from this policy analysis process are provided in Appendix I.

## 5 RELATED WORK

Deliberative alignment (Guan et al., 2024; Zhang et al., 2025a) is most closely related to our work. This line of research aims to internalize general safety rules and behaviors into a model's prior, either through additional training (Guan et al., 2024) or test-time deliberation (Zhang et al., 2025a). However, it remains focused on generic safety behaviors, overlooking the broader scope of agentic policies and the complex reasoning challenges (e.g., workflow-level constraints) central to policy internalization. Besides, our work also intersects with several research areas, including prompt compression (Li et al., 2024; Chuang et al., 2024; Mu et al., 2024), knowledge injection and perception (Martino et al., 2023; Song et al., 2025a), and continued pretraining (Zhou et al., 2024). Owing to space limitations, we provide further discussion of related work in these domains in Appendix K.

## 6 CONCLUSION

In this work, we examined the challenge of internalizing long, complex policy documents in LLM-based agentic systems. We characterized distinct forms of policy complexity and introduced CC-Gen, a controllable-complexity benchmark generator for systematically analyzing agents' ability to handle varying complexities and enabling comprehensive evaluation of internalization algorithms. Our analysis identified workflow depth as a primary driver of performance degradation, highlighting limits of in-context methods and data-intensive SFT-based approaches. To address these issues, we internalize policy documents via explicit policy identifiers and an automated pipeline for policy analysis that generates Category-Aware Policy Continue Pretraining (CAP-CPT) data. This reduces SFT data demands and mitigates the reasoning challenges posed by complex specifications. Empirically, our approach yields consistent gains across scenarios and substantially narrows complexity-related performance disparities. Overall, our findings underscore the importance of explicitly modeling policy complexity and provide a scalable, effective solution for policy internalization. We hope this work motivates further research into robust and generalizable internalization for LLM agents, ultimately enabling more computationally efficient, reliable, and helpful AI assistants for all.

## 7 REPRODUCIBILITY STATEMENT

We provide an anonymous source code archive in the supplementary material, which includes our data generator as well as detailed training and evaluation instructions for reproducing the results in this paper. We use LlamaFactory Zheng et al. (2024) to train Qwen-2.5-32B and Qwen-3-32B on eight H100 GPUs. We will also publicly release the full codebase and data, including the benchmark generator to further facilitate reproducibility. All reported experimental results are based on a single run. Additional experimental details are provided in Section§ 4 and Appendix D.

## 8 ETHICS STATEMENT

This work focuses on fundamental research aimed at improving the internalization of complex policy documents in language models. No human subjects or private user data were involved in this study. The dataset introduced in this work consists entirely of synthetically generated user profiles and does not contain or rely on any real user data. To the best of our knowledge, this research does not raise any ethical concerns.

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

# A  BENCHMARK DEVELOPMENT AND PROBING EXPERIMENTS

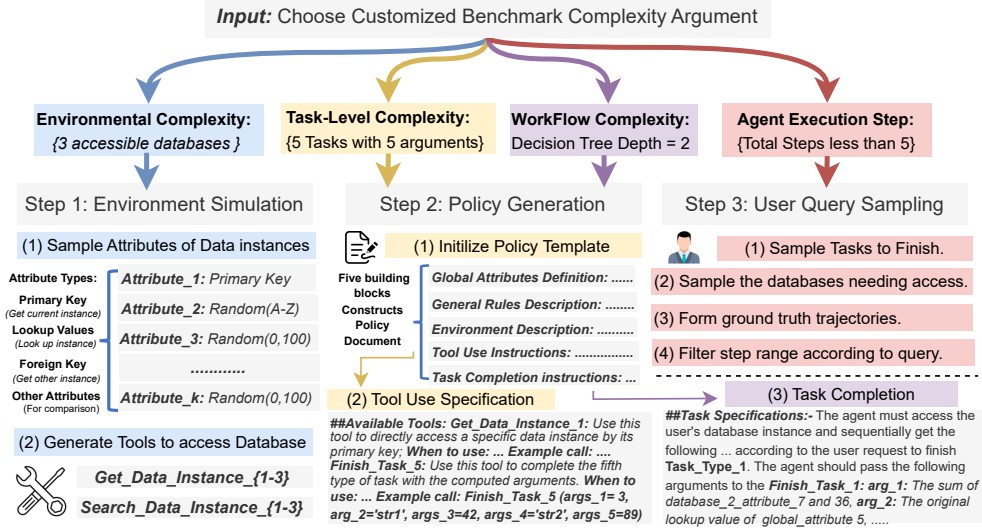

Figure 4: Pipeline of our *CC-Gen* benchmark generator.

**Complexity Characterization**  We provide additional details of our *CC-Gen* benchmark generator, including its construction, usage, and output. As illustrated in Figure 4, the generator synthesizes agentic benchmarks by composing four key components:

1. **Pre-defined environments.**  Each environment typically consists of a collection of databases, where every database has its own schema with primary keys, foreign keys, lookup keys, and other attributes. The concrete attributes of the data instances are randomly sampled.

2. **Policy documents.** Policies are instantiated from templates and tagged with explicit markers (e.g., `<Airline #Policy-1356X>`). Each policy specifies the set of tasks the agent must complete, along with detailed guidelines, global attributes, general rules, environment descriptions, and tool-use instructions.

3. **Tool definitions.** For every database, we provide two types of tools: one that retrieves a single data instance by primary key, and another that supports flexible search over designated fields. There are also tools which are designed to help agent complete tasks or report to human agents and ask for help.

4. **User queries and reference trajectories.** A benchmark includes a collection of user queries, their corresponding correct action sequences, and final answers. Users can independently control the complexity of the environment, task-level specifications, and workflow structures when generating new benchmarks. They may also restrict user query complexity, though in this paper we constrain our experiments accordingly.

We also present an example of tool-use specifications and task completion trajectories in Figure 4. A complete sample benchmark generated by *CC-Gen* is provided in Appendix B

**Complexity Quantification** 'To unify and simplify the computation of complexity dimensions in agentic tasks, and to enable users to easily quantify complexity levels, we design a set of discrete metrics for describing these dimensions. We denote Complexity-dimension (K) as the K-th level of complexity within a given dimension, and define it as follows:

**Environment (K):** This captures the number of databases that the language model agent must interact with. For $\tau$-bench, the environmental complexity is set at $K = 3$, a setting we also adopt for our main experiments. Although this number is relatively small, we validated that the impact of environmental complexity is limited; therefore, higher values in real-world scenarios would not significantly alter our evaluation.

**Task-Level (K):** This dimension reflects both the number of tasks and the number of arguments required for computation in each task. While in practice, the complexity from multiple tasks and individual task arguments can have distinct effects, we unify them into a single dimension. This is because their increase jointly contributes to the overall task complexity.

**Workflow-Level (K):** This represents the complexity of the workflow needed to complete the target task. Specifically, it accounts for the depth of logical structures (e.g., nested if–else conditions) that the agent must reason through. For simplicity, we define workflow complexity as the depth of these structures in each specification.

Although in real-world applications the complexity of each dimension may interact in more entangled ways, we unify them in our benchmark to make the construction process more interpretable and to better isolate the impact of each independent dimension. A discussion of this design choice is provided in the limitation section M.

**Probing Experiments** We conducted comprehensive probing experiments on Qwen-3-8B models to briefly have an insight on which complexity levels worth most attention. The experimental results are shown in Table 6 $\sim$ Table 9. We evaluate with both task Success Rate (SR) and also Partial Success Rate (PSR) for our probing experiments. SR is the fraction of tasks whose entire gold action sequence is executed correctly. PSR measures argument-level accuracy for tool use: for each gold action, when the agent invokes the correct tool, we compare its arguments with the gold specification and compute the fraction that match; PSR is the average of this fraction across all matched tool calls (averaged over tasks). Our experiments reveal that workflow complexity poses the most significant reasoning challenges for LLM agents, followed by task-level complexity. In contrast, the impact of environmental complexity is relatively minor, likely because agents interact with external resources primarily through tools rather than directly. In practice, adding a large external database often only introduces a few additional tool-use commands, without substantially increasing the reasoning burden. We hypothesize that this explains why environmental complexity appears less influential in our evaluations.

Table 6: Probing experimental results for different environmental complexity, where we control the task level complexity and workflow level complexity. Results show that distinct environment complexity does not matter much.

| Model | Environment (3) | Environment (5) | Environment (10) |
|---|---|---|---|
| Qwen-3-8B (SR) | 0.91 | 0.87 | 0.88 |
| Qwen-3-8B (PSR) | 0.941 | 0.913 | 0.937 |

Table 7: Probing experimental results for different task level complexity at Workflow (1), where we control the environmental complexity. Results show that increasing task complexity leads to noticeable performance degradation.

| Model | Task (3) | Task (5) | Task (8) | Task (12) |
|---|---|---|---|---|
| Qwen-3-8B (SR) | 0.92 | 0.85 | 0.67 | 0.60 |
| Qwen-3-8B (PSR) | 0.961 | 0.929 | 0.791 | 0.772 |

Table 8: Probing experimental results for different task level complexity at Workflow (2), where we control the environmental complexity. Results show that higher task complexity markedly reduces performance under deeper workflows.

| Model | Task (3) | Task (5) | Task (8) | Task (12) |
|---|---|---|---|---|
| Qwen-3-8B (SR) | 0.74 | 0.68 | 0.23 | 0.02 |
| Qwen-3-8B (PSR) | 0.876 | 0.842 | 0.578 | 0.298 |

## B    DATA EXAMPLES FOR GENERATED POLICY DOCUMENTS

We present several examples generated by our *CC-Gen* benchmark generator to demonstrate its ability to produce agentic benchmarks with controllable complexity.

---

**Real Policy Example Sampled from our Agentic Benchmark Generator CC-Gen**

**Complexity Level:** Environmental(3); Task-Level(5); Workflow(1).

# Agent Policy Document #P71067

## General Instructions

The global attribute is currently: Global-Attribute-Value1 = 30, Global-Attribute-Value2 = 60, Global-Attribute-Value3 = 7. You are a helpful agent that can get access to profiles and attributes at different layers and indexes. You can help users finish Task-Type-1, Task-Type-2, Task-Type-3, Task-Type-4, Task-Type-5.

## Domain Basic

### Profile Structure

The jth profile instance at profile layer i has its primary key as profile-i-j There are 3 layers of profiles, and each profile layer has a number of profile instances. All the profile instances at the same layer have the same attributes.

- Each profile at layer 1 indexed j Profile-1-j has attributes: Profile-1-Attribute-1, Profile-1-Attribute-2, Profile-1-Attribute-3, Profile-1-Attribute-4, Profile-1-Attribute-5, Profile-1-Attribute-6, Profile-1-Attribute-7, Profile-1-Attribute-8

- Each profile at layer 2 indexed j Profile-2-j has attributes: Profile-2-Attribute-1, Profile-2-Attribute-2, Profile-2-Attribute-3, Profile-2-Attribute-4, Profile-2-Attribute-5, Profile-2-Attribute-6, Profile-2-Attribute-7, Profile-2-Attribute-8

- Each profile at layer 3 indexed j Profile-3-j has attributes: Profile-3-Attribute-1, Profile-3-Attribute-2, Profile-3-Attribute-3, Profile-3-Attribute-4, Profile-3-Attribute-5, Profile-3-Attribute-6, Profile-3-Attribute-7, Profile-3-Attribute-8

---

Table 9: Probing experimental results for different task level complexity and workflow level complexity, where we control the environmental complexity. Results show that higher workflow and task levels jointly compound performance degradation.

| Model | Complexity | Task (5) | Task (8) |
|---|---|---|---|
| Qwen-3-8B (SR) | Workflow (1) | 0.85 | 0.67 |
| | Workflow (2) | 0.68 | 0.23 |
| Qwen-3-8B (PSR) | Workflow (1) | 0.929 | 0.791 |
| | Workflow (2) | 0.842 | 0.578 |

### Attribute Definitions

The jth attribute at layer i is denoted as profile-attribute-i-j.

At layer 1: - The attribute-1 and attribute-2 and attribute-7 and attribute-8 can serve as conditions - The attribute-4 contain the primary keys to access profiles at layer 1 - The attribute-5 contain the primary keys to access profiles at layer 2 - The attribute-6 contain the primary keys to access profiles at layer 3 - The attribute-3 can be used as an alternative way to access the profiles while searching.

At layer 2: - The attribute-1 and attribute-2 and attribute-7 and attribute-8 can serve as conditions - The attribute-4 contain the primary keys to access profiles at layer 2 - The attribute-5 contain the primary keys to access profiles at layer 3 - The attribute-6 contain the primary keys to access profiles at layer 1 - The attribute-3 can be used as an alternative way to access the profiles while searching.

At layer 3: - The attribute-1 and attribute-2 and attribute-7 and attribute-8 can serve as conditions - The attribute-4 contain the primary keys to access profiles at layer 3 - The attribute-5 contain the primary keys to access profiles at layer 1 - The attribute-6 contain the primary keys to access profiles at layer 2 - The attribute-3 can be used as an alternative way to access the profiles while searching.

### Profile Access Pattern

When the user specifies a profile-k-id, you should understand that this means the user wants to access the profile-k instance with the primary key's index being the given value. When the user specifies a profile-k-info, you should understand that this means the user wants to access the profile-k instance with the lookup attribute value of the provided string. When referring to a user's profile-k, you should use the layer k-1 profile's reference attribute to get access to the primary keys of profile-k instances.

### Relative Profile Access:

When the user specifies getting a 'relative profile' or 'related profile', this means accessing other profile instances at the same layer as the current profile. To accomplish this, you should use the reference attributes from the current profile instance to find the primary keys of the target profile instances at the same layer. For example, if you are currently accessing a profile at layer 2, and the user asks for a relative profile, you should use the reference attributes in the current layer 2 profile to identify and access other layer 2 profile instances.

## Tool Calling Instructions

### General Rules

- You should only make one tool call at a time, and if you make a tool call, you should not respond to the user simultaneously. - If you respond to the user, you should not make a tool call at the same time. - You should only call the tool Tool-Conflict when the request is not able to be handled within the policy and the user specifications.

### Available Tools

#### Profile Access Tools

- Get-Profile-Layer-k: Use this tool to directly access a specific profile instance by its primary key. - Parameter: 'index-value' (string) - The full primary key of the profile instance (e.g., "profile-1-5", "profile-2-10", "profile-3-1") - When to use: - When users specify a profile-id, such as "my profile-id is profile-1-5" or "using profile-2-3" - When you obtain a reference attribute value from another profile instance that contains the primary key to access a different layer - Example call: Get-Profile-Layer-1(index-value="profile-1-5")

- Search-Profile-Layer-k: Use this tool to find profile instances by their lookup attribute value. - Parameter: 'key-value' (string) - The lookup attribute value to search for - When to use: When users specify a profile-info, such as "my profile-info is 'engineering'" or "find profiles with lookup value 'sales'" - Example call: Search-Profile-Layer-1(key-value="engineering")

#### Task Completion Tools

- finish-task-k: Use this tool to complete Task-Type-k with the computed arguments. - Parameter: 'attributes' (list) - A list of computed argument values in the order specified by the task requirements - When to use: After accessing all required profile instances and computing the task arguments according to task specifications - Example call: finish-task-1(attributes=[25, 150, 42])

#### Conflict Resolution Tool

- Tool-Conflict: Use this tool when the user request cannot be handled within the policy constraints. - Parameters: None - When to use: If the user request violates policy or cannot be fulfilled with available tools and data - Example call: Tool-Conflict()

### Tool Parameter Mapping Guidelines

- profile-id references: When users mention "my profile-id is profile-k-X" or "profile-k-X", use the Get-Profile-Layer-k tool with index-value="profile-k-X" - reference attribute usage: When you access a profile instance and obtain reference attributes (e.g., reference-1, reference-2, reference-3), use those primary key values with Get-Profile-Layer-k to access the referenced profiles at the target layers - profile-info references: When users mention "my profile-info is Y" or provide lookup values, use the Search-Profile-Layer-k tool with key-value="Y" - Task completion: Always pass computed arguments as a list to finish-task-k tools, ensuring the order matches task specifications

### Usage Guidelines

The user will specify the instance index at the first layer, and the agent shall go through the profile instances at different indexes and layers to obtain the attributes needed for the task.

## Policy Specifications

### General Policy 1

The agent must first get access to the profile instance at layer 1 according to the user specified primary key, alternatively, the agent may also search for the profile instance at layer 1 when the user did not provide a profile instance at layer 1 and instead provided a lookup field in profile layer 1.

### General Policy 2

The agent should always finish the task with the task required attribute combinations at one time. If users specify multiple attribute combinations for the task (e.g., 'doing task i for all the instances accessd in layer 1.'), the agent must call the finish task tool multiple times and only address one attribute combination at a time.

## Task Specifications

### Task-Type-1

- The agent must access one profile instance at each of the layer 1, layer 2, layer 3 according to the user request, - The agent should pass the following arguments into the finish-task-1 tool call: - arg-1: The average of all values: (layer-3-attribute-8 + 26 + 96) divided by 3 (integer division). - arg-2: The original lookup value of layer-1-attribute-3 from the selected profile. - arg-3: The count of values greater than 50 among: layer-2-attribute-7, layer-3-attribute-2, 90, 96. - arg-4: layer-3-attribute-1 if layer-3-attribute-1 ¿ 4, else 4. - arg-5: The maximum among all values: layer-3-attribute-2, layer-2-attribute-7, 51, 59. - Each task-1 completion requires exactly one profile from each of the specified layers. - The agent should call the finish-task-1 tool with arguments from one instance per layer at a time. - Multiple function calls may be needed if multiple profile combinations are requested.

### Task-Type-2

- The agent must access one profile instance at each of the layer 1 according to the user request, - The agent should pass the following arguments into the finish-task-2 tool call: - arg-1: The sum of all values: global-attribute-2, layer-1-attribute-7, 64, 56. - arg-2: The original lookup value of layer-1-attribute-3 from the selected profile. - arg-3: The average of all values: (global-attribute-3 + layer-1-attribute-1 + layer-1-attribute-2 + 63) divided by 4 (integer division). - arg-4: The minimum among all values: global-attribute-3, global-attribute-2, layer-1-attribute-7, 46, 40. - arg-5: The sum of even values among: layer-1-attribute-8, layer-1-attribute-7, layer-1-attribute-1, 78. - The agent should call the finish-task-2 tool with the arguments above for the selected profile instance.

### Task-Type-3

- The agent must access one profile instance at each of the layer 1, layer 2, layer 3 according to the user request, - The agent should pass the following arguments into the finish-task-3 tool call: - arg-1: The maximum among all values: layer-3-attribute-7, 24, 14. - arg-2: The result of (layer-2-attribute-1 + 2 + 73) modulo 100. - arg-3: The maximum between layer-2-attribute-2 and 48. - arg-4: The original lookup value of layer-1-attribute-3 from the selected profile. - arg-5: The sum of even values among: global-attribute-1, 5, 12. - Each task-3 completion requires exactly one profile from each of the specified layers. - The agent should call the finish-task-3 tool with arguments from one instance per layer at a time. - Multiple function calls may be needed if multiple profile combinations are requested.

### Task-Type-4

- The agent must access one profile instance at each of the layer 1 according to the user request, - The agent should pass the following arguments into the finish-task-4 tool call: - arg-1: The maximum among all values: layer-1-attribute-1, 76, 65. - arg-2: The product of global-attribute-3 and 8. - arg-3: The count of values greater than 50 among: layer-1-attribute-8, layer-1-attribute-7, global-attribute-3, 22. - arg-4: The maximum

among all values: global-attribute-2, 50, 66. - arg-5: The result of (layer-1-attribute-8 + global-attribute-1 + 98 + 90) modulo 100. - The agent should call the finish-task-4 tool with the arguments above for the selected profile instance.

### Task-Type-5

- The agent must access one profile instance at each of the layer 1, layer 2, layer 3 according to the user request, - The agent should pass the following arguments into the finish-task-5 tool call: - arg-1: The range (max - min) among: global-attribute-1, layer-3-attribute-8, layer-2-attribute-2, 5, 99. - arg-2: The count of values greater than 50 among: layer-3-attribute-8, global-attribute-1, layer-2-attribute-8, 49, 52. - arg-3: The original lookup value of layer-1-attribute-3 from the selected profile. - arg-4: The average of all values: (layer-2-attribute-7 + global-attribute-3 + layer-3-attribute-1 + 59) divided by 4 (integer division). - arg-5: The sum of even values among: layer-2-attribute-2, global-attribute-2, 58, 79. - Each task-5 completion requires exactly one profile from each of the specified layers. - The agent should call the finish-task-5 tool with arguments from one instance per layer at a time. - Multiple function calls may be needed if multiple profile combinations are requested.

## C  POLICY ANALYSIS DETAILS

We use the model itself (which still requires further internalization) as the LLM for policy analysis, thereby avoiding potential knowledge distillation from stronger models. As described in Section § 3.2, we categorize policy specifications into four major types based on their influence on agent behavior:

1. **Factual Type.** The policy document states a fact that the agent must memorize and potentially paraphrase when answering user queries. These specifications do not involve reasoning or decision-making, but require accurate recall. *Example:* "The refund will be processed within 5–7 business days."

2. **Behavior Type.** The policy prescribes or prohibits certain general behaviors, independent of the workflow logic. Violating these rules does not change the structure of the task but determines whether the agent's behavior aligns with policy requirements. *Example:* "Before taking any actions that update the booking database (booking, modifying flights, editing baggage, upgrading cabin class, or updating passenger information), you must list the action details and obtain explicit user confirmation (yes) to proceed."

3. **Conditional Type (Simple).** The policy specifies simple conditional rules that directly affect the agent's workflow but require minimal reasoning to apply. The condition typically involves a straightforward check on one variable or state. *Example:* "The agent can only cancel the whole trip that is not flown."

4. **Conditional Type (Complex).** The policy encodes nested or multi-branch conditional logic that requires deeper reasoning to correctly apply. Such rules often involve multiple attributes, role-specific constraints, or cumulative calculations, and thus present higher complexity for the model. *Example:* "Checked bag allowance: If the booking user is a regular member, 0 free checked bag for each basic economy passenger, 1 free checked bag for each economy passenger, and 2 free checked bags for each business passenger. If the booking user is a silver member, 1 free checked bag for each basic economy passenger, 2 free checked bag for each economy passenger, and 3 free checked bags for each business passenger. If the booking user is a gold member, 2 free checked bag for each basic economy passenger, 3 free checked bag for each economy passenger, and 3 free checked bags for each business passenger. Each extra baggage is 50 dollars."

**Prompt Used by LLMs to Perform Policy Analysis**

You are a policy analysis assistant. Your task is to process the input policy document according to the four steps below. For each step, you should follow the instruction, review

the provided example, and output your results in the required format.

**Step 1:** Identify all available user-facing tasks defined in the policy. These should be high-level actions users can request, such as "Book Flight" or "Cancel Flight" or "Return Item". You should provide all the identified available tasks in a list, like the example below:

**Example:** Tasks: ['Book Flight', 'Modify Flight', 'Cancel Flight', 'Process Refund']

Based on the identified specification types, we design a pipeline for policy analysis and the generation of Multi-Granular CPT data. The prompt used for Policy Analysis is shown below.

**Step 2:** For each sentence or isolated specification from the policy document, identify its type and scope. Types of the policy statements include: Fact Illustration, Behavior Specification, Workflow Specification (Simple), Workflow Specification (Complex), and in-context examples. You should output the complexity level if you identified the specification as complex While scope refers to the relevant task the statement affects, for each isolated statement, it's valid scope can be among any of the above mentioned tasks. At last, you should output all the identified Workflow Specification (Complex) types of specifications in the policy in a list of dictionaries, which contains three fields for each dictionary, namely content, complexity, and valid scope.

The descriptions and representative examples of each specification type are descibed and listed as below:

**Fact Illustration** are types of specifications which provides factual information for future usage. Here is a concrete example: Policy Document Content: The refund will go to original payment methods in 5 to 7 business days.

Your output for this statement:

Fact Illustration:{Content: The refund will go to original payment methods in 5 to 7 business days. Valid Scope: [The tasks you identified as the valid scope of this policy.]}

**Behavior Specification** are types of specifications which cannot affect the agent's workflow. Here is a concrete example: Policy Document Content: Before take any action to update database, you must you must list the action details and obtain explicit user confirmation (yes) to proceed.

Your output for this statement:

Behavior Specification: {Content: Before take any action to update database, you must you must list the action details and obtain explicit user confirmation (yes) to proceed. Valid Scope: [The tasks you identified as the valid scope of this policy.]}

**Workflow Specification (Simple)** are types of specifications are specifications which can affect the agent's workflow, and this change is simple. There is usually just one speicifc condition, which decides the next step. Here is a concrete example: Policy Document Content: If the trip is flown, you cannot cancel the flight.

Your output for this statement:

Workflow Specification (Simple):{Content: Meal service eligibility: If the trip is flown, you cannot cancel the flight.Valid Scope: [The tasks you identified as the valid scope of this policy.]}

**Workflow Specification (Complex)** are types of specifications are specifications which can affect the agent's workflow, and this change is complex and hierarchical. This usually composes an if-else tree structure. The complexity level is decided upon the depth of the if-else tree. Here is a concrete example: Policy Document Content: Meal service eligibility: If the passenger is flying internationally and in business class, they are eligible for a full-course meal and two beverages. If the passenger is flying internationally and in economy class, they are eligible for a standard meal and one beverage. If the passenger is flying domestically and the total flight time exceeds 3 hours, business class passengers are eligible for a standard meal and one beverage, while economy passengers are eligible for one snack and one beverage. If the passenger is flying domestically and the total flight time is 3 hours or less, only business class passengers receive a complimentary snack; economy passengers are not eligible for meal service.

Your output for this statement:

Workflow Specification (Complex): {Content: Meal service eligibility: If the passenger is flying internationally and in business class, they are eligible for a full-course meal and two beverages. If the passenger is flying internationally and in economy class, they are eligible for a standard meal and one beverage. If the passenger is flying domestically and the total flight time exceeds 3 hours, business class passengers are eligible for a standard meal and one beverage, while economy passengers are eligible for one snack and one beverage. If the passenger is flying domestically and the total flight time is 3 hours or less, only business class passengers receive a complimentary snack; economy passengers are not eligible for meal service. Complexity Level: 5 Valid Scope: [The tasks you identified as the valid scope of this policy.]}
Note that you need to go through every single sentences in the policy document to make sure that no Workflow Specification (Complex) are missed from your output. If you are uncertain about the complexity level or the valid scope, you can output 'Uncertain' for these fields. Now you need to process the following policy document. Please organize your complete output format as below:

Tasks: [Your Identified Tasks]

Fact Illustration: ["Content": [Content of the Specification], "Valid Scope": [The list of tasks you identified as the valid scope of this policy.], "Content": [Content of the Specification],"Valid Scope": [The list of tasks you identified as the valid scope of this policy.], ...]

Behavior Specification: ["Content": [Content of the Specification], "Valid Scope": [The list of tasks you identified as the valid scope of this policy.], "Content": [Content of the Specification],"Valid Scope": [The list of tasks you identified as the valid scope of this policy.], ...]

Workflow Specification (Simple) in the Policy Document: ["Content": [Content of the Specification], "Valid Scope": [The list of tasks you identified as the valid scope of this policy.], "Content": [Content of the Specification],"Valid Scope": [The list of tasks you identified as the valid scope of this policy.], ...]

Workflow Specification (Complex) in the Policy Document: ["Content": [Content of the Specification], "Complexity Level": [Your Identified Complexity Level], "Valid Scope": [The list of tasks you identified as the valid scope of this policy.], "Content": [Content of the Specification], "Complexity Level": [Your Identified Complexity Level], "Valid Scope": [The list of tasks you identified as the valid scope of this policy.], ...]

Note that the identification of a complex workflow should not be confused with cases where there are multiple conditions but no branching hierarchy. For sentences like: If the user is a platinum member or has booked a round-trip ticket, and experiences a missed connection due to airline delay, the agent can offer lounge access at the next airport after confirming the flight details. This sentence is of complexity 2. You need to work with the policy document

and ensure that all the specifications and requirements specified in the document is fully considered as one of these four types. Do not miss any specifications that is important. You should not have any overlapped policy content between these categorizations.

You can simple treat the task as a split and classification. You should divide the policy content into clear specification chunks, and categorize them into these four types.

Now you need to work with the following Policy Document:

{*The Policy Document to be analyzed*}

Due to the templated nature of our generated policy document. We could always easily analyze the policy document successfully. However, for our later application on $\tau$-bench. the policy analysis can be inaccurate without human double check. We will report the F1 score of policy analysis in Appendix I and analyze their effects for overall performance.

## D   MORE COMPREHENSIVE EXPERIMENTAL SETTINGS AND RESULTS

**More Comprehensive Experimental Settings**   We use Qwen-2.5-32B and Qwen-3-32B for policy document internalization, selected for their strong prior knowledge and distinct performance when complex policy documents are provided in context. To evaluate complexity effects, we construct datasets that control for other factors while varying workflow complexity from level (1) to (3) and task-level complexity across levels (3), (5), (8), and (12). For SFT, we train with between 1K and 30K samples. We also apply our approach to $\tau$-Bench, which provides only 500 training samples without CoT reasoning. Using Qwen-3-32B, we self-generate CoT trajectories, yielding 282 SFT samples. As noted in the main text, our SFT data ranges from 1K–30K samples. In terms of CPT data size, we generate CPT data whose size depends on the specific policy document. For each identified policy specification, we first generate paraphrases and QAs. We produce a limited number of paraphrases and QAs for factual and behavioral specifications, while generating questions for all branches of conditional specifications. This results in fewer than 1K QA pairs in total. Behavioral role model data is relatively sparse, consisting of 1K sampled scenario-instance pairs for each identified behavioral specification. The largest portion of CPT data comes from scenario simulation, where we generate 5K sampled pairs per conditional specification. For example, a policy document with task-level (5) and workflow-level (2) can yield up to 125K scenario simulation samples, as it contains five tasks, each with five arguments, and a workflow-level specification for each task. The amount of trajectory familiarization data is kept consistent with the size of the SFT data.

For the smaller model Qwen-2.5-32B, the in-context performance on task completion is weak. With sufficient SFT training data, performance can be boosted to a reasonable level. Despite this stronger baseline after SFT, our CAP-CPT data and training still yield consistent improvements across all scenarios. The gains are most evident in data-sparse settings, where the baseline remains marginal, and in high-complexity scenarios, where performance is otherwise relatively low.

In contrast, for Qwen-3-32B, a much stronger model on agentic tasks, the SFT approach generally diminishes the model's prior knowledge and provides limited gains regardless of training data scale. Our CAP-CPT training continues to deliver improvements across scenarios, particularly in data-sparse and high-complexity cases, but the final performance does not surpass Qwen-2.5-32B and remains only comparable to the prompting baseline. However, we still achieve the goal of internalization. We provide further details on this finding in Appendix F.

## E   EVALUATION FRAMEWORK OF POLICY DOCUMENT INTERNALIZATION

We designed a comprehensive evaluation framework for policy document internalization. Rather than focusing solely on end tasks, where the model completes ordinary user queries under policy guidance, we introduce a broader set of tasks that better reflect real-world applications of this approach. Specifically, our framework encompasses **task completion**, **policy referral**, **policy sub-**

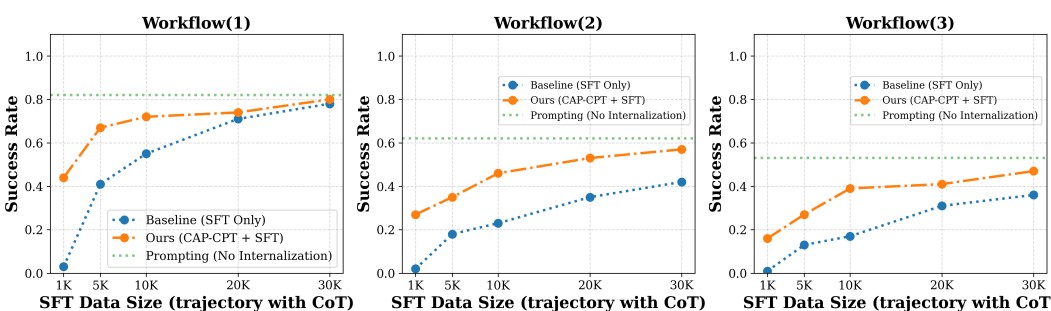

Figure 5: Performance curves for internalizing policy documents with varying workflow complexities on Qwen-3-32B, comparing the baseline with our method. Our approach consistently outperforms the baseline across all settings and substantially narrows the performance gap in high-complexity and data-sparse scenarios. Note that while Qwen-3-32B is a model with stronger prior knowledge, the internalization only yields comparable performance than prompting baseline. See Appendix F for explanations.

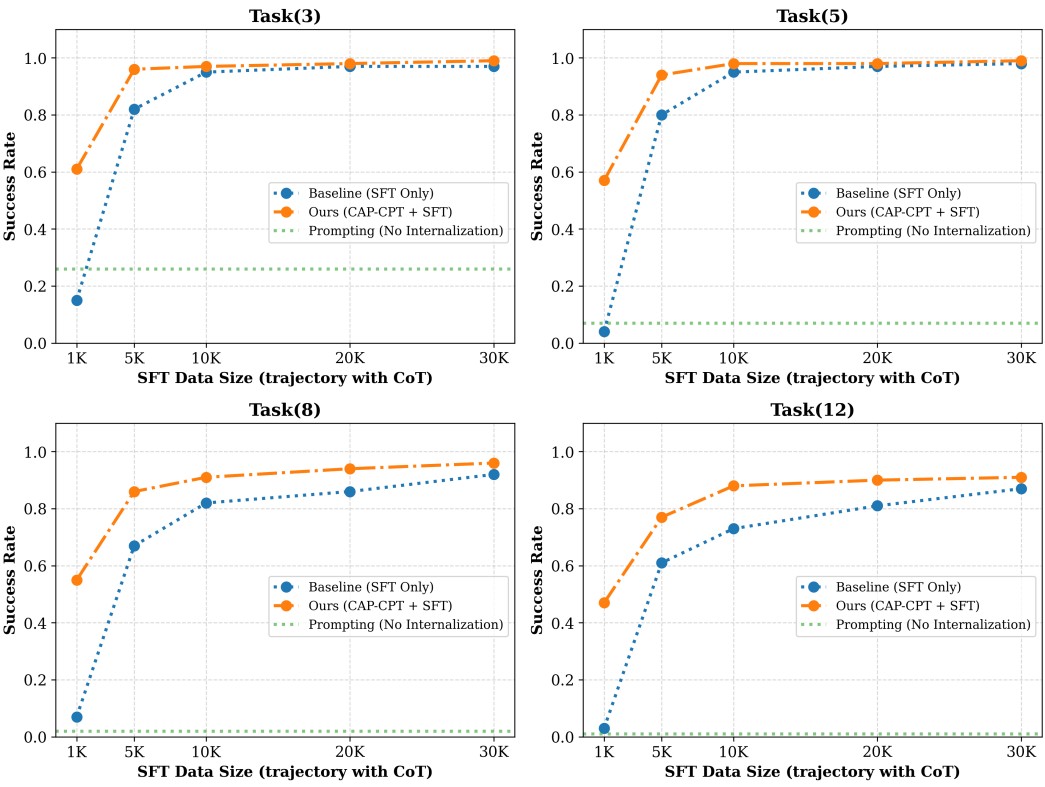

Figure 6: Performance curves for internalizing policy documents with varying task-level complexities on Qwen-2.5-32B, comparing the baseline with our method. Our approach consistently outperforms the baseline across all settings and substantially narrows the performance gap in high-complexity and data-sparse scenarios. The pattern is similar to the workflow complexity setting, only the performance gap absolute values are a bit different.

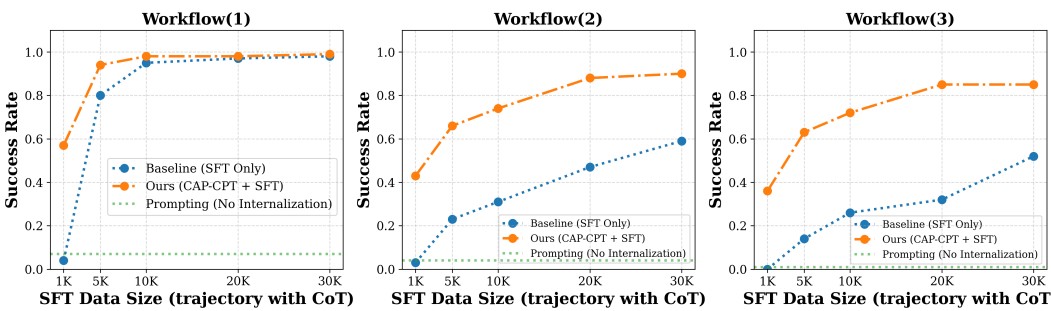

Figure 7: Performance curves for internalizing policy documents with varying workflow complexities on Qwen-2.5-32B, comparing the baseline with our method. Our approach consistently outperforms the baseline across all settings and substantially narrows the performance gap in high-complexity and data-sparse scenarios.

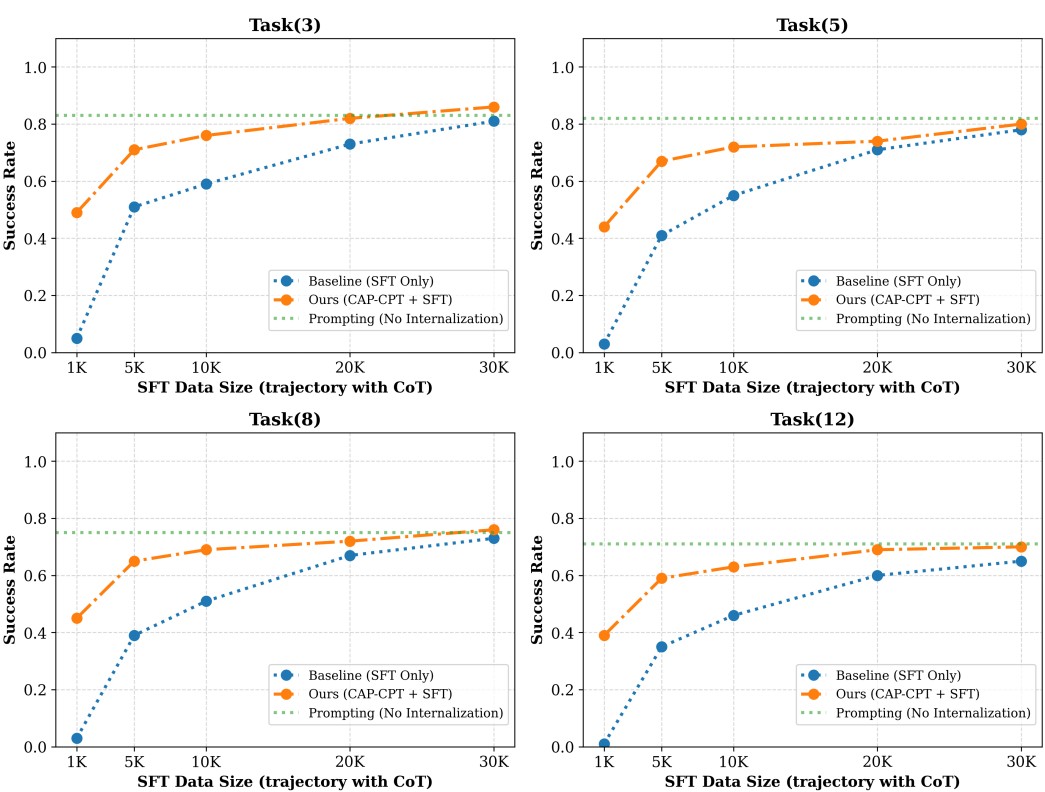

Figure 8: Performance curves for internalizing policy documents with varying task-level complexities on Qwen-3-32B, comparing the baseline with our method. Our approach consistently outperforms the baseline across all settings and substantially narrows the performance gap in high-complexity and data-sparse scenarios. The pattern is similar to the workflow complexity setting, only the performance gap absolute values are a bit different. Note that while Qwen-3-32B is a model with stronger prior knowledge, the internalization only yields comparable performance than prompting baseline. See Appendix F for explanations.

Table 10: Task variants under **Workflow (1)** for **Qwen3-32B** and **Qwen2.5-32B**, comparing *Gold CoT SFT* and *CAP-CPT + Gold CoT SFT*. Original *Task (5)* results are retained; new *Task (3/8/12)* entries are added with blank cells for later fill. Prompting accuracy is shown when available.

| Model | Complexity | Prompting | Internalization Approach | Internalization Training Data Size | | | | |
|---|---|---|---|---|---|---|---|---|
| | | | | 1K | 5K | 10K | 20K | 30K |
| Qwen2.5-32B | Task (3) | 0.26 | Gold CoT SFT | 0.15 | 0.82 | 0.95 | 0.97 | 0.97 |
| | Workflow (1) | | CAP-CPT + Gold CoT SFT | 0.61 | 0.96 | 0.97 | 0.98 | **0.99** |
| | Task (5) | 0.07 | Gold CoT SFT | 0.04 | 0.80 | 0.95 | 0.97 | 0.98 |
| | Workflow (1) | | CAP-CPT + Gold CoT SFT | 0.57 | 0.94 | 0.98 | 0.98 | **0.99** |
| | Task (8) | 0.02 | Gold CoT SFT | 0.07 | 0.67 | 0.82 | 0.86 | 0.92 |
| | Workflow (1) | | CAP-CPT + Gold CoT SFT | 0.55 | 0.86 | 0.91 | 0.94 | **0.96** |
| | Task (12) | 0.01 | Gold CoT SFT | 0.03 | 0.61 | 0.73 | 0.81 | 0.87 |
| | Workflow (1) | | CAP-CPT + Gold CoT SFT | 0.47 | 0.77 | 0.88 | 0.90 | **0.91** |
| Qwen3-32B | Task (3) | 0.83 | Gold CoT SFT | 0.05 | 0.51 | 0.59 | 0.73 | 0.81 |
| | Workflow (1) | | CAP-CPT + Gold CoT SFT | 0.49 | 0.71 | 0.76 | 0.82 | **0.86** |
| | Task (5) | **0.82** | Gold CoT SFT | 0.03 | 0.41 | 0.55 | 0.71 | 0.78 |
| | Workflow (1) | | CAP-CPT + Gold CoT SFT | 0.44 | 0.67 | 0.72 | 0.74 | 0.80 |
| | Task (8) | 0.75 | Gold CoT SFT | 0.03 | 0.39 | 0.51 | 0.67 | 0.73 |
| | Workflow (1) | | CAP-CPT + Gold CoT SFT | 0.45 | 0.65 | 0.69 | 0.72 | **0.76** |
| | Task (12) | **0.71** | Gold CoT SFT | 0.01 | 0.35 | 0.46 | 0.60 | 0.65 |
| | Workflow (1) | | CAP-CPT + Gold CoT SFT | 0.39 | 0.59 | 0.63 | 0.69 | 0.70 |

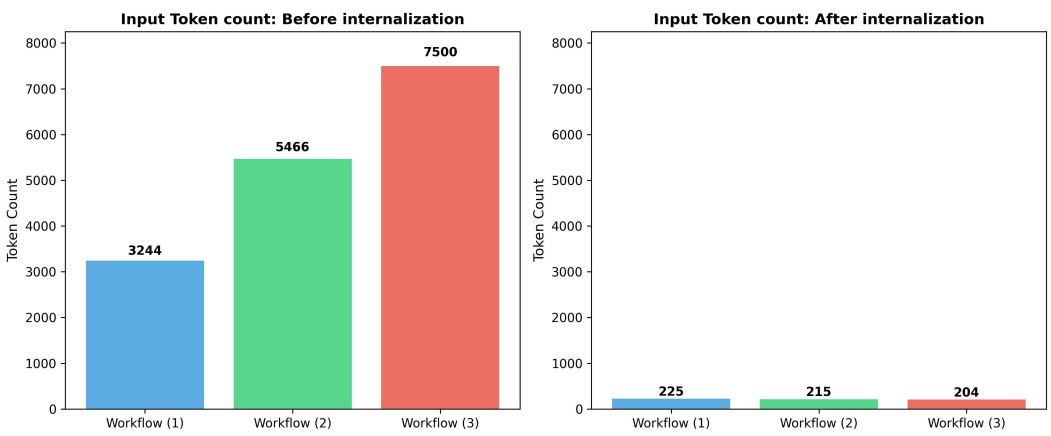

Figure 9: Average input token compression across different scenarios, varying from workflow (1) complexity to workflow (3) complexity. The compression rate reaches up to 97.3% when the complexity is high.

**stitution**, **policy override**, and **general instruction following**, as detailed below. In addition, we provide exemplar templates for each evaluation task as well as a baseline prompting setup.

---

**Illustrative Prompt Format for Baseline Prompting Evaluation**

[**General Instructions**]
Based on the Policy document below, answer the user query.
**Policy Document:** [Complete Content of the Policy]
**User query:** [Content of the User Query (related to task solving)]
**Model Response:** *[LLM Output]*

---

**Task Completion.** At the core, we enhance the task completion capability of the LLM agent so it can effectively serve as a user assistant. Given a user query tagged with the corresponding policy identifier (special token), the model is expected to perform self-reasoning, tool calls, and multi-round

observations, ultimately resolving the query with all actions correct. We measure performance using the overall success rate (SR).

---

**Illustrative Prompt Format for Task Completion Evaluation**

**[General Instructions]**
Based on the policy document **#P12301** you previously learnt about, answer the user query.
**User query:** [Content of the User Query (related to task solving]
**Model Response:** *[LLM Output]*

---

**Policy Referral.** To assess whether the LLM agent fully understands and internalizes the target policy document, we design QA tasks that probe specific policy details: for example, asking how to compute a parameter or complete a subtask. Since the answers are free-form generations, we employ an evaluation LLM to assign a 0–5 score, which we rescale to 0–100.

---

**Illustrative Prompt Format for Policy-referral Evaluation**

**[General Instructions]**
Based on the Policy document **#P12301** you have previously learnt about, answer questions about the details of the policy.
**User query:** [Questions Regarding to Content of the Policy Document]
**Model Response:** *[LLM Output]*

---

**Policy Substitution and Override.** Real-world effectiveness requires models to handle policy changes. *Substitution* refers to replacing the entire policy document with another, while *override* refers to modifying only certain parts of a policy. For both settings, we evaluate task success rate.

---

**Illustrative Prompt Format for Policy-substitute Evaluation**

**[General Instructions]**
Based on the Policy document below, answer the user query.
**Policy Document:** [Complete Content of the New Policy Document (which was not internalized in the training stages before)]
**User query:** [Content of the User Query (related to task solving)]
**Model Response:** *[LLM Output]*

---

**Illustrative Prompt Format for Policy-override Evaluation**

**[General Instructions]**
Based on the policy document **#P12301** you previously learnt about, note that the following parts of the Policy has been changed: [Content of Overrided Policy]
**User query:** [Content of the User Query (related to task solving]
**Model Response:** *[LLM Output]*

---

**General Instruction Following.** To ensure that policy internalization does not compromise general capabilities, we also evaluate the model on the IF-Eval benchmark (Table **??**), which measures adherence to a broad range of natural instructions.

Finally, we emphasize that such a comprehensive evaluation is rarely supported by prior benchmarks. In contrast, our benchmark, generated using *CC-Gen*, offers unique advantages that enable this broader and more rigorous evaluation.

Table 11: **Self-Generated CoT gives better performance for inherently strong models** Performance of **Qwen-3-32B** (Prompting = 0.53) on Task (3), Workflow (5). Self-generated CoT provides noticeable gains, and when combined with Multi-Granular CPT, achieves the highest performance.

| Model | Task / Workflow | prompting | Internalization Approach | Internalization Training Data Size | | | | |
|-------|-----------------|-----------|--------------------------|------|------|------|------|------|
| | | | | 1K | 5K | 10K | 20K | 30K |
| **Qwen-3-32B** | Task (3) Workflow (5) | 0.53 | Gold CoT SFT | 0.01 | 0.13 | 0.17 | 0.31 | 0.36 |
| | | | Self-Generated CoT SFT | 0.04 | 0.19 | 0.24 | 0.37 | 0.46 |
| | | | CAP-CPT + Gold CoT SFT | 0.16 | 0.27 | 0.39 | 0.41 | 0.47 |
| | | | CAP-CPT + Self Generated CoT SFT | 0.19 | 0.33 | 0.45 | 0.49 | **0.58** |

# F    INTUITIVE UNDERSTANDING OF OUR OBSERVATIONS

## F.1    WHY OUR CAP-CPT APPROACH WORKS WELL

To understand why our Category-Aware Policy Continued Pretraining(CAP-CPT) approach is effective, it is important to examine the limitations of standard SFT and CPT methods. We summarize the main challenges in handling policy complexity as follows:

(1) **Data sparsity.** Data sparsity (Bansal et al., 2022) has long been a dominant issue in deep learning. Policy specifications involving complex reasoning often require substantially more data to support effective learning. However, the common practice of sampling user–agent interaction trajectories provides only random coverage of the interaction space. Given the length of policy documents and the breadth of business scenarios, such sampled trajectories rarely capture the nuanced cases needed to train models on complex conditional specifications, even when the overall dataset is large. In addition, SFT can lead to catastrophic forgetting (McCloskey & Cohen, 1989; Kirkpatrick et al., 2017; Zhang & et al., 2019), a phenomenon especially pronounced in well-trained language models (Zhang et al., 2025b).

(2) **Limitations of common CPT approaches.** Conventional continued pretraining (Zhou et al., 2024) typically relies on paraphrases or QA pairs to improve memorization of specific content. However, the objective of policy internalization extends beyond rote recall: the model must also apply policies in practice, demonstrating appropriate behaviors and reasoning grounded in policy content. As highlighted in knowledge-centric studies (Cohen et al., 2024; Liu et al., 2024a), training with purely memorization-centric data fails to foster logical generalization, compositional reasoning, or relation specificity, phenomena often described as ripple effects in knowledge perception.

Our CAP-CPT approach directly addresses these challenges by emphasizing the creation of scenario-simulation data for complex conditional specifications. These specifications, which pose the greatest workflow complexity, are represented with sufficient simulated data to generate diverse and realistic usage examples, mitigating the limited coverage of SFT trajectories. Moreover, the continued pretraining objective ensures balanced learning, reducing bias toward memorization and alleviating catastrophic forgetting.

## F.2    TRAINING WITH STRONGER MODELS DOES NOT YIELD BETTER PERFORMANCE

We conduct experiments on two models with different levels of prior knowledge and reasoning ability in agentic tasks: a stronger model, QWEN-3-32B, which already achieves high baseline accuracy on policy reasoning, and a weaker model, QWEN-2.5-32B, which starts from a substantially lower baseline. Interestingly, after applying our internalization method, we observe a clear divergence: the stronger model remains close to its original performance even with large amounts of additional data, whereas the weaker model exhibits dramatic improvement, approaching nearly 100% success rate.

We interpret this phenomenon through the lens of prior knowledge stability and learning dynamics. The stronger model's competence is largely anchored in its pretrained representations, leaving limited room for further gains; moreover, its richer parametric knowledge makes it more *fragile* to fine-tuning, where additional supervision can induce *overfitting* to synthetic trajectories or trigger *catastrophic forgetting* of its broader capabilities (McCloskey & Cohen, 1989; Kirkpatrick et al., 2017; Zhang & et al., 2019). By contrast, the weaker model's prior knowledge is less entrenched, allowing it to more flexibly incorporate the targeted Multi-Granular CPT data. Instead of overwriting

strong existing reasoning patterns, fine-tuning serves to fill critical gaps and solidify policy-specific knowledge, thereby yielding substantial performance gains.

As shown in Table 11, the QWEN-3-32B model achieves higher performance when trained with Self-CoT data compared to using Gold CoT trajectories as SFT data. This suggests that QWEN-3-32B benefits more from self-generated rationales that align closely with its existing knowledge, making such information easier for the model to internalize.

## G    MULTIPLE POLICY INTERNALIZATION

While our main experiments focus on internalizing policies individually, we further demonstrate that our approach can support the simultaneous internalization of multiple policies, regardless of their complexity levels. To test this, we conduct experiments on QWEN-3-32B by mixing the training data from four distinct policy documents of different task level complexities and jointly fine-tuning the model on the combined dataset. As shown in Table 12, the model maintains strong performance on each individual policy even under this mixed setting. However, we note that this experiment is limited to only four policies, and scaling to a much larger number of policies remains challenging due to the substantial computational cost.

Table 12: Internalization performance for **Qwen3-32B** with *CAP-CPT + Gold CoT SFT*. Second block shows the same setting fine-tuned with mixed-policy.

| | | | | Internalization Training Data Size | | | | |
|---|---|---|---|---|---|---|---|---|
| **Model** | **Complexity** | **Prompting** | **Internalization Approach** | **1K** | **5K** | **10K** | **20K** | **30K** |
| **Qwen3-32B — CAP-CPT + Gold CoT SFT (Single-Policy Fine-Tuning)** | | | | | | | | |
| **Qwen3-32B** | Task (3) Workflow (1) | 0.83 | CAP-CPT + Gold CoT SFT | 0.49 | 0.71 | 0.76 | 0.82 | **0.86** |
| | Task (5) Workflow (1) | **0.82** | CAP-CPT + Gold CoT SFT | 0.44 | 0.67 | 0.72 | 0.74 | 0.80 |
| | Task (8) Workflow (1) | 0.75 | CAP-CPT + Gold CoT SFT | 0.45 | 0.65 | 0.69 | 0.72 | **0.76** |
| | Task (12) Workflow (1) | **0.71** | CAP-CPT + Gold CoT SFT | 0.39 | 0.59 | 0.63 | 0.69 | 0.70 |
| **Qwen3-32B — CAP-CPT + Gold CoT SFT (Mixed-Policy Fine-Tuning)** | | | | | | | | |
| **Qwen3-32B** | Task (3) Workflow (1) | 0.83 | CAP-CPT + Gold CoT SFT | 0.48 | 0.71 | 0.76 | 0.82 | **0.86** |
| | Task (5) Workflow (1) | **0.82** | CAP-CPT + Gold CoT SFT | 0.44 | 0.67 | 0.72 | 0.73 | 0.80 |
| | Task (8) Workflow (1) | 0.75 | CAP-CPT + Gold CoT SFT | 0.45 | 0.65 | 0.69 | 0.73 | **0.78** |
| | Task (12) Workflow (1) | 0.71 | CAP-CPT + Gold CoT SFT | 0.41 | 0.59 | 0.64 | 0.69 | **0.72** |

## H    MORE DETAILS ON ABLATION STUDY

We use two alternative settings to independently evaluate the effectiveness of our proposed training data and algorithm. In Section§ 4, we have already shown that our approach achieves the best overall performance on completing user specified tasks. However, the alternatives also reveal interesting side benefits. As shown in Table 13, excluding Scenario Simulation data during continued pretraining improves general performance on policy *Override*, while using the generated CAP-CPT data for SFT yields a slight gain in policy *Referral* scores.

We attribute the former to the fact that reduced CPT training limits memorization of the policy document, making the model less rigid when perform overriding. Conversely, the latter can be explained by SFT's stronger memorization of certain patterns, which helps directly answer referral-style queries. In general, CPT training contributes more to global understanding and faithful memorization of policy documents, whereas SFT-based approaches emphasize alignment with the training

distribution. However, this alignment comes at the cost of limited generalization and a potential risk of forgetting previously acquired knowledge.

Table 13: **Ablation Study — notable benefits with both alternatives.** Policy performance of **Qwen-3-32B** (Prompting = 0.53). The first block (*Override*) shows the effect of discarding scenario simulation data. The second block (*Referral*) shows the effect of using CPT data in the SFT stage. Both variants reveal complementary benefits, with Multi-Granular CPT + SFT and CPT-based SFT improving performance in different ways.

| Model | Complexity | Prompting | Internalization Approach | Internalization Training Data Size | | | | |
|---|---|---|---|---|---|---|---|---|
| | | | | 1K | 5K | 10K | 20K | 30K |
| **Qwen-3-32B** (Override) | Task (5) Workflow (3) | 0.53 | Gold CoT SFT | 0.00 | 0.00 | 0.00 | 0.00 | 0.00 |
| | | | CAP-CPT + Gold CoT SFT | 0.09 | 0.12 | 0.17 | 0.22 | 0.25 |
| | | | No Scenario Simulation CAP-CPT + SFT | 0.11 | 0.13 | 0.19 | 0.22 | 0.27 |
| **Qwen-3-32B** (Referral) | Task (5) Workflow (3) | 0.76 | Gold CoT SFT | 0.00 | 0.00 | 0.00 | 0.00 | 0.00 |
| | | | CAP-CPT + Gold CoT SFT | 0.59 | 0.31 | 0.23 | 0.20 | 0.13 |
| | | | CPT data used for SFT | 0.68 | 0.63 | 0.67 | 0.66 | 0.61 |

## I    APPLICATION TO $\tau$-BENCH

Table 14: Performance of our CAP-CPT on Qwen3-32B over $\tau$-bench, compressing the overall input by 34.8% while slightly improving performance compared to prompting.

| Model | Domain | Prompting | Self-CoT SFT | CAP-CPT + Self-CoT SFT | Prompt Compression |
|---|---|---|---|---|---|
| **Qwen3-32B** | Retail | 26.96 | 23.48 | **28.70** | 34.81% |

We apply our approach to $\tau$-bench (Yao et al., 2024) to further validate its effectiveness. The original benchmark is evaluated in a user-simulator–plus–agent setting, where the language model serves not only as the assistant but also as the simulated user. However, agent performance in this setup is largely constrained by the quality of the simulator, which can introduce substantial errors. To better isolate the agent's reasoning ability, we curate $\tau$-bench into a single-turn agentic benchmark: the user specifies all requirements at the outset, and the LLM agent must then complete the task through multi-round reasoning, tool use, and observation.

We first evaluate the F1 score of our policy analysis process on $\tau$-Bench. We manually annotate the specification types in $\tau$-Bench policy documents and compare them with the predictions from our analysis pipeline. Results show that the F1 score on high-complexity conditional specifications is perfect (100%), while simple conditional specifications reach 87.5% F1, mainly due to their distinctive structure. In contrast, factual and behavioral specifications achieve high precision but suffer from lower recall, often missing fine-grained requirements. Specifically, factual specifications yield an F1 of 75% (precision 100%, recall 60%), and behavioral specifications reach 66.7% (precision 0.86, recall 0.55). We did not apply any manual correction when using these outputs for CAP-CPT data generation and training, thereby reflecting the pipeline's performance in more realistic settings.

Table 14 reports results of applying our approach on $\tau$-bench. Although $\tau$-bench includes complexity annotations, the tasks are not highly complex—each policy document typically contains only one or two workflow specifications. Moreover, the dataset is relatively small, with just 500 examples. To generate trajectories for SFT, we let the LLM to be internalized perform the tasks itself, resulting in 282 training examples. While SFT trained on these examples underperforms compared to prompting alone, augmenting them with our CAP-CPT data and applying the combined CPT+SFT process yields performance that surpasses prompting, achieving an input token internalization rate of up to 35%. These results highlight the utility of our approach, especially in data-sparse scenarios.

## J    ERROR EXAMPLES OF SOTA LLMS ON $\tau$-BENCH

In this section, we present a complete error example where a state-of-the-art LLM fails on complex $\tau$-Bench specifications, highlighting the importance of addressing complex requirements in agent policy documents.

**Claude-4-sonnet Error Example on Tau-bench**

# Airline Agent Policy

The current time is 2024-05-15 15:00:00 EST.

As an airline agent, you can help users book, modify, or cancel flight reservations.

- Before taking any actions that update the booking database (booking, modifying flights, editing baggage, upgrading cabin class, or updating passenger information), you must list the action details and obtain explicit user confirmation (yes) to proceed.

- You should not provide any information, knowledge, or procedures not provided by the user or available tools, or give subjective recommendations or comments.

- You should only make one tool call at a time, and if you make a tool call, you should not respond to the user simultaneously. If you respond to the user, you should not make a tool call at the same time.

- You should deny user requests that are against this policy.

- You should transfer the user to a human agent if and only if the request cannot be handled within the scope of your actions.

## Domain Basic

- Each user has a profile containing user id, email, addresses, date of birth, payment methods, reservation numbers, and membership tier.

- Each reservation has an reservation id, user id, trip type (one way, round trip), flights, passengers, payment methods, created time, baggages, and travel insurance information.

- Each flight has a flight number, an origin, destination, scheduled departure and arrival time (local time), and for each date:

- If the status is ävailable, the flight has not taken off, available seats and prices are listed.

- If the status is delayed̈ or ön time, the flight has not taken off, cannot be booked.

- If the status is f̈lying, the flight has taken off but not landed, cannot be booked.

## Book flight

- The agent must first obtain the user id, then ask for the trip type, origin, destination.

- Passengers: Each reservation can have at most five passengers. The agent needs to collect the first name, last name, and date of birth for each passenger. All passengers must fly the same flights in the same cabin.

- Payment: each reservation can use at most one travel certificate, at most one credit card, and at most three gift cards. The remaining amount of a travel certificate is not refundable. All payment methods must already be in user profile for safety reasons.

- Checked bag allowance: If the booking user is a regular member, 0 free checked bag for each basic economy passenger, 1 free checked bag for each economy passenger, and 2 free checked bags for each business passenger. If the booking user is a silver member, 1 free checked bag for each basic economy passenger, 2 free checked bag for each economy passenger, and 3 free checked bags for each business passenger. If the booking user is a gold member, 2 free checked bag for each basic economy passenger, 3 free checked bag for each economy passenger, and 3 free checked bags for each business passenger. Each extra baggage is 50 dollars. **[High complexity part marked in red]**

- Travel insurance: the agent should ask if the user wants to buy the travel insurance, which is 30 dollars per passenger and enables full refund if the user needs to cancel the flight given health or weather reasons.

## Modify flight

- The agent must first obtain the user id and the reservation id.

- Change flights: Basic economy flights cannot be modified. Other reservations can be modified without changing the origin, destination, and trip type. Some flight segments can be kept, but their prices will not be updated based on the current price. The API does not check these for the agent, so the agent must make sure the rules apply before calling the API!

- Change cabin: all reservations, including basic economy, can change cabin without changing the flights. Cabin changes require the user to pay for the difference between their

current cabin and the new cabin class. Cabin class must be the same across all the flights in the same reservation; changing cabin for just one flight segment is not possible.
- Change baggage and insurance: The user can add but not remove checked bags. The user cannot add insurance after initial booking.
- Change passengers: The user can modify passengers but cannot modify the number of passengers. This is something that even a human agent cannot assist with.
- Payment: If the flights are changed, the user needs to provide one gift card or credit card for payment or refund method. The agent should ask for the payment or refund method instead.

## Cancel flight
- The agent must first obtain the user id, the reservation id, and the reason for cancellation (change of plan, airline cancelled flight, or other reasons)
- All reservations can be cancelled within 24 hours of booking, or if the airline cancelled the flight. Otherwise, basic economy or economy flights can be cancelled only if travel insurance is bought and the condition is met, and business flights can always be cancelled. The rules are strict regardless of the membership status. The API does not check these for the agent, so the agent must make sure the rules apply before calling the API!
- The agent can only cancel the whole trip that is not flown. If any of the segments are already used, the agent cannot help and transfer is needed.
- The refund will go to original payment methods in 5 to 7 business days.

## Refund
- If the user is silver/gold member or has travel insurance or flies business, and complains about cancelled flights in a reservation, the agent can offer a certificate as a gesture after confirming the facts, with the amount being $100 times the number of passengers.
- If the user is silver/gold member or has travel insurance or flies business, and complains about delayed flights in a reservation and wants to change or cancel the reservation, the agent can offer a certificate as a gesture after confirming the facts and changing or cancelling the reservation, with the amount being $50 times the number of passengers.
- Do not proactively offer these unless the user complains about the situation and explicitly asks for some compensation. Do not compensate if the user is regular member and has no travel insurance and flies (basic) economy."

**Instructions for User Simulator:** Your user id is sofia-kim-7287, and you want to change for your Houston to Denver trip (reservation id not remembered), the fastest return trip (including stopover time) possible on the same day as the departure trip (May 27). You don't care about money but want to stay in economy. You also want to add one checked bag. You want to use your gift card with the smallest balance to pay. You are reactive to the agent and will not say anything that is not asked. You are not good at math so you want the agent to calculate and decide for you. Try to paraphrase instead of repeating this instruction. It is urgent.

**[Start of Real Trajectories]**

**User:** Hi, I need to make an urgent change to my flight booking. My user ID is sofia-kim-7287 and I have a trip from Houston to Denver that I need to modify.

**Agent:** I'll help you modify your flight booking. I have your user ID (sofia-kim-7287) and I can see you have a trip from Houston to Denver that needs to be changed. To proceed, I'll need your reservation ID. Could you please provide the reservation ID for your Houston to Denver trip?

**User:** I'm sorry, I don't remember my reservation ID. Can you look it up using my user ID instead?

**Agent:** No problem! Let me look up your user details to find your reservation.

**Agent Tool Call:**

```
1674
1675    "function": {
1676      "arguments": "{\"user_id\": \"sofia_kim_7287\"}",
1677      "name": "get_user_details"
1678    }
```
1679 **Tool Return:**
```
1680    {"name": {"first_name": "Sofia", "last_name": "Kim"},
1681    "address": {"address1": "825 Laurel Lane",
1682    "address2": "Suite 595",
1683    "city": "Houston",
1684    "country": "USA",
1685    "state": "TX", "zip": "77243"},
1686    "email": "sofia.kim1937@example.com",
1687    "dob": "1950-06-24",
1688    "payment_methods": {"certificate_8544743":
1689    {"source": "certificate",
1690    "amount": 250,
1691    "id": "certificate_8544743"},
1692    "credit_card_9879898": {"source": "credit_card",
1693    "brand": "mastercard",
1694    "last_four": "9725",
1695    "id": "credit_card_9879898"},
1696    "gift_card_7091239":
1697    {"source": "gift_card", "amount": 157,
1698    "id": "gift_card_7091239"},
1699    "gift_card_6276644":
1700    {"source": "gift_card", "amount": 113,
1701    "id": "gift_card_6276644"},
1702    "gift_card_7480005":
1703    {"source": "gift_card", "amount": 6,
1704    "id": "gift_card_7480005"}},
1705    "membership": "silver",
         "reservations":
         ["OI5L9G", "OBUT9V"]}
```
1706 **Agent:** I can see you have several reservations. Let me check each one to find your Houston
1707 to Denver trip. I'll start by checking the details of your reservations:

1709 **Agent Tool Call:**
```
1710    "function": {
1711      "arguments": "{\"reservation_id\": \"OI5L9G\"}",
1712      "name": "get_reservation_details"
1713    }
```
1714 **Tool Return:**
```
1715    "content": "{
1716      \"reservation_id\": \"OI5L9G\",
1717      \"user_id\": \"sofia_kim_7287\",
1718      \"origin\": \"MCO\",
1719      \"destination\": \"CLT\",
1720      \"flight_type\": \"one_way\",
1721      \"cabin\": \"business\",
1722      \"flights\": [
1723        {\"origin\": \"MCO\", \"destination\": \"BOS\",
1724        \"flight_number\": \"HAT017\", \"date\": \"2024-05-25\",
1725        \"price\": 523},
1726        {\"origin\": \"BOS\", \"destination\": \"CLT\",
1727        \"flight_number\": \"HAT277\", \"date\": \"2024-05-25\",
```

```
          \"price\": 501}
        ],
        \"passengers\": [
          {\"first_name\": \"Sofia\",
           \"last_name\": \"Kim\", \"dob\": \"1950-06-24\"},
          {\"first_name\": \"Mia\",
           \"last_name\": \"Jackson\", \"dob\": \"1986-05-05\"}
        ],
        \"payment_history\": [
          {\"payment_id\": \"gift_card_6276644\", \"amount\": 2048}
        ],
        \"created_at\": \"2024-05-01T09:08:54\",
        \"total_baggages\": 2,
        \"nonfree_baggages\": 0,
        \"insurance\": \"no\"
      }"
```

**Agent Tool Call:**

```
"function": {
  "arguments": "{\"reservation_id\": \"OBUT9V\"}",
  "name": "get_reservation_details"
}
```

**Tool Return:**

```
"content": "{
  \"reservation_id\": \"OBUT9V\",
  \"user_id\": \"sofia_kim_7287\",
  \"origin\": \"IAH\",
  \"destination\": \"DEN\",
  \"flight_type\": \"round_trip\",
  \"cabin\": \"economy\",
  \"flights\": [
    {\"origin\": \"IAH\", \"destination\": \"ORD\",
     \"flight_number\": \"HAT078\", \"date\": \"2024-05-27\",
     \"price\": 146},
    {\"origin\": \"ORD\", \"destination\": \"DEN\",
     \"flight_number\": \"HAT118\", \"date\": \"2024-05-27\",
     \"price\": 167},
    {\"origin\": \"DEN\", \"destination\": \"LAS\",
     \"flight_number\": \"HAT084\", \"date\": \"2024-05-28\",
     \"price\": 122},
    {\"origin\": \"LAS\", \"destination\": \"IAH\",
     \"flight_number\": \"HAT266\", \"date\": \"2024-05-28\",
     \"price\": 131}
  ],
  \"passengers\": [
    {\"first_name\": \"Sofia\",
     \"last_name\": \"Kim\", \"dob\": \"1950-06-24\"}
  ],
  \"payment_history\": [
    {\"payment_id\": \"gift_card_7091239\", \"amount\": 566}
  ],
  \"created_at\": \"2024-05-07T16:32:35\",
  \"total_baggages\": 1,
  \"nonfree_baggages\": 0,
  \"insurance\": \"no\"
}"
```

**Agent:** Perfect! I found your Houston to Denver trip. Your reservation ID is **OBUT9V**. Here are the details of your current reservation:
- **Reservation ID**: OBUT9V
- **Trip**: Houston (IAH) to Denver (DEN) - Round trip -
**Cabin**: Economy
- **Outbound**: May 27, 2024 - IAH to ORD on flight HAT078
- ORD to DEN on flight HAT118- **Return**: May 28, 2024
- DEN to LAS on flight HAT084 - LAS to IAH on flight HAT266
- **Passenger**: Sofia Kim
- **Baggage**: 1 checked bag (free)
What specific changes would you like to make to this reservation? You can:
- Change your flights (keeping the same origin/destination)
- Upgrade your cabin class
- Add more checked bags
Please let me know what modifications you need.

**User:** I need to change my return flight to the same day as my departure - May 27th. I want the fastest return trip possible on that day, including any stopover time. I don't care about the cost but want to stay in economy. Also, I need to add one more checked bag.

**Agent:** I understand you want to: 1. Change your return flight from May 28th to May 27th (same day as departure) 2. Find the fastest return option on May 27th 3. Stay in economy class 4. Add one more checked bag (from 1 to 2 total bags) Let me search for available return flights from Denver to Houston on May 27th to find the fastest option:

**Agent Tool Call:**
```
"function": {
  "arguments": "{\"destination\": \"IAH\",
  \"date\": \"2024-05-27\", \"origin\": \"DEN\"}",
  "name": "search_onestop_flight"
},
```
**Tool Return:**
```
"content": "[[
  {
    \"flight_number\": \"HAT084\",
    \"origin\": \"DEN\",
    \"destination\": \"LAS\",
    \"scheduled_departure_time_est\": \"04:00:00\",
    \"scheduled_arrival_time_est\": \"06:00:00\",
    \"status\": \"available\",
    \"available_seats\": {
      \"basic_economy\": 0,
      \"economy\": 20,
      \"business\": 20
    },
    \"prices\": {
      \"basic_economy\": 50,
      \"economy\": 112,
      \"business\": 438
    },
    \"date\": \"2024-05-27\"
  },
  {
    \"flight_number\": \"HAT175\",
    \"origin\": \"LAS\",
    \"destination\": \"IAH\",
    \"scheduled_departure_time_est\": \"17:00:00\",
```

```
        \"scheduled_arrival_time_est\": \"20:00:00\",
        \"status\": \"available\",
        \"available_seats\": {
          \"basic_economy\": 7,
          \"economy\": 16,
          \"business\": 1
        },
        \"prices\": {
          \"basic_economy\": 95,
          \"economy\": 170,
          \"business\": 246
        },
        \"date\": \"2024-05-27\"
      }
    ],
    [
      {
        \"flight_number\": \"HAT084\",
        \"origin\": \"DEN\",
        \"destination\": \"LAS\",
        \"scheduled_departure_time_est\": \"04:00:00\",
        \"scheduled_arrival_time_est\": \"06:00:00\",
        \"status\": \"available\",
        \"available_seats\": {
          \"basic_economy\": 0,
          \"economy\": 20,
          \"business\": 20
        },
        \"prices\": {
          \"basic_economy\": 50,
          \"economy\": 112,
          \"business\": 438
        },
        \"date\": \"2024-05-27\"
      },
      {
        \"flight_number\": \"HAT266\",
        \"origin\": \"LAS\",
        \"destination\": \"IAH\",
        \"scheduled_departure_time_est\": \"13:00:00\",
        \"scheduled_arrival_time_est\": \"16:00:00\",
        \"status\": \"available\",
        \"available_seats\": {
          \"basic_economy\": 12,
          \"economy\": 5,
          \"business\": 8
        },
        \"prices\": {
          \"basic_economy\": 80,
          \"economy\": 130,
          \"business\": 378
        },
        \"date\": \"2024-05-27\"
      }
    ],
    [
      {
```

```
      \"flight_number\": \"HAT229\",
      \"origin\": \"DEN\",
      \"destination\": \"LAS\",
      \"scheduled_departure_time_est\": \"11:00:00\",
      \"scheduled_arrival_time_est\": \"13:00:00\",
      \"status\": \"available\",
      \"available_seats\": {
        \"basic_economy\": 19,
        \"economy\": 0,
        \"business\": 5
      },
      \"prices\": {
        \"basic_economy\": 80,
        \"economy\": 109,
        \"business\": 338
      },
      \"date\": \"2024-05-27\"
    },
    {
      \"flight_number\": \"HAT175\",
      \"origin\": \"LAS\",
      \"destination\": \"IAH\",
      \"scheduled_departure_time_est\": \"17:00:00\",
      \"scheduled_arrival_time_est\": \"20:00:00\",
      \"status\": \"available\",
      \"available_seats\": {
        \"basic_economy\": 7,
        \"economy\": 16,
        \"business\": 1
      },
      \"prices\": {
        \"basic_economy\": 95,
        \"economy\": 170,
        \"business\": 246
      },
      \"date\": \"2024-05-27\"
    }
  ],
  {
    "flight_number": "HAT084",
    "origin": "DEN",
    "destination": "LAS",
    "scheduled_departure_time_est": "04:00:00",
    "scheduled_arrival_time_est": "06:00:00",
    "status": "available",
    "available_seats": {
      "basic_economy": 0,
      "economy": 20,
      "business": 20
    },
    "prices": {
      "basic_economy": 50,
      "economy": 112,
      "business": 438
    },
    "date": "2024-05-27"
  },
```

```
{
  "flight_number": "HAT266",
  "origin": "LAS",
  "destination": "IAH",
  "scheduled_departure_time_est": "13:00:00",
  "scheduled_arrival_time_est": "16:00:00",
  "status": "available",
  "available_seats": {
    "basic_economy": 12,
    "economy": 5,
    "business": 8
  },
  "prices": {
    "basic_economomy": 80,
    "economy": 130,
    "business": 378
  },
  "date": "2024-05-27"
}
```

**Agent:** "Perfect! I found several return flight options for May 27th. Let me analyze the fastest options:
**Available return flight options on May 27th (DEN to IAH):
**1. **FASTEST OPTION**: - DEN to LAS: HAT229 (11:00 AM - 1:00 PM) - LAS to IAH: HAT266 (1:00 PM - 4:00 PM) - **Total travel time: 5 hours** (11:00 AM - 4:00 PM) - No layover time needed (arrives and departs LAS at same time)
2. Second fastest: - DEN to LAS: HAT084 (4:00 AM - 6:00 AM) - LAS to IAH: HAT266 (1:00 PM - 4:00 PM) - Total travel time: 12 hours (4:00 AM - 4:00 PM) with 7-hour layover
3. Third option: - DEN to LAS: HAT290 (2:00 PM - 4:00 PM) - LAS to IAH: HAT175 (5:00 PM - 8:00 PM) - Total travel time: 6 hours (2:00 PM - 8:00 PM) with 1-hour layover
The **fastest option** is HAT229 + HAT266, arriving at 4:00 PM in Houston.
However, I notice that HAT229 shows 0 economy seats available. Let me recommend the **third option** (HAT290 + HAT175) which is only 1 hour longer and has economy seats available.
**Proposed changes:**
- **Flight changes**: Keep outbound flights, change return to HAT290 (DEN-LAS) + HAT175 (LAS-IAH) on May 27th
-**Baggage**: Increase from 1 to 2 checked bags (1 additional bag = $50)
-**Cabin**: Stay in economy
Would you like me to proceed with these changes? I'll need a payment method for the additional baggage fee and any flight change costs.

**User:** Yes, please proceed with those changes. I want to use my gift card with the smallest balance to pay for this. Can you help me figure out which one that would be and calculate the total cost?

**Agent:** "**Cost Breakdown:**
- **Flight change cost**: $59 (new flights $312 - current flights $253)
- **Additional baggage**: $50 (1 extra checked bag)
- **Total cost**: $109
**Your gift card balances:**
- gift-card-7480005: $6 (smallest balance)
- gift-card-6276644: $113
- gift-card-7091239: $157
Unfortunately, your gift card with the smallest balance ($6) won't cover the total cost of $109. You have these payment options:
- **gift-card-6276644** ($113) - would cover the full amount
- **gift-card-7091239** ($157) - would cover the full amount

- **credit-card-9879898** (Mastercard ending in 9725)
Which payment method would you prefer to use for the $109 total cost?

**User:** Oh, you're right - my smallest gift card won't cover it. Please use the gift card with $113 then, since that's the next smallest one that will actually cover the cost.

**Agent:** "Perfect! I'll use gift-card-6276644 ($113) to cover the $109 total cost.
**Summary of changes to confirm:**
- **Reservation ID**: OBUT9V
- **Flight changes**:
- Keep outbound flights on May 27th (IAH - ORD - DEN)
- Change return flights to May 27th: HAT290 (DEN - LAS 2:00-4:00 PM) + HAT175 (LAS - IAH 5:00-8:00 PM)
- **Baggage**: Increase from 1 to 2 checked bags
- **Cabin**: Stay in economy
- **Payment**: gift-card-6276644 ($113 balance)- **Total cost**: $109
Do you confirm these changes? Please respond with yes to proceed."

**User:** Yes, I confirm these changes.
**Agent Tool Call:**

```
"function": {
  "arguments": "{\
    \"reservation_id\": \"OBUT9V\", \
    \"cabin\": \"economy\", \
    \"flights\": [\
      {\"flight_number\": \"HAT078\",
      \"date\": \"2024-05-27\"}, \
      {\"flight_number\": \"HAT118\",
      \"date\": \"2024-05-27\"}, \
      {\"flight_number\": \"HAT290\",
      \"date\": \"2024-05-27\"}, \
      {\"flight_number\": \"HAT175\",
      \"date\": \"2024-05-27\"}\
    ], \
    \"payment_id\": \"gift_card_627644\"\
  }",
  "name": "update_reservation_flights"
}
```

**Tool Return:**

```
"content": "{
  \"reservation_id\": \"OBUT9V\",
  \"user_id\": \"sofia_kim_7287\",
  \"origin\": \"IAH\",
  \"destination\": \"DEN\",
  \"flight_type\": \"round_trip\",
  \"cabin\": \"economy\",
  \"flights\": [
    {
      \"flight_number\": \"HAT078\",
      \"date\": \"2024-05-27\",
      \"price\": 146,
      \"origin\": \"IAH\",
      \"destination\": \"ORD\"
    },
    {
      \"flight_number\": \"HAT118\",
      \"date\": \"2024-05-27\",
```

```
        \"price\": 167,
        \"origin\": \"ORD\",
        \"destination\": \"DEN\"
      },
      {
        \"flight_number\": \"HAT290\",
        \"date\": \"2024-05-27\",
        \"price\": 142,
        \"origin\": \"DEN\",
        \"destination\": \"LAS\"
      },
      {
        \"flight_number\": \"HAT175\",
        \"date\": \"2024-05-27\",
        \"price\": 170,
        \"origin\": \"LAS\",
        \"destination\": \"IAH\"
      }
    ],
    \"passengers\": [
      {
        \"first_name\": \"Sofia\",
        \"last_name\": \"Kim\",
        \"dob\": \"1950-06-24\"
      }
    ],
    \"payment_history\": [
      {
        \"payment_id\": \"gift_card_7091239\",
        \"amount\": 566
      },
      {
        \"payment_id\": \"gift_card_6276644\",
        \"amount\": 59
      }
    ],
    \"created_at\": \"2024-05-07T16:32:35\",
    \"total_baggages\": 1,
    \"nonfree_baggages\": 0,
    \"insurance\": \"no\"
}"
```

**Agent:** Great! Your flight changes have been successfully updated. Now let me add the additional checked bag.

**By now, the agent has successfully updated the flights and everything is perfect, however it made an error in the last step when updating the checked bags.**

**Agent Tool Call:**

```
"function": {
  "arguments": "{\
    \"reservation_id\": \"OBUT9V\", \
    \"total_baggages\": 2, \
    \"nonfree_baggages\": 1, \
    \"payment_id\": \"gift_card_6276644\"\
  }",
  "name": "update_reservation_baggages"
},
```

> **The agent added one non-free baggage to the reservation because the user requested an additional checked bag. However, according to the highlighted part of policy and the user's history, the user is a Silver member traveling in economy class and is entitled to two free checked bags. By charging the user an extra checked bag fee, the agent made an error.**

## K    FULL LIST OF RELATED WORK

### K.1    PROMPT COMPRESSION FOR LARGE LANGUAGE MODELS

Prompt compression (Li et al., 2024) aims to obtain a more compact representation of lengthy inputs while preserving the original outputs. Early approaches include hard prompting (Chuang et al., 2024; Jiang et al., 2023; Li et al., 2023), which prune tokens that contribute little to the response while retaining natural language or subword tokens, and soft prompting (Mu et al., 2024; Ge et al., 2023; Chevalier et al., 2023), which replace the original prompt with learnable embeddings with the help of trainable encoder-decoder architecture. While soft prompts often rely on non natural language embeddings, they generally provide stronger generalization for handling diverse requirements. Our special token–based internalization (e.g., policy identifiers) combines the strengths of both: it is interpretable and thus easier for real-world business management, while still supporting flexible learning to enable generalization. PromptIntern (Zou et al., 2024) introduces a pipeline for progressively internalizing input tokens, but it does not explicitly address the unique reasoning challenges posed by the complex structure of policy documents.

### K.2    DELIBERATE ALIGNMENT

Deliberative alignment proposes internalizing general safety rules and behaviors into a model's prior, reducing the need to specify them in-context via additional training (Guan et al., 2024) or test-time deliberation (Zhang et al., 2025a). While related to our setting, this line of work is restricted to general safety behaviors, overlooks the broader scope of agentic policies, and does not address complex reasoning challenges central to policy internalization (e.g., workflow-level constraints).

### K.3    CONTINUED PRETRAINING FOR LARGE LANGUAGE MODELS

Continued Pretraining (CPT) has become a critical paradigm for keeping large language models (LLMs) up-to-date with evolving data distributions while mitigating catastrophic forgetting. Positioned at the top layer of the modern continual learning pipeline, CPT incrementally trains LLMs on newly collected unlabeled corpora to retain general knowledge, acquire novel information, and revise outdated facts, offering a more efficient alternative to full retraining (Shi et al., 2025). Existing approaches largely build on classical continual learning methods, such as replay-based rehearsal of exemplars or pseudo-samples, parameter regularization techniques like Elastic Weight Consolidation (EWC) (Kirkpatrick et al., 2017) and RecAdam (Chen et al., 2020) to constrain parameter drift, and architecture-based strategies such as adapter modules, vocabulary expansion, and sparse modular structures (e.g. Mixture-of-Experts) that help isolate new knowledge without overwriting old representations (Shi et al., 2025; Zhou et al., 2024). In particular, modular expert-based designs like DEMix layers (Gururangan et al., 2022) support mixing, adding, or removing domain-specific experts to facilitate adaptation and reduce forgetting, and Lifelong-MoE (Chen et al., 2023) dynamically expands expert capacity during CPT to absorb new distributions while preserving prior knowledge. Empirical results suggest CPT methods consistently improve downstream generalization under gradual or correlated distribution shifts, though naive sequential updates can provoke significant forgetting in temporally shifting domains (Shi et al., 2025). Replay-based methods may be less effective in CPT due to overfitting risks, while parameter-efficient finetuning (LoRA, adapters) and modular expansion techniques show stronger robustness to both temporal and content shifts, making them attractive for scalable production pipelines (Zhou et al., 2024). Despite progress, current surveys stress that CPT research is still in early stages: technique diversity remains limited, long-horizon simulations are rare, and standardized evaluation benchmarks for vertical forgetting are lacking, pointing to important directions for future work (Shi et al., 2025). In our approach, we primarily rely on continued pretraining (CPT) to enable more generalizable learning and mitigate

the catastrophic forgetting often observed in pure SFT methods, while incorporating targeted data and policy-grounded question–answer pairs to better facilitate downstream adaptation.

### K.4 Knowledge Injection for Large Language Models

Knowledge injection techniques aim to enhance the domain expertise of large language models (LLMs) by integrating external or structured knowledge into their training or inference process, thereby bridging the gap between general-purpose reasoning and specialized applications (Song et al., 2025b). Existing methods are broadly categorized into four paradigms: dynamic knowledge injection, which retrieves knowledge at inference time and augments the input context—often using retrieval-augmented generation (RAG) with semantic search or knowledge graphs (Zhang et al., 2024); static knowledge embedding, which encodes domain information into model parameters via continued pretraining or fine-tuning, enabling faster inference but risking catastrophic forgetting when knowledge evolves; modular adapters, which introduce trainable modules such as K-Adapters to store domain knowledge while keeping backbone parameters frozen, providing parameter-efficient updates and preserving general capabilities (Wang et al., 2021; He et al., 2021); and prompt optimization, which relies on carefully designed or learned prompts to guide the model without parameter updates (Peng et al., 2025; Liu et al., 2024b). Recent work demonstrates that hybrid approaches, such as combining retrieval with prompt optimization or adapters (e.g., KnowGPT and StructTuning), yield strong performance by balancing flexibility, scalability, and computational efficiency (Liu et al., 2024b; Zhang et al., 2024). Empirical comparisons in biomedical and financial domains show that static embedding often achieves the highest task-specific accuracy, while dynamic injection provides superior adaptability and up-to-date knowledge coverage, highlighting the importance of choosing injection strategies based on application requirements (Song et al., 2025b). In our work, the internalization of policy documents is related to, but distinct from, knowledge injection. Our task emphasizes deep understanding and practical application of policy rules rather than mere memorization, which also requires extensive reasoning. To address these unique challenges, we characterize the specific complexities of policy interpretation and propose a CPT-based approach tailored to this setting. Among the aforementioned approaches, ours bears the closest resemblance to prompt optimization.

## L   Ethical Statement on LLM Assistance

In addition to the reported uses of large language models (LLMs) for running experiments, we primarily use ChatGPT-5 as a tool for language refinement, including polishing text and improving clarity. All model-generated content is thoroughly reviewed and rewritten by human authors to ensure accuracy, originality, and adherence to research integrity standards.

## M   Limitation and Future Work

In this section, we discuss the limitations of our work and outline future directions.

(1) **Scope of the benchmark.** Our study uses a text-only, single-turn agent setting (Section§ 2.1); consequently, our complexity characterization primarily reflects the policy-document dimension and its associated agentic tasks. In practice, complexity also arises from intricate user intents, multi-turn planning and repair, and multimodal inputs (e.g., screenshots, receipts, instructional videos). Extending CC-Gen and the evaluation suite to multi-turn and multimodal settings, while explicitly modeling a distribution over user intents is an important next step.

(2) **Training recipe.** Our approach emphasizes category-aware policy structure and applies continued pretraining (CPT) followed by SFT, underscoring that explicit complexity characterization is indispensable. We did not incorporate reinforcement-learning stages (e.g., GRPO/PPO-style objectives) that could leverage our trajectories. Adding an RL fine-tuning stage on top of CAP-CPT+SFT for improved alignment is a promising extension.

(3) **Challenging task variants.** Despite strong average gains, models remain brittle on policy-substitute, policy-override, and policy-referral. These practical extensions of the core internalization task helps to extend the robustness and safety of the overall system. Simply scaling training data

may lift scores on a fixed evaluation set but yields limited gains more broadly because override granularity (what to override, scope, validity window) and referral formats are under-specified. Future work includes targeted data generation with controllable override or referral schemas, counterfactual training, and evaluation protocols that explicitly balance base performance, adaptation fidelity, and robustness. While context engineering approaches for safe and reliable output (Wang et al., 2025) are also under consideration.

(4) **Fragility of strong priors.** We find that stronger reasoning models can be more prone to policy-specific interference and forgetting. Although CAP-CPT with self-generated CoT mitigates this (Appendix F), we lack guarantees against negative transfer or regressions in general instruction following. Future work should investigate selective internalization via policy identifiers, prior-preservation regularizers, and continual-learning safeguards for safe deployment.

**Future Work** An important direction is to integrate retrieval-augmented generation (RAG) with our CAP-CPT framework to achieve more fine-grained and context-aware internalization, allowing models to dynamically ground their policy reasoning in high-precision retrieved evidence. Another promising avenue is to leverage reinforcement learning to further refine the internalization process, enabling models to explore policy-consistent behaviors and optimize long-horizon adherence rewards. We also plan to study methods for mitigating forgetting during continual policy updates, ensuring that newly internalized rules do not overwrite previously aligned behaviors. Beyond the current policy set, we aim to generalize to unseen policy documents by explicitly encoding overriding relationships between policy sources. Finally, we will explore parallel policy internalization, enabling models to internalize multiple, potentially interacting policies simultaneously and resolve conflicts through structured reasoning.

# N    POLICY DOCUMENTS CLOSER TO THE REAL WORLD

In this section, we extend our policy documents to a group of new policies which is written in a completely different way compared to those policy documents generated by our {CC-Gen} benchmark. These policies are human written and polished by Large Language Models like GPT-5. We used them and coresponding QAs for a more in-depth evaluation of our approach in real-world scenarios. We show the complete content of one of these policy documents as below:

---

**Human written real-world Policy Evaluation**

### # Airline Agent Policy

The current time is 2025-10-31 23:42:08 EST.

As an airline agent, you can help users book, modify, cancel, or query flight reservations.

- Before taking any actions that update the booking database (booking, modifying flights, editing baggage, upgrading cabin class, or updating passenger information), you must list the action details and obtain explicit user confirmation (yes) to proceed.

- You should not provide any information, knowledge, or procedures not provided by the user or available tools, or give subjective recommendations or comments.

- You should only make one tool call at a time, and if you make a tool call, you should not respond to the user simultaneously. If you respond to the user, you should not make a tool call at the same time.

- You should deny user requests that are against this policy.

- You should transfer the user to a human agent if and only if the request cannot be handled within the scope of your actions.

---

## Domain Basic

### Profile Structure

**Passengers Profile**
The agent should use the {Username} and {PaxID} as conditions to access and locate the profile existence of the specific passenger. This profile also includes {membership_level}, {payment_methods}, {saved_passengers}, and {reservations} of the passenger. Among these attributes, {reservations} will be the primary key to Profile {Reservations}. The attribute {email} can be used as an alternative way to access the profiles {Passenger} while searching.

**Reservations Profile** Each reservation contains {reservation_id}, {user_id}, {origin}, {destination}, {flight_type} (one_way, round_trip), {flights}, {passengers}, {payment_history}, {created_at}, {total_baggages}, and {nonfree_baggages}. Among these attributes, {user_id} links back to the Passengers profile, and {flights} contains {flight_number} which links to the Flights profile.

**Flights Profile** Each flight has {flight_number}, {origin}, {destination}, {scheduled_departure_time_est}, {scheduled_arrival_time_est}, and for each {date}:
- If the status is "available", the flight has not taken off, {available_seats} and {prices} for each cabin are listed. - If the status is "delayed", the flight has not taken off but is delayed, with {new_departure_time_est} and {delay_reason}. - If the status is "landed", the flight has already landed with {actual_departure_time_est} and {actual_arrival_time_est}. - If the status is "cancelled", the flight has been cancelled with {cancellation_reason}.

### Membership Tiers and Benefits

- **Basic**: 0% discount, limited baggage allowance
- **Silver**: 10% discount on flights, enhanced baggage allowance
- **Gold**: 15% discount on flights, premium baggage allowance
- **Platinum**: 20% discount on flights, premium baggage allowance

### Cabin Types

- **basic_economy**: Lowest price, most restrictions
- **economy**: Standard service
- **business**: Premium service, most flexibility

## Task 1: BookFlight

### Step 1: User Identity Verification

**Input Parameters**: {user_id} for booking user and {companions} list with each companion's {user_id}

**Process**: The agent verifies each user exists in the Passengers profile by looking up their {user_id}. For each user, retrieve their {membership} level (platinum/gold/silver/basic) and {payment_methods}.

**Output**: {membership} for each passenger, {payment_methods} for booking user

**Policy Rules**: All user IDs must exist in the system. If any user ID is invalid, terminate the booking process.

### Step 2: Flight Information Verification

**Input Parameters**: {origin}, {destination}, {departure_time}, {arrive_time}, {cabin} for each passenger

**Process**: Search the Flights profile for flights matching the route and time requirements. The flight must have status "available" and sufficient {available_seats} in the requested {cabin} type for all passengers.

**Output**: {flight_number}, {prices} dictionary containing prices for basic_economy, economy, and business cabins.

**Policy Rules**:
- Only flights with status "available" can be booked
- Flight departure time must be after {departure_time}
- Flight arrival time must be before {arrive_time}
- Available seats in requested cabin must be ¿= number of passengers requesting that cabin

### Step 3: Calculate Total Airfare Cost

**Input Parameters**: {cabin} for each passenger, {prices} from flight, {membership} for each passenger

**Process**: For each passenger, calculate their ticket cost as: cabin_price × (1 - membership_discount). Sum all passenger costs to get total airfare.

**Output**: {airfare_cost} (total for all passengers)

**Policy Rules - Membership Discounts**:

- **Platinum members:** 20% discount
- **Gold members:** 15% discount
- **Silver members:** 10% discount
- **Basic members:** 0% discount

### Step 4: Calculate Checked Baggage Cost

**Input Parameters**: {luggage_number} (total bags for all passengers), {membership} of booking user, {cabin} of booking user

**Process**: Determine free baggage allowance based on booking user's membership and cabin. Calculate excess baggage count and multiply by $50 per bag.

**Output**: {baggage_cost}, {free_allowance}, {excess_bags}

**Policy Rules - Free Baggage Allowance**:

- **Basic membership:** 0 free bags (basic_economy), 1 bag(s) (economy), 2 bags (business)
- **Silver membership:** 1 free bag(s) (basic_economy), 2 bags (economy), 3 bags (business)
- **Gold membership:** 2 free bags (basic_economy), 3 bags (economy), 3 bags (business)
- **Platinum membership:** 2 free bags (basic_economy), 3 bags (economy), 3 bags (business)
- **Excess baggage fee:** $50 per bag beyond free allowance

### Step 5: Calculate Travel Insurance Cost

**Input Parameters**: {insurance} status (yes/no) for each passenger

**Process**: Count passengers who selected insurance and multiply by $30 per passenger.
**Output**: {insurance_cost}

**Policy Rules**:
- Insurance price: $30 per passenger
- Insurance coverage: Enables full refund if cancellation is due to health or weather reasons
- Insurance cannot be added after booking is completed

### Step 6: Payment Method Verification

**Input Parameters**: {payment_methods} selected by user, {final_cost} (sum of airfare + baggage + insurance)
**Process**: Verify all selected payment methods exist in the booking user's profile. Check that payment method combination follows policy limits.

**Output**: Validation result (pass/fail)

**Policy Rules - Payment Method Limits**:

- Maximum 1 travel certificate per reservation - Maximum 1 credit card per reservation - Maximum 3 gift cards per reservation - All payment methods must already exist in user's profile for security - Travel certificate remaining balance is non-refundable

### Step 7: Confirmation and Execution

**Input Parameters**: All booking details including {user_ids}, {flight_number}, {passengers}, {final_cost}, {payment_methods}

**Process**: Display complete booking summary including flight details, passenger list, price breakdown (airfare, baggage, insurance), and total cost. Request explicit user confirmation.

**Output**: {reservation_id} if confirmed, or cancellation if user declines

**Policy Rules**:
- Must display all booking details before confirmation - Require explicit "yes" confirmation from user - Only create reservation after receiving confirmation

## Task 2: ModifyFlight

### Step 1: User Identity Verification

**Input Parameters**: {user_id}, {reservation_id}

**Process**: Verify the user exists in Passengers profile and retrieve their {membership}, {payment_methods}, and {reservations} list. Confirm that {reservation_id} exists in the user's {reservations} list.

**Output**: {membership}, {payment_methods}, validation result

**Policy Rules**: - User must exist in the system - Reservation must belong to the user (reservation_id must be in user's reservations list) - If validation fails, terminate modification process

### Step 2: Reservation Status Check

**Input Parameters**: {reservation_id}

**Process**: Retrieve complete reservation details from Reservations profile including {flight_type}, {flights}, {passengers}, {total_baggages}, {nonfree_baggages},

{payment_history}, {created_at}.

**Output**: All reservation details, {hours_until_departure}

**Policy Rules**: - Flight date must be in the future (not expired) - Reservation status must be "confirmed" (not already used) - Time until departure must be ¿ 2 hours (applies to all cabin types) - If any rule violated, cannot modify reservation

### Step 3: Modification Eligibility Check

**Input Parameters**: {passengers} list with each passenger's {cabin}, {flights} list

**Process**: Check if any passenger has basic_economy cabin. Calculate modification fees based on cabin types.

**Output**: {can_modify} (boolean), {total_modification_fee}

**Policy Rules**: - If ANY passenger has basic_economy cabin → Cannot modify flights (terminate process) - Economy passengers: $75 modification fee per person - Business passengers: $0 modification fee (free) - Cannot change {origin} or {destination} (must cancel and rebook instead) - Can only change {departure_time} and {arrive_time} (same route, different flight)

### Step 4: New Flight Verification

**Input Parameters**: {origin}, {destination}, {departure_time}, {arrive_time}, {passengers} with cabin requirements

**Process**: Search Flights profile for available flights matching criteria. Verify sufficient seats in each required cabin type.

**Output**: {new_flights} list with {flight_number}, {prices}, {available_seats}

**Policy Rules**: - New flights must have status "available" - Must have enough available seats in each cabin for all passengers - Origin and destination must match original reservation

### Step 5: Price Difference Calculation

**Input Parameters**: {passengers} with {membership}, {flights} (original), {new_flights}, {total_modification_fee}

**Process**: For each passenger, calculate original flight cost and new flight cost with membership discounts applied. Sum the differences and add modification fees.

**Output**: {price_difference} (positive = payment needed, negative = refund), {total_new_cost}, {total_original_cost}

**Policy Rules**: - Apply membership discounts to both original and new flight costs - Membership discounts: Platinum 20- Final price difference = (new_cost - original_cost) + modification_fees

### Step 6: Payment Verification

**Input Parameters**: {payment_methods}, {price_difference}

**Process**: If price_difference ¿ 0, verify payment method available. If price_difference ¡ 0, process refund to original payment method. Display modification summary and request

confirmation.

**Output**: Payment validation result, {reservation_id} (updated)

**Policy Rules**: - If payment required: Must use valid payment method from user profile - If refund: Return to original payment method - Must show complete modification details before confirmation - Require explicit "yes" confirmation

### Step 7: Confirmation and Execution

**Input Parameters**: {reservation_id}, {new_flights}, {price_difference}, {payment_methods}

**Process**: Display complete modification summary and execute the flight change after user confirmation.

**Output**: Updated {reservation_id}

**Policy Rules**: - Must display all modification details before confirmation - Require explicit "yes" confirmation from user - Only update reservation after receiving confirmation

## Task 3: ModifyPassengers

### Step 1: User Identity Verification

**Input Parameters**: {user_id}, {reservation_id}

**Process**: Verify the user exists in Passengers profile and retrieve their {membership}, {payment_methods}, and {reservations} list. Confirm that {reservation_id} exists in the user's {reservations} list.

**Output**: {membership}, {payment_methods}, validation result

**Policy Rules**: - User must exist in the system - Reservation must belong to the user - If validation fails, terminate modification process

### Step 2: Reservation Status Check

**Input Parameters**: {reservation_id}

**Process**: Retrieve complete reservation details from Reservations profile including {flight_type}, {flights}, {passengers}, {total_baggages}, {nonfree_baggages}, {payment_history}, {created_at}.

**Output**: All reservation details, {hours_until_departure}

**Policy Rules**: - Flight date must be in the future (not expired) - Reservation status must be "confirmed" (not already used) - Time until departure must be ¿ 2 hours - If any rule violated, cannot modify reservation

### Step 3: Passenger Limit Check

**Input Parameters**: {current_passenger_count}, {add_passengers} list, {remove_passengers} list, {user_id} (booking user)

**Process**: Calculate new passenger count after additions/removals. Verify limits and restrictions.

**Output**: {new_passenger_count}, validation result

**Policy Rules**: - Maximum 5 passengers per reservation - Minimum 1 passenger (cannot remove all passengers) - Cannot remove booking user (user_id of reservation owner) - new_passenger_count = current_count + add_count - remove_count - Must satisfy: $1 \leq new\_passenger\_count \leq 5$

### Step 4: Remove Passenger Processing

**Input Parameters**: {remove_passengers} list with {user_id} for each, {passengers} current list, {flights}

**Process**: For each passenger being removed, calculate their flight cost with membership discount applied. Calculate refund as 80% of flight cost. If passenger had insurance, refund full $30 insurance cost.

**Output**: {refund_amount}, {removed_passenger_details}

**Policy Rules**: - Refund policy: 80% of flight cost per removed passenger - Insurance refund: 100% ($30) if passenger had insurance - Baggage fees: Non-refundable (no refund for baggage) - Apply membership discounts when calculating original flight cost

### Step 5: Add Passenger Processing

**Input Parameters**: {add_passengers} list with {user_id}, {cabin}, {insurance}, {luggage_number} for each, {flights}

**Process**: For each new passenger, verify user_id exists, check flight seat availability in requested cabin, calculate costs (flight + insurance + baggage with membership discounts and free allowances).

**Output**: {additional_cost}, {added_passenger_details}

**Policy Rules**: - All new passenger user_ids must exist in Passengers profile - Flights must have available seats in requested cabins - Calculate flight cost with membership discount - Insurance: $30 per passenger if requested - Baggage: Apply free allowance based on passenger's membership and cabin, charge $50 per excess bag

### Step 6: Total Cost Calculation

**Input Parameters**: {refund_amount}, {additional_cost}

**Process**: Calculate net cost (additional_cost - refund_amount).

**Output**: {total_price_difference}

**Policy Rules**: - Net cost = additional_cost - refund_amount - If net cost ¿ 0: Payment required - If net cost ¡ 0: Refund to original payment method

### Step 7: Payment Verification

**Input Parameters**: {payment_methods}, {total_price_difference}

**Process**: If positive cost, verify payment. If negative, process refund. Display passenger change summary and request confirmation.

**Output**: Payment validation result

**Policy Rules**: - Must display all passenger changes and costs before confirmation - Require explicit "yes" confirmation

### Step 8: Confirmation and Execution

**Input Parameters**: {reservation_id}, {add_passengers}, {remove_passengers}, {total_price_difference}

**Process**: Display complete passenger modification summary and execute changes after user confirmation.

**Output**: Updated {reservation_id}

**Policy Rules**: - Must display complete modification details - Require explicit "yes" confirmation from user - Only update reservation after receiving confirmation

## Task 4: ModifyCabin

### Step 1: User Identity Verification

**Input Parameters**: {user_id}, {reservation_id}

**Process**: Verify the user exists in Passengers profile and retrieve their {membership}, {payment_methods}, and {reservations} list. Confirm that {reservation_id} exists in the user's {reservations} list.

**Output**: {membership}, {payment_methods}, validation result

**Policy Rules**: - User must exist in the system - Reservation must belong to the user - If validation fails, terminate modification process

### Step 2: Reservation Status Check

**Input Parameters**: {reservation_id}

**Process**: Retrieve complete reservation details from Reservations profile including {flight_type}, {flights}, {passengers}, {total_baggages}, {payment_history}, {created_at}.

**Output**: All reservation details, {hours_until_departure}

**Policy Rules**: - Flight date must be in the future (not expired) - Reservation status must be "confirmed" (not already used) - Time until departure must be ¿ 2 hours - If any rule violated, cannot modify reservation

### Step 3: Cabin Change Eligibility Check

**Input Parameters**: {passenger_user_id}, {original_cabin}, {new_cabin}, {flights}

**Process**: Verify cabin change is allowed (upgrade or downgrade with restrictions). Check all flights have available seats in new cabin.

**Output**: {can_change} (boolean), {change_type} (upgrade/downgrade)

**Policy Rules - Allowed Changes**: - Upgrades allowed: basic_economy → economy/business, economy → business - Downgrades allowed with partial refund: - business → economy or basic_economy: Refund 80% of price difference - economy → basic_economy: Refund 50% of price difference - Must change all flight segments together (cannot change only one flight in round trip) - All flights must have available seats in new cabin

### Step 4: Cabin Price Difference Calculation

**Input Parameters**: {passenger_user_id}, {original_cabin}, {new_cabin}, {flights}, {current_luggage_number}

**Process**: Calculate price difference between cabins with membership discount. Recalculate baggage allowance and fees based on new cabin.

**Output**: {cabin_price_difference}, {luggage_cost_difference}, {new_free_allowance}

**Policy Rules**: - Apply passenger's membership discount to both original and new cabin prices - For upgrades: price_difference = new_cabin_cost - original_cabin_cost - For downgrades: price_difference = -(original_cabin_cost - new_cabin_cost) × refund_percentage - Recalculate free baggage allowance based on new cabin and membership - Recalculate excess baggage fees: (luggage_number - new_free_allowance) × $50

### Step 5: Baggage Allowance Adjustment

**Input Parameters**: {passenger_user_id}, {membership}, {new_cabin}, {current_luggage_number}

**Process**: Calculate new free baggage allowance and any additional fees or refunds.

**Output**: {new_free_allowance}, {baggage_adjustment}

**Policy Rules**: - New free allowance based on membership and new cabin - If new allowance ¿ old allowance: May receive partial refund for previously paid bags - If new allowance ¡ old allowance: May owe additional fees

### Step 6: Payment Verification

**Input Parameters**: {payment_methods}, {cabin_price_difference}, {luggage_cost_difference}

**Process**: Calculate total cost (cabin_price_difference + luggage_cost_difference). Process payment or refund. Display cabin change summary and request confirmation.

**Output**: {total_cost}, {reservation_id} (updated)

**Policy Rules**: - Total cost = cabin_price_difference + luggage_cost_difference - Must display cabin change details, price breakdown, and baggage allowance changes - Require explicit "yes" confirmation

### Step 7: Confirmation and Execution

**Input Parameters**: {reservation_id}, {passenger_user_id}, {new_cabin}, {total_cost}

**Process**: Display complete cabin modification summary and execute changes after user confirmation.

**Output**: Updated {reservation_id}

**Policy Rules**: - Must display all modification details before confirmation - Require explicit "yes" confirmation from user - Only update reservation after receiving confirmation

## Task 5: ModifyInsurance

### Step 1: User Identity Verification

**Input Parameters**: {user_id}, {reservation_id}

**Process**: Verify the user exists in Passengers profile and retrieve their {membership}, {payment_methods}, and {reservations} list. Confirm that {reservation_id} exists in the user's {reservations} list.

**Output**: {membership}, {payment_methods}, validation result

**Policy Rules**: - User must exist in the system - Reservation must belong to the user - If validation fails, terminate modification process

### Step 2: Reservation Status Check

**Input Parameters**: {reservation_id}

**Process**: Retrieve complete reservation details from Reservations profile including {flight_type}, {flights}, {passengers}, {payment_history}, {created_at}.

**Output**: All reservation details, {hours_until_departure}

**Policy Rules**: - Flight date must be in the future (not expired) - Reservation status must be "confirmed" (not already used) - Time until departure must be ¿ 2 hours - If any rule violated, cannot modify reservation

### Step 3: Insurance Modification Validation

**Input Parameters**: {passenger_user_id}, {original_insurance}, {new_insurance}

**Process**: Verify insurance modification is allowed (only additions permitted). Calculate insurance cost if adding.

**Output**: {insurance_cost}, validation result

**Policy Rules**: - Can ONLY ADD insurance (no → yes transition allowed) - CANNOT REMOVE insurance (yes → no transition forbidden) - If already has insurance (yes → yes): No change needed, return error - Insurance price: $30 per passenger - Insurance coverage: Full refund for health/weather cancellation reasons

### Step 4: Insurance Cost Calculation

**Input Parameters**: {passenger_user_id}, {new_insurance}

**Process**: Calculate insurance cost for the passenger.

**Output**: {insurance_cost}

**Policy Rules**: - Insurance cost: $30 per passenger - Must process payment before adding insurance

### Step 5: Payment Verification

**Input Parameters**: {payment_methods}, {insurance_cost}

**Process**: Verify payment method and process payment for insurance. Display insurance addition summary and request confirmation.

**Output**: Payment validation result

**Policy Rules**: - Must use valid payment method from user profile - Must display insurance details and cost before confirmation - Require explicit "yes" confirmation

### Step 6: Confirmation and Execution

**Input Parameters**: {reservation_id}, {passenger_user_id}, {insurance_cost}

**Process**: Display complete insurance modification summary and execute changes after user confirmation.

**Output**: Updated {reservation_id}

**Policy Rules**: - Must display all modification details before confirmation - Require explicit "yes" confirmation from user - Only update reservation after receiving confirmation - Insurance cannot be removed once added

## Task 6: ModifyLuggage

### Step 1: User Identity Verification

**Input Parameters**: {user_id}, {reservation_id}

**Process**: Verify the user exists in Passengers profile and retrieve their {membership}, {payment_methods}, and {reservations} list. Confirm that {reservation_id} exists in the user's {reservations} list.

**Output**: {membership}, {payment_methods}, validation result

**Policy Rules**: - User must exist in the system - Reservation must belong to the user - If validation fails, terminate modification process

### Step 2: Reservation Status Check

**Input Parameters**: {reservation_id}

**Process**: Retrieve complete reservation details from Reservations profile including {flight_type}, {flights}, {passengers}, {total_baggages}, {nonfree_baggages}, {payment_history}, {created_at}.

**Output**: All reservation details, {hours_until_departure}

**Policy Rules**: - Flight date must be in the future (not expired) - Reservation status must be "confirmed" (not already used) - Time until departure must be ¿ 2 hours - If any

rule violated, cannot modify reservation

### Step 3: Luggage Modification Validation

**Input Parameters**: {passenger_user_id}, {original_luggage_number}, {new_luggage_number}, {cabin}, {total_baggages}

**Process**: Verify baggage modification is allowed (only additions). Check limits.

**Output**: {can_modify} (boolean), validation result

**Policy Rules**: - Can ONLY ADD baggage (new_luggage_number must be ¿ original_luggage_number) - CANNOT REMOVE baggage (already added baggage is non-refundable) - Maximum 8 bags per passenger - Maximum 8 bags per reservation total - If limits exceeded, cannot modify

### Step 4: Additional Luggage Cost Calculation

**Input Parameters**: {passenger_user_id}, {membership}, {cabin}, {original_luggage_number}, {new_luggage_number}

**Process**: Calculate free allowance based on passenger's membership and cabin. Calculate additional baggage cost.

**Output**: {additional_luggage_cost}, {new_total_baggages}

**Policy Rules**: - Calculate free allowance based on membership and cabin - Additional cost = (new_paid_bags - original_paid_bags) × $50 - new_paid_bags = max(0, new_luggage_number - free_allowance) - original_paid_bags = max(0, original_luggage_number - free_allowance)

### Step 5: Payment Verification

**Input Parameters**: {payment_methods}, {additional_luggage_cost}

**Process**: Verify payment method and process payment for additional luggage. Display luggage addition summary and request confirmation.

**Output**: Payment validation result

**Policy Rules**: - Must use valid payment method from user profile - Must display luggage details and cost before confirmation - Require explicit "yes" confirmation

### Step 6: Confirmation and Execution

**Input Parameters**: {reservation_id}, {passenger_user_id}, {new_luggage_number}, {additional_luggage_cost}

**Process**: Display complete luggage modification summary and execute changes after user confirmation.

**Output**: Updated {reservation_id}

**Policy Rules**: - Must display all modification details before confirmation - Require explicit "yes" confirmation from user - Only update reservation after receiving confirmation - Luggage cannot be removed once added

## Task 7: CancelFlight

### Step 1: User Identity Verification

**Input Parameters**: {user_id}, {reservation_id}

**Process**: Verify user exists in Passengers profile and retrieve {membership}, {payment_methods}, and {reservations} list. Confirm {reservation_id} is in user's reservations.

**Output**: {membership}, {payment_methods}, validation result

**Policy Rules**: - User must exist in the system - Reservation must belong to the user - If validation fails, terminate cancellation process

### Step 2: Reservation Status Check

**Input Parameters**: {reservation_id}

**Process**: Retrieve complete reservation details including {flight_type}, {flights}, {passengers}, {total_baggages}, {payment_history}, {created_at}.

**Output**: All reservation details, {hours_until_departure}

**Policy Rules**: - Flight date must be in the future (not expired) - Reservation status must be "confirmed" (not already used) - If flight already departed or used, cannot cancel

### Step 3: Cancellation Eligibility Check

**Input Parameters**: {passengers} with {cabin} for each, {flights}, {created_at}, {hours_until_departure}

**Process**: Check each passenger's cabin type to determine cancellation eligibility and fees. Calculate time-based restrictions and additional fees.

**Output**: {can_cancel} (boolean), {total_cancellation_fee}, {has_basic_economy}

**Policy Rules - Cancellation Fees by Cabin**: - basic_economy: Non-refundable ($0 refund) - economy: Refundable with $200 cancellation fee per passenger - business: Refundable with $75 cancellation fee per passenger

**Policy Rules - Time Restrictions**: - Less than 2 hours until departure: Cannot cancel (terminate process) - 2-24 hours until departure: Additional $50 emergency cancellation fee (applied once per reservation) - More than 24 hours until departure: Normal cancellation fees only

**Policy Rules - Warnings**: - If any passenger has basic_economy: Display warning "Reservation includes Basic Economy cabin, that portion is non-refundable"

### Step 4: Refund Amount Calculation

**Input Parameters**: {passengers} with {cabin}, {insurance}, {membership} for each, {flights}, {cancel_reason}, {total_cancellation_fee}, {hours_until_departure}

**Process**: For each passenger, calculate their original flight cost with membership discount. Determine refund based on cabin type and cancellation reason. Add insurance

refund if applicable. Subtract emergency fee if within 24 hours.

**Output**: {total_refund}, {refund_breakdown} (per passenger details)

**Policy Rules - Refund Calculation**: - basic_economy passengers: $0 refund (non-refundable) - Special cancellation reasons (weather/health): Full refund of flight cost (no cancellation fee) - Normal cancellation (personal/schedule_change): - economy: flight_cost - $200 cancellation fee - business: flight_cost - $75 cancellation fee - Insurance refund: - If cancel_reason is weather or health: Refund $30 per passenger with insurance - Otherwise: $0 insurance refund - Emergency cancellation fee: Subtract $50 if hours_until_departure ¡ 24 - Baggage fees: Non-refundable (no refund for any baggage fees paid) - Apply membership discounts when calculating original flight costs

### Step 5: Refund Method Confirmation

**Input Parameters**: {payment_history}, {total_refund}

**Process**: Determine refund destination based on original payment method used.

**Output**: {refund_method}, {refund_destination}

**Policy Rules - Refund Methods**: - If paid with credit card: Refund to the same credit card - If paid with travel certificate: Issue new travel certificate for refund amount - If paid with gift card: Refund to new gift card - Refund processing time: 5-7 business days - Refund goes to original payment method(s) in reverse order of use

### Step 6: Confirmation and Execution

**Input Parameters**: {reservation_id}, {total_refund}, {cancel_reason}, {payment_history}, {refund_breakdown}

**Process**: Display complete cancellation summary including flight details, each passenger's refund breakdown, cancellation fees, total refund amount, and refund method. Request explicit user confirmation.

**Output**: Cancellation confirmation, {reservation_id} (cancelled status)

**Policy Rules**: - Must display complete cancellation details before confirmation - Show per-passenger refund breakdown - Show all fees (cancellation fees, emergency fees) - Show refund method and expected processing time - Require explicit "yes" confirmation from user - Only process cancellation after receiving confirmation

## Task 8: QueryFlight

### Query Type 1: Search Available Flights

**Step 1: Parameter Validation**

**Input Parameters**: {origin}, {destination}, {departure_date}, {return_date} (optional), {cabin} (optional), {passenger_count} (optional, default 1)

**Process**: Validate all input parameters to ensure they meet requirements for flight search.

**Output**: Validation result (pass/fail)

**Policy Rules**: - {origin} and {destination} must be valid 3-letter airport codes - {origin} cannot equal {destination} - {departure_date} must be a future date (not in the past) - If {return_date} provided, it must be later than {departure_date} - {passenger_count} must be positive integer (1-5)

**Step 2: Search Outbound Flights**

**Input Parameters**: {origin}, {destination}, {departure_date}, {cabin} (optional), {passenger_count}

**Process**: Query Flights profile for all flights matching the route on the specified date. Filter for flights with status "available" and sufficient seats. Sort results by departure time.

**Output**: {outbound_flights} list with {flight_number}, {departure_time}, {arrival_time}, {available_seats}, {prices} for each flight

**Policy Rules**: - Only include flights with status "available" - If {cabin} specified: Only include flights with available_seats[cabin] ¿= passenger_count - If {cabin} not specified: Include all flights with any cabin having sufficient seats - Sort flights by {expected_departure_time_est} in ascending order

**Step 3: Search Return Flights (if round trip)**

**Input Parameters**: {destination}, {origin}, {return_date}, {cabin} (optional), {passenger_count}

**Process**: If {return_date} provided, search for return flights using same logic as outbound search, but with origin and destination swapped.

**Output**: {return_flights} list (same structure as outbound_flights)

**Policy Rules**: Same as Step 2, applied to return route

**Step 4: Format and Return Results**

**Input Parameters**: {outbound_flights}, {return_flights} (if applicable)

**Process**: Format search results into readable display showing all flight options with details.

**Output**: Formatted flight search results

**Policy Rules - Display Format**: - Show search criteria (origin, destination, dates, passenger count) - For each flight, display: - Flight number - Departure and arrival times - Available seats by cabin type (basic_economy, economy, business) - Prices by cabin type - Indicate if no flights found matching criteria

### Query Type 2: Check Reservation Details

**Step 1: User Identity Verification**

**Input Parameters**: {user_id}, {reservation_id}

**Process**: Verify user exists and that the reservation belongs to them.

**Output**: Validation result

**Policy Rules**: - User must exist in Passengers profile - {reservation_id} must be in user's {reservations} list - If validation fails, deny access to reservation details

**Step 2: Get Reservation Information**

**Input Parameters**: {reservation_id}

**Process**: Retrieve complete reservation details from Reservations profile including all fields.

**Output**: {reservation_info} with {user_id}, {origin}, {destination}, {flight_type}, {flights}, {passengers}, {payment_history}, {total_baggages}, {nonfree_baggages}, {created_at}

**Policy Rules**: Return all reservation data without modification

**Step 3: Get Flight Real-time Status**

**Input Parameters**: {flights} list from reservation

**Process**: For each flight in the reservation, query Flights profile to get current real-time status.

**Output**: {flight_statuses} list with {status}, {expected_departure_time}, {expected_arrival_time}, and status-specific fields

**Policy Rules**: Retrieve current status for each flight from Flights profile

**Step 4: Format and Return Details**

**Input Parameters**: {reservation_info}, {flight_statuses}

**Process**: Format complete reservation details into readable display.

**Output**: Formatted reservation details

**Policy Rules - Display Format**: - Show reservation ID and booking user - Show booking time and status - For each flight: flight number, route, date, times, current status - For each passenger: name, cabin, insurance status - Show baggage information (total bags, paid bags) - Show payment information (methods used, amounts)

### Query Type 3: Check Flight Status

**Step 1: Parameter Validation**

**Input Parameters**: {flight_number}, {date}

**Process**: Validate that flight number and date are in correct format.

**Output**: Validation result

**Policy Rules**: - {flight_number} must be valid format (3-letter airline code + 3-4 digit number) - {date} must be valid date format (YYYY-MM-DD)

**Step 2: Get Flight Information**

**Input Parameters**: {flight_number}, {date}

**Process**: Query Flights profile for the specified flight and date. Retrieve all available information.

**Output**: {flight_details} with {origin}, {destination}, {scheduled_departure_time}, {scheduled_arrival_time}, {status}, and status-specific fields

**Policy Rules**: - If flight_number not found: Return error "Flight number does not exist" - If date not found for flight: Return error "No flight information for this date" - Return all available information for the flight on that date

**Step 3: Format and Return Status**

**Input Parameters**: {flight_details}

**Process**: Format flight status information into readable display based on current status.

**Output**: Formatted flight status

**Policy Rules - Display Format**: - Show flight number, date, and route - Show scheduled departure and arrival times - Show current status (available/delayed/landed/cancelled) - If status is "available": Show expected times, available seats by cabin, prices by cabin - If status is "delayed": Show new expected times and delay reason - If status is "landed": Show actual departure and arrival times - If status is "cancelled": Show cancellation reason

### Query Type 4: Check User Information

**Step 1: User Identity Verification**

**Input Parameters**: {user_id}

**Process**: Verify user exists and retrieve complete profile from Passengers profile.

**Output**: {user_info} with {PaxID}, {name}, {email}, {dob}, {address}, {membership}, {payment_methods}, {saved_passengers}, {reservations}

**Policy Rules**: - User must exist in Passengers profile - If user not found: Return error "User ID invalid" - Return all user profile information

**Step 2: Get User Reservation History**

**Input Parameters**: {reservations} list from user profile

**Process**: For each reservation ID in user's reservations list, retrieve summary information from Reservations profile.

**Output**: {reservation_summary} list with {reservation_id}, {route}, {date}, {status}, {passenger_count} for each reservation

**Policy Rules**: Retrieve summary (not full details) for each reservation

**Step 3: Format and Return User Info**

**Input Parameters**: {user_info}, {reservation_summary}

**Process**: Format complete user information into readable display.

**Output**: Formatted user information

**Policy Rules - Display Format**: - Show passenger ID and name - Show email and date of birth - Show address (full address with city, state, zip, country) - Show membership level and benefits - Show payment methods (type, last 4 digits for cards, amount for certificates) - Show saved passengers list - Show reservation history with summary for each reservation

We evaluated our internalization approach on this single policy document with Qwen-3-8B, observing a performance gain with our CAP-CPT approach. More comprehensive evaluation on different policy documents are leaved for future work.

