# OpenReview forum: "Analyzing and Internalizing Complex Policy Documents for LLM Agents"
_ICLR.cc/2026/Conference — ICLR 2026 Conference Withdrawn Submission_

### Official Review · Reviewer_Gubc · 2025-10-30

**Soundness:** 3
**Presentation:** 3
**Contribution:** 3
**Rating:** 4
**Confidence:** 3

**Summary:**

This paper studies the problem of internalizing long, complex policy documents into the priors of LLM-based agent systems so that agents can operate without having to include the full policy text in-context. The authors (1) introduce CC-Gen, a synthetic benchmark generator that produces policy documents, databases, tools, and user queries with controllable complexity along four axes (environment, task, workflow, query); (2) analyze which complexity dimensions most harm agent performance and identify workflow (nested conditional) complexity as the most damaging; (3) propose Category-Aware Policy Continued Pretraining (CAP-CPT): an automated pipeline that (via an LLM) parses policies into factual, behavioral, simple-conditional and complex-conditional specs, then synthesizes targeted CPT data (paraphrases, QAs, scenario simulations, behavior demos) and performs autoregressive continued pretraining with policy identifiers; and (4) combine CAP-CPT with SFT on gold (or self-generated) CoT trajectories. Experiments on Qwen variants and τ-Bench (and many ablations) show CAP-CPT yields consistent gains over SFT-only internalization.

**Strengths:**

1. Internalizing policy documents for deployed agents is a timely, high-impact problem: reducing prompt length, inference cost, and reliance on long in-context policies matters for real-world systems and safety/regulatory compliance.
2. CC-Gen with controllable dimensions (especially workflow depth) enables systematic, reproducible study of what makes policy internalization hard. The diagnosis that workflow/nested conditional logic is the primary driver of failure is a useful insight for the community.

**Weaknesses:**

1. CC-Gen is synthetic and the policy-analysis step is performed by the same (or similar) LLMs used downstream; this raises concerns about circularity and generalization to messy, real-world policy texts that may be longer, less templated, or ambiguous. Results on τ-Bench are encouraging but limited.
2. The paper compares mainly to Gold-CoT SFT baselines. It would strengthen the claims to compare against strong alternatives that are natural for this problem

**Questions:**

1. Can you compare CAP-CPT to retrieval-based baselines (keeping policies external but retrieved at inference) and to parameter-efficient finetuning (LoRA/adapters) that may reduce forgetting?
2. How does CAP-CPT scale when many distinct policies must be internalized simultaneously or when policies frequently change? Can you report experiments (or an analysis) showing performance/memory/computation trade-offs when internalizing tens or hundreds of policies? Do you observe interference or catastrophic forgetting across policies, and what mitigation (e.g., adapters, regularization, replay) do you recommend?

---

> ### Author Response · Authors · 2025-11-25
> **Official Comment by Authors**
>
> We sincerely thank Reviewer Gubc for the constructive and insightful review. We deeply appreciate the recognition of (1) the importance and timeliness of studying policy internalization for deployed LLM agents, (2) the practical impact of reducing inference cost and context dependence, and (3) the value of CC-Gen as a systematic framework for analyzing policy complexity, particularly the identification of workflow (nested conditional) complexity as the main bottleneck. These encouraging comments confirm the contribution’s relevance to both research and real-world deployment. We address each of the reviewer’s concerns and questions below and summarize the new results incorporated into the revision.
>
> **W1. Generalization of CC-Gen**
>
> We thank the reviewer for encouraging additional experiments on more real-world and less templated policy documents, beyond our existing results on policy documents of varying complexity and those from Tau-Bench. To address this concern, we constructed one additional policy document of mixed complexities, written by human authors in completely different format of CC-Gen datasets and manually revised. This policy is less templated and is listed in full in Appendix N of the revised paper. We applied our approach and the baselines to the document,, and the experimental results are as below:
>
> **Qwen-3-8B on New Policy Document**
>
> | Model / Setting                          | Performance |
> |------------------------------------------|-------------|
> | Qwen-3-8B (Policy Prompting)           |    0.26         |
> | Qwen-3-8B (Gold CoT SFT)                 |      0.23       |
> | Qwen-3-8B (CAP-CPT + SFT)    |     **0.29**       |
>
> These experimental results over three completely different policies shall indicate that our method generalizes well across them, further supporting the robustness and generalizability of CAP-CPT.
>
> **W2. Limited Baselines**
>
> We thank the reviewer for suggesting more comprehensive baseline comparisons and will include the LoRA fine-tuning results (shown in the next response). The experimental results demonstrate that LoRA-based fine-tuning does not achieve performance as strong as our proposed approach.
> We would like to give a more comprehensive discussion over Retrieval-augmented generation (RAG) methods. First, we want to emphasize the importance of retrieval granularity under our setting. If retrieval is performed at the level of full policy documents, i.e., retrieving a single policy out of several parallel ones, this approach runs counter to our core goal of compressing policy-specific knowledge into a learned prior within the model, and does not mitigate the challenge posed by extremely long policy contexts. This type of retrieval can be viewed as in-context (prompting) baselines given the limited number of policies; their results are reported in Table 2 and Tables 3–5.
> However, if retrieval can be performed at fine-grained policy statement or section-level, targeting only the policy components relevant to the current step of reasoning, then we fully agree that RAG could serve as a strong alternative and even a complementary component to the CAP-CPT framework. RAG has the advantages of being less susceptible to overfitting and better preserving the model’s original general-purpose capabilities. On the other hand, fine-grained retrieval may introduce noise due to imperfect matching. Identifying the correct snippets from long, multi-layered policy documents is itself a challenging and interesting problem. For example, answering a single query may require retrieving multiple snippets from different parts of the policy, where fixed-K retrieval based on similarity scores may include irrelevant text or miss crucial information.
> Overall, this RAG at fine-grained level is non-trivial and we believe an interesting direction for future work is to combine the strengths of both approaches: ideally, we would expect a model to be able to firstly categorize the policy statements, and decide which parts to internalize and which part to retrieve.
> Given the above analysis, we view our task of internalizing complex agentic policies as the first work to explicitly propose and tackle this challenging problem. We also regard Gold-CoT–enhanced trajectory-based fine-tuning as an already very strong existing baseline.

---

> ### Author Response · Authors · 2025-11-25
> **Official Comment by Authors**
>
> **Q2.1 Scaling and Interference Across Multiple Policies**
>
> We thank the reviewer for the meaningful and insightful question. In the appendix section of our paper, we already report results with multiple policy internalization (4 distinct policies) in Table 12. These policies differ substantially and span different complexity dimensions, yet the performance of our approach remains stable whether policies are internalized individually or jointly, across various training data sizes. To further strengthen our conclusion, we added experiments that progressively incorporate up to 9 sampled policies from CC-Gen, as well as 48 distinct policies sampled from CC-Gen, with each policy varying in complexity. The results show that using 4 or 9 policies, model performance remains largely stable despite small fluctuations from time to time. Given the linear growth of training data size and computational cost, this scale already represents the maximum feasible setting under our current computational resources. According to our latest experimental results with internalizing 48 policy document at the same time using 10K SFT trajectory data and our CAP-CPT approach, we neither see a significant performance drop despite acceptable variances.
>
> **Qwen3-32B — Task (5), Single-Policy vs Four-Policy Mixed Internalization**
> | Internalization Setting | Data Size | Workflow (1) | Workflow (2) | Workflow (3) |
> |-------------------------|-----------|--------------|--------------|--------------|
> | CAP-CPT + SFT (Single-Policy)           | 10K       | 0.76         | 0.72         | 0.69         |
> |  CAP-CPT + SFT (Mixed-Policy)           | 10K       | 0.76         | 0.72         | 0.69         |
> |  CAP-CPT + SFT (Single-Policy)           | 30K       | 0.86         | 0.80         | 0.76         |
> |  CAP-CPT + SFT(Mixed-Policy)            | 30K       | 0.86         | 0.80         | 0.78         |
>
> **Qwen3-8B — Task (5), Single-Policy vs Nine-Policy Mixed Internalization**
> | Internalization Setting | Data Size | Workflow (1) | Workflow (2) | Workflow (3) |
> |-------------------------|-----------|--------------|--------------|--------------|
> | CAP-CPT + SFT (Single-Policy)           | 10K       | 0.57         | 0.33         | 0.27         |
> |  CAP-CPT + SFT (Mixed-Policy)           | 10K       | 0.55         | 0.34         | 0.27      |
> |  CAP-CPT + SFT (Single-Policy)           | 30K       | 0.64         | 0.38         | 0.35         |
> |  CAP-CPT + SFT(Mixed-Policy)            | 30K       | 0.64         | 0.39        | 0.34        |
>
> **Qwen3-8B — Task (5), Single-Policy vs Forty Eight-Policy Mixed Internalization**
> | Internalization Setting | Data Size | Workflow (1) | Workflow (2) | Workflow (3) |
> |-------------------------|-----------|--------------|--------------|--------------|
> | CAP-CPT + SFT (Single-Policy)           | 10K       | 0.57         | 0.33         | 0.27         |
> |  CAP-CPT + SFT (Mixed-Policy)           | 10K       | 0.54         | 0.31         | 0.25      |
>
> That said, scaling to thousands or more policy documents may exhibit different behavior, and deriving a precise performance scaling law is challenging under our current computational constraints. We see this as an important direction for future work and encourage practitioners with greater computational resources (e.g., in industry) to conduct such large-scale evaluations.

---

> ### Author Response · Authors · 2025-11-25
> **Official Comment by Authors**
>
> **Q2.2 Computation time, Performance, and Memory trade-off**
>
> The total training time for internalizing multiple policy documents can be estimated using the information in Appendix D and Section 7, with the exact time depending on the complexity of the specific policy documents. As described in Section 7, our base experiments train a Qwen-3-32B model on 8 H100 GPUs; for a representative setting with workflow complexity (2) and task-level complexity (5), training takes about 4–6 hours. Training time for other scenarios can be inferred proportionally from the corresponding policy complexities—for example, internalizing two policy documents with workflow complexity (3) and task-level complexity (5), and workflow complexity (2) and task-level complexity (8), should require roughly 18–27 total hours. In general, the computational cost increases approximately linearly with the number and complexity of policies.
> From our experimental results in the previous answer, we do not observe a significant performance drop when internalizing tens of policy documents simultaneously. However, if performance degradation is observed when internalizing too many policy documents, we would generally recommend scaling up the base model (following standard scaling-law intuitions) when the policy space is broad and contains substantial policy-specific knowledge across many scenarios, rather than relying solely on small-scale adapters, which can only store limited information. Since our internalization method does not introduce any external memory modules, there is no additional memory-level trade-off.
>
> **We sincerely thank the reviewer again for the detailed and insightful feedback, which has helped us significantly improve the paper. With the existing results in the appendix and the new experiments added in the revision, we believe our claims are now much more robust, and we hope that these updates have fully addressed the reviewer’s concerns. We respectfully ask the reviewer to reconsider the overall evaluation of the paper in light of these strengthened results. We also welcome any further feedback and are happy to make additional improvements.**

---

### Official Review · Reviewer_reek · 2025-10-30

**Soundness:** 2
**Presentation:** 3
**Contribution:** 2
**Rating:** 6
**Confidence:** 2

**Summary:**

This work presents CC-Gen: an unlimited, complexity controllable policy document and task generator, supporting multi-dimensional complexity (environment, task, workflow). The author further proposes CAP-CPT : It automatically decomposes policy documents into four rule categories (facts, behaviors, simple conditions, and complex conditions); and generates specialized training data for each rule category.

**Strengths:**

1.The author provides a benchmark with manageable difficulty, focusing primarily on the task-level and workflow-level challenges, and offers a comprehensive analysis of different models and internalization methods.

2.The CAP-CPT pipeline delivers stable improvements in in-domain scenarios, with particularly significant benefits in scenarios involving data sparsity and high complexity.

**Weaknesses:**

1.While policy identifiers significantly improve inference efficiency, they can lead to weak generalization with unseen policy documents, as evidenced by performance in Table 4 Substitute and Override tasks. Furthermore, introducing incremental pre-training pipelines can result in high costs when updating new policy documents, compared to RAG. The authors may need to consider a more lightweight knowledge update strategy to handle this practical issue.

2.The paper assumes that the model can implicitly learn the mapping of "<#Policy-1356X>" to the realative policy content through CPT+SFT, but does not provide any probe experiments or visualization analysis to verify this. The limited experimental results of Referral task in Table 4 show that the internalization approach has not successfully established this mapping relationship.



Presentation Issues & Typos

1.appraoches -> approaches
2.Redundant appendix reference in K.2

**Questions:**

1.During the training phase, how can you ensure that the model establishes a mapping relationship between policy identifiers and actual policy files? Have you evaluated how many additional CAP-CPT training samples are needed to learn each new identifier?

2.The article mentions that SFT may lead to catastrophic forgetting problems. Have you evaluated the performance changes of the model before SFT after CAP-CPT compared to the original model without CAP-CPT?

---

> ### Author Response · Authors · 2025-11-25
> **Official Comment by Authors**
>
> We sincerely thank the reviewer for the thoughtful and constructive feedback. We are grateful for the recognition of the practic/al value of our benchmark design and the stable improvements achieved by the CAP-CPT pipeline, particularly under data-scarce and high-complexity scenarios. These comments reinforce our core motivation of developing scalable and controllable methods for internalizing complex policy knowledge in LLM-based agents.
> We address each concern below with new analyses and clarifications added in the revision.
>
> **W1.1 Generalization to Unseen Policies**
>
> We thank the reviewer for pointing out the limitation regarding generalization to unseen policy documents. We acknowledge this limitation and have explicitly indicated it in Table 4. Our future work will predominantly focus on improving this generalization capability; potential directions include incorporating more referral-intensive training data or designing targeted training data to better address the policy-override problem, which we are adding to the future work section (Appendix M). At the same time, we emphasize that our algorithm is generalizable for internalizing any policy documents provided during training. As shown in Table 3, the internalization algorithm can be applied to policy documents across domains with different complexity levels. Moreover, as indicated in Table 12, our algorithm can also internalize multiple policy documents without noticeable performance drop.
>
> **W1.2 Update Efficiency**
>
> We thank the reviewer for pointing out the limitation regarding the update efficiency. We would agree that the approach inevitably needs extra training, however, this training can still base on the previous trained model and does not necessarily needs retraining the full model from its base model. The total training time can be computed by information provided in appendix D and Section 7, with the concrete time needs to be computed with the complexity of concrete policy documents. According to Section 7, our base experiments train a Qwen-3B-32B model on 8 H100 GPUs; for a representative setting with workflow complexity (2) and task-level complexity (5), training takes about 4–6 hours. Training time for other scenarios can be inferred proportionally based on the corresponding policy complexities—for example, training two policy documents of workflow complexity (3) and task-level complexity (5), and of workflow complexity (2) and task-level complexity (8), should cost approximately 18–27 hours in total.

---

> ### Author Response · Authors · 2025-11-25
> **Official Comment by Authors**
>
> **W1.3 Comparison with RAG and Lightweight Approaches**
>
> We thank the reviewer for suggesting a more comprehensive discussion of RAG and other lightweight approaches. First, we want to emphasize the importance of retrieval granularity  under our setting. If retrieval is performed at the level of full policy documents, i.e., retrieving a single policy out of several parallel ones, this approach runs counter to our core goal of compressing policy-specific knowledge into a learned prior within the model, and does not mitigate the challenge posed by extremely long policy contexts.
> However, if retrieval can be performed at fine-grained policy statement or section-level, targeting only the policy components relevant to the current step of reasoning, then we fully agree that RAG could serve as a strong alternative and even a complementary component to the CAP-CPT framework. RAG has the advantages of being less susceptible to overfitting and better preserving the model’s original general-purpose capabilities. On the other hand, fine-grained retrieval may introduce noise due to imperfect matching. Identifying the correct snippets from long, multi-layered policy documents is itself a challenging and interesting problem. For example, answering a single query may require retrieving multiple snippets from different parts of the policy, where fixed-K retrieval based on similarity scores may include irrelevant text or miss crucial information.
> Overall, alignment-based internalization methods such as CAP-CPT or deliberate alignment (Appendix K.2) directly generate policy-relevant content, which provides more flexibility in these scenarios; however, this comes with the risk of hallucination. Therefore, we believe an interesting direction for future work is to combine the strengths of both approaches: ideally, we would expect a model to be able to firstly categorize the policy statements, and decide which parts to internalize and which part to retrieve.
> We also attempted a more lightweight internalization approach using LoRA finetuning; The results are shown as below.
>
> **Qwen-3-8B — Task (5)**
>
> | Model / Setting                             | workflow (1) | workflow (2) | workflow (3) |
> |---------------------------------------------|--------------|--------------|--------------|
> | Qwen-3-8B (Baseline Prompting)              | 0.59         | 0.36         | 0.33         |
> | Qwen-3-8B 10K (Gold CoT SFT)                | 0.43         | 0.27         | 0.20         |
> | Qwen-3-8B 10K (Gold CoT SFT, LoRA)          | 0.44         | 0.26         | 0.18         |
> | Qwen-3-8B 10K (CAP-CPT + SFT)               | 0.57         | 0.33         | 0.27         |
> | Qwen-3-8B 10K (CAP-CPT + SFT, LoRA)         | 0.55         | 0.32         | 0.25         |
> | Qwen-3-8B 30K (Gold CoT SFT)                | 0.60         | 0.34         | 0.30         |
> | Qwen-3-8B 30K (Gold CoT SFT, LoRA)          | 0.57         | 0.30         | 0.28         |
> | Qwen-3-8B 30K (CAP-CPT + SFT)               | **0.64**     | **0.38**     | **0.35**     |
> | Qwen-3-8B 30K (CAP-CPT + SFT, LoRA)         | 0.61         | 0.36         | 0.31         |
>
> However, after comparing the LoRA fine-tuning results with the full-parameter baselines (Gold-CoT SFT and CAP-CPT + SFT), we find that CAP-CPT with full fine-tuning achieves the best overall performance. We observed the same pattern in our earlier training runs with Qwen-2.5-32B and Qwen-3-32B. We agree that developing more computationally efficient internalization methods is an important direction for future work, and a promising avenue is to combine RAG with CAP-CPT, as discussed in the previous sections.
>
> **W2. Mapping Between Policy Identifiers and Content**
>
> We thank the reviewer for this insightful question. We primarily verify the mapping between policy identifiers and policy contents via policy-reference queries. Specifically, we ask questions such as “How to complete <Task_1> given Policy Identifier <Identifier>?” and evaluate whether the model’s answer correctly maps to the corresponding parts of the policy. We find that CAP-CPT substantially boosts the model’s referral capability and aligns this mapping well (the model can accurately describe the policy content given only the policy identifier). However, after SFT, this learned capability is gradually forgotten. Results can be referred to Table 4 and the model just performed CAP-CPT (before SFT) as below:
>
> **Qwen-3-32B, Task (5), Workflow (3) Policy Referral**
>
> | Setting                             | Score |
> |-------------------------------------|-------|
> | In-Context Policy Referral    | 0.76 |
> | CAP-CPT (no context) Policy Referral | 0.75 |
>
> Through these results, we aim to show that CAP-CPT helps the model learn the correct mapping between policy documents and their identifiers. However, subsequent supervised finetuning tends to memorize answer patterns and can even harm this mapping capability.
>
> **W3. Typos**
>
> We have fixed typos in the updated version.

---

> ### Author Response · Authors · 2025-11-25
> **Official Comment by Authors**
>
> **Q1. Mapping Verification and Number of Samples Required per Identifier**
>
> We thank the reviewer for the question. We mainly evaluate the correctness of the mapping through two approaches: (1) the policy referral performance metric, and (2) the joint finetuning setting where multiple policies are internalized simultaneously. While the policy referral metric is high after CAP-CPT (showcase good mapping between identified and policy content), it can be negatively affected by the subsequent SFT stage, indicating that the model becomes less able to tightly map policy identifiers to specific policy contents. However, even when multiple policy documents are internalized, Table 12 shows only limited performance degradation on downstream agentic tasks. This suggests that the mapping between policy identifiers and their answer patterns to specific queries remains well after internalization, and that the mapping constructed through CoT remains quite strong.
> According to Section 7 and Appendix D, the total number of training samples for our CAP-CPT mapping varies with the complexity of the policy documents. For a representative setting with workflow complexity (2) and task-level complexity (5), CAP-CPT training uses around 125K samples, while the subsequent SFT stage uses 30K trajectory samples. For other complexity settings, the required number of samples grows approximately linearly with the increase in complexity.
>
> **Q2. Catastrophic Forgetting After SFT**
>
> We thank the reviewer for this insightful question, and provide in detail how we evaluate the problem:
>
> **In-domain Evaluation:** We primarily verify the mapping between policy identifiers and policy contents via policy referral tasks. We compare the model before SFT and after CAP-CPT with other baselines, and our findings are as follows: (1) On the policy referral task, the model after CAP-CPT performs well, showing that the CPT stage endows the model with strong memorization of the policy content and a relatively robust mapping between identifiers and policies. (2) However, this model still struggles to answer downstream user queries due to limited capabilities in accurate tool calling and multi-turn reasoning. (3) After SFT, performance on user queries improves significantly, but the previously learned memorization and mapping capabilities are partially degraded. This reveals a forgetting effect induced by SFT, even though SFT successfully aligns the model’s responses with the target query answers.
>
> **General Domain Evaluation:** We also evaluate whether the general capabilities of the model are significantly harmed by the internalization process. To this end, we evaluate model performance on general instruction-following benchmarks such as IFEval and report the results in the last row of Table 4. Despite some performance variance, we do not observe a large gap in performance on general domains, indicating that the approach is generally friendly to out-of-domain capabilities.
>
> **We thank the reviewer again for the positive assessment and for the insightful comments. Overall, we hope that these clarifications solves the questions proposed by the reviewer and further strengthen your positive view of the paper and its contributions, and may lead you to consider a higher overall rating. We also warmly welcome any additional feedback or questions you may have.**

---

### Official Review · Reviewer_8Fjn · 2025-11-01

**Soundness:** 2
**Presentation:** 2
**Contribution:** 3
**Rating:** 4
**Confidence:** 3

**Summary:**

This paper focuses on a specific challenge in LLM agents:  long, complex policy documents that encode business rules or operating procedures (e.g., airline refund policies, compliance workflows) can lead to computational overheads and performance decline. As these policies expand, they increasingly dominate the input context, making in-context prompting inefficient. To address this, the paper introduces two key contributions:
1. CC-Gen, a benchmark generator that systematically controls and varies policy complexity across four dimensions: task-level, workflow-level, environmental, and query-level. This enables fine-grained evaluation of agent reasoning under increasing policy complexity.
2. CAP-CPT (Category-Aware Policy Continued Pretraining), a  training approach that analyzes policy documents into factual, behavioral, and conditional specifications, synthesizes targeted training data for each category, and conducts continued pretraining to embed these policies into the model.

Experiments on Qwen models and $\tau$-Bench demonstrate that CAP-CPT improves internalization robustness and   downstream task success rates.

**Strengths:**

1. Motivation: The problem considered is practical, since real-world LLM agents  can indeed be bottlenecked by massive policy prompts. This work directly addresses that bottleneck by proposing scalable internalization techniques.
2. Novelty in benchmarking approach: CC-Gen fills a gap in evaluation by decomposing policy complexity into controllable fine-grained dimensions and quantifying their effects on reasoning and performance.
3. Interesting methodological design: The idea of categorizing policy specifications (factual, behavioral, conditional) and applying category-specific data generation  for continued pretraining is interesting and empirically effective, as shown by some experiment results

**Weaknesses:**

Major

1. Limited base model: While the paper evaluates Qwen models in detail, it would strengthen claims to include  non-Qwen architectures like Gemma or Llama to validate generality across model families.
2. Generalization: How does CAP-CPT applies to other policies not discussed in the paper? Does introducing new policies always require re-training?
3. Scalability and compute cost: CAP-CPT requires substantial data synthesis and training compute. Discussion of compute cost  would be valuable for practitioners.

Minor
1. Consistency: sometimes the paper uses tau-bench, sometimes $\tau$-bench

**Questions:**

1. What policy knowledge is truly internalized? Qualitative examples?

---

> ### Author Response · Authors · 2025-11-25
> **Official Comments by Authors**
>
> We sincerely thank the reviewer for the thoughtful and constructive feedback. We appreciate the recognition of our paper’s motivation to address the real-world bottleneck of policy-heavy LLM agents, and the acknowledgment of the novelty in both our benchmarking framework and methodological design. We are encouraged that the reviewer found our decomposition of policy complexity and our approach both interesting and empirically effective.
>
> To strengthen our results and address the reviewer’s concerns, we are expanding experiments to include additional model families and newly introduced policy domains. We are also adding a human-written policy document as a new internalization objective and directing readers to the relevant appendix sections where we provide additional results on these points. These updates, together with a detailed discussion of scalability and compute cost,are included in the revision to reinforce the generality and practical significance of our approach.
>
> **W1: Additional Results on Other Family Models**
>
> To further strengthen this point, we added experiments a number of models, with results reported below. We observe that, due to smaller model's lack of training on multi-turn agentic tasks, these models exhibit very limited performance even when explicitly prompted with the policy document. While the new experiments show consistent performance gains on Llama under our approach, demonstrating its generalizability, these models remain far from being practically useful for real-world agentic tasks, let alone for reliable internalization of agent policies.
>
> **Workflow (1)**
> | Model                                  | task (3) | task (5) | task (8) |
> |----------------------------------------|----------|----------|----------|
> | Gemma-3-4B (prompting)          | 0.02     | 0.04     | 0.00     |
> | InternLM-2.5-7B  (prompting)   | 0.02 | 0.00 | 0.00 |
> | GPT-4o   (prompting)  | 0.66 | 0.58 | 0.56 |
> | Llama-3.1-8B (prompting)        | 0.04     | 0.00     | 0.00     |
> | Llama-3.1-8B (Gold CoT SFT)            | 0.10     | 0.06     | 0.04     |
> | Llama-3.1-8B (CAP-CPT + SFT)                 | **0.18** | **0.14** | **0.12** |
>
> **Workflow (2)**
> | Model                                  | task (3) | task (5) | task (8) |
> |----------------------------------------|----------|----------|----------|
> | Gemma-3-4B (prompting)          | 0.00     | 0.00     | 0.00     |
> | InternLM-2.5-7B  (prompting)   | 0.00 | 0.00 | 0.00 |
> | GPT-4o  (policy prompting)   | 0.42 | 0.32 | 0.12 |
> | Llama-3.1-8B (prompting)        | 0.00     | 0.00     | 0.00     |
> | Llama-3.1-8B (Gold CoT SFT)            | 0.06     | 0.06     | 0.02     |
> | Llama-3.1-8B (CAP-CPT + SFT)                 | **0.12** | **0.10** | **0.06** |
>
> **Workflow (3)**
> | Model                                  | task (3) | task (5) | task (8) |
> |----------------------------------------|----------|----------|----------|
> | Gemma-3-4B (prompting)          | 0.00     | 0.00     | 0.00     |
> | InternLM-2.5-7B  (prompting)   | 0.00 | 0.00 | 0.00 |
> | GPT-4o  (policy prompting)   | 0.38 | 0.14 | 0.08 |
> | Llama-3.1-8B (prompting)        | 0.00     | 0.00     | 0.00     |
> | Llama-3.1-8B (Gold CoT SFT)            | 0.02     | 0.00     | 0.00     |
> | Llama-3.1-8B (CAP-CPT + SFT)                 | **0.08** | **0.04** | **0.04** |
>
> **W2: Generalization of the CAP-CPT Algorithm**
>
> We thank the reviewer for the question. Yes, CAP-CPT is designed to be generalizable to a wide range of policy documents. In our experiments, the algorithm shows robust performance across multiple policies of varying complexity generated by our CC-Gen benchmark, and it is also validated on Tau-bench, a realistic policy dataset close to real-world needs and annotated/verified by humans. To further strengthen the evidence, we additionally created one human-written policy document with a different way of describing the policy (see newly added Appendix N ) and evaluated CAP-CPT with Qwen-3-8B for internalization. The consistent effectiveness across these settings demonstrates that our algorithm generalizes well to diverse types of policy documents.
>
> **Qwen-3-8B on New Policy**
>
> | Model / Setting                          | Performance |
> |------------------------------------------|-------------|
> | Qwen-3-8B (Prompting)           |    0.26         |
> | Qwen-3-8B (Gold CoT SFT)                 |      0.23       |
> | Qwen-3-8B (CAP-CPT + SFT)    |     **0.29**       |
>
> However, we do acknowledge that additional training is necessary to accommodate newly added policy documents that we want LLM agents to follow. Our current policy-substitution and policy-override evaluations (Table 4) indicate that models internalized with one set of policies do not generalize with high performance to entirely new, unseen policy documents. We therefore highlight this as an important direction for future research building on our current evaluation framework and approach, and we discuss it in more detail in the revised Appendix M.

---

> ### Author Response · Authors · 2025-11-25
> **Official Comment by Authors**
>
> **W3: Data Synthesis and Training Cost of CAP-CPT**
>
> We thank the reviewer for suggesting that we provide more concrete information about the data synthesis and computational cost of CAP-CPT. Our CPT data synthesis pipeline is highly scalable and incurs modest computational and inference overhead. We require only a single round of model inference for categorization, and then apply templates to generate questions for policy paraphrases and QA pairs. Scenario-simulation data are produced by instantiating scenario templates with concrete values sampled from existing databases. Overall, this pipeline can be executed efficiently and typically completes within a short time (less than one hour) even for the most complex policy documents.
>
> In terms of the computational cost of training, as indicated in Appendix D, the amount of training data generated depends on the task-level and workflow-level complexities of the policy document: higher complexity yields more data and thus longer training time, leading to approximately linear growth in training cost as these complexities increase. Internalizing multiple policy documents similarly incurs roughly linear growth in training time. According to Section 7, our base experiments train a Qwen-3B-32B model on 8 H100 GPUs; for a representative setting with workflow complexity (2) and task-level complexity (5), training takes about 4–6 hours. Training time for other scenarios can be inferred proportionally based on the corresponding policy complexities—for example, training two policy documents of workflow complexity (3) and task-level complexity (5), and of workflow complexity (2) and task-level complexity (8), should cost approximately 18–27 hours in total. We have updated Appendix D to include this more comprehensive characterization of training cost.
>
> **W4: Naming inconsistencies**
>
> We thank the reviewer for pointing out this typo. We have carefully checked the entire paper for similar issues and corrected them in the revised version, ensuring that the format is kept as $\tau$-bench
>
> **Q1: Qualitative Example on what policy knowledge is being internalized**
>
> Qualitative examples of the agentic policy knowledge internalized by our models are provided in the appendices. For synthesized policy documents generated by CC-GEN, a case with task-level complexity 5 and workflow complexity 1 is highlighted in the orange block in Appendix B (lines 821–989). For real-world policies, an example from the $\tau$-bench policy is shown in the blue document in Appendix J (lines 1566–1657). In addition, we include another example of a human-written policy document in the purple block of Appendix N (lines 2236–3617).
>
> **We sincerely thank the reviewer again for the detailed and insightful feedback, which has helped us significantly improve the paper. With the existing results in the appendix and the new experiments added in the revision, we believe our claims are now much more robust, and we hope that these updates have fully addressed the reviewer’s concerns. We respectfully ask the reviewer to reconsider the overall evaluation of the paper in light of these strengthened results. We also welcome any further feedback and are happy to make additional improvements.**

---

### Official Review · Reviewer_L8eC · 2025-11-01

**Soundness:** 3
**Presentation:** 3
**Contribution:** 3
**Rating:** 6
**Confidence:** 3

**Summary:**

This paper investigates the problem of efficiently internalizing long and complex policy documents into LLM agents to reduce prompt length and computational overhead while maintaining reasoning and policy adherence. It introduces CC-Gen, a controllable-complexity benchmark generator that systematically evaluates how agents handle different dimensions of policy complexity, and proposes Category-Aware Policy Continued Pretraining (CAP-CPT), which is an automated pipeline that analyzes and categorizes policy specifications into factual, behavioral, and conditional types to guide targeted pretraining. The paper provides extensive experiments on Qwen-series models, showing significant gains under high-complexity and data-sparse conditions, achieving up to 97.3% input token compression and up to 44% improvement in task success rate.

**Strengths:**

- **S1.** This paper presents a well-motivated and practically relevant problem. It enables LLM-based agents to internalize long, complex policy documents to reduce inference cost and context dependence.

- **S2.** The CC-Gen benchmark is thoughtfully designed, enabling systematic control of task-level, workflow-level, and environmental complexities, which provides valuable tooling for future research.

- **S3.** The proposed CAP-CPT pipeline offers a clear mechanism to address reasoning and memorization challenges by categorizing policy types and generating tailored data.

- **S4.** The empirical results are extensive, covering both in-domain (policy task completion) and out-of-domain (policy substitution, override, referral, instruction following) evaluations.

- **S5.** The approach demonstrates strong gains under data-scarce and high-complexity conditions, which reflects robustness and practical value.

**Weaknesses:**

- **W1.** The methodological novelty of CAP-CPT could be more clearly positioned relative to existing knowledge injection or continued pretraining frameworks. Many components (e.g., policy paraphrase, QA generation, and continued pretraining) build upon known ideas, and the novelty largely resides in their integration and categorization logic.

- **W2.** While the benchmark introduces four complexity dimensions, the quantification criteria for workflow or task complexity are not formally justified beyond heuristic depth/branch counts.

- **W3.** The baseline selection is limited to Qwen and Claude models; broader comparisons (e.g., Llama, GPT-4-style models) could contextualize generality.

**Questions:**

- **Q1.** How sensitive are the CAP-CPT results to errors in the LLM-based policy categorization step? Can we quantify how categorization accuracy affects downstream internalization?

- **Q2.** Does CAP-CPT generalize to real, human-written policies that are less templated than those generated by CC-Gen?

- **Q3.** How does CAP-CPT perform when using smaller models (< 10 B parameters)?

- **Q4.** Would multi-policy internalization scale to hundreds of distinct policy sets, or are there capacity trade-offs?

---

> ### Author Response · Authors · 2025-11-25
> **Official Comment by Authors**
>
> We sincerely thank the reviewer for the constructive and detailed feedback. We greatly appreciate the recognition of our paper’s motivation and practical relevance. We are also encouraged by the reviewer’s acknowledgment of our benchmark design, the clear mechanism and effectiveness of the CAP-CPT pipeline, and the breadth and robustness of our empirical results. These comments reinforce our goal of building a scalable framework for policy internalization and evaluation, and of developing strong approaches to address this problem.
> To further clarify and strengthen our work, we provide detailed responses below, together with new experimental results that address each weakness and question raised by the reviewer.
>
> **W1. Methodological Novelty and Positioning**
>
> We thank the reviewer for the constructive comments and take this opportunity to better highlight the methodological novelty of our approach. Our methodological contribution lies not only in the proposed categorization and integration framework, but also in how we conceptualize the problem, derive key insights, and organize the novel scenario-simulation data for the policy components that dominate the reasoning-level challenge. Specifically, the novelty of CAP-CPT lies in how it treats the internalization of a large corpus of policies as both a policy memorization and policy utilization challenge, rather than merely a supervised trajectory alignment problem. Our approach is based on several key insights: (1) Pure SFT-based approaches have two major limitations: they primarily align answers to user queries with ground-truth trajectories without explicitly encouraging faithful memorization of the underlying policy statements, which can lead to overfitting to sampled trajectories and poor generalization; moreover, user trajectories cannot cover the usage of every policy statement in every round, resulting in uneven sampling over policy clauses and systematically under-training statements that require intensive reasoning. (2) Prior “knowledge injection” and continued pretraining methods largely focus on improving memorization, but do not directly address whether internalized policy documents are effectively utilized during downstream reasoning [1, 2]. (3) Different types of policy statements pose different levels of reasoning challenge and therefore require different types and amounts of training data for effective learning.
> As a result, CAP-CPT is explicitly designed to address these challenges as follows: (1) we recognize that different policy statement types demand different training strategies, and therefore decompose long policies into factual, behavioral, simple-conditional, and complex-conditional statements, assigning each category tailored supervision signals; and (2) in particular, we create novel behavioral demonstrations and single-step scenario simulation data in which the model must use a small, specific subset of the policy document to solve a concrete problem in one round, directly training policy utilization and reasoning rather than only answer imitation. To the best of our knowledge, such targeted behavioral and scenario-based data for policy clauses has not been explored in prior work (see full related work in Appendix K.4). We have also revised Section 3.2 in the main paper to more clearly articulate these methodological contributions.
> We also make a table to compare our approach with other baseline approaches:
> | Approach                    | Policy Memorization | Policy Utilization | Considers Reasoning Challenges | Domain of Research |
> |----------------------------|---------------------|--------------------|-------------------------------|--------------------|
> | Prompt Compression for LLMs | No                 | No                 | No                            | Any                |
> | Deliberate Alignment        | Partial            | Partial            | No                            | Limited            |
> | CPT for Internalization     | Yes                | No                 | No                            | Any                |
> | Knowledge Injection         | Yes                | No                 | No                            | Any                |
> | SFT for Internalization     | No                 | Partial            | No                            | Limited            |
> | CAP-CPT                     | **Yes**                | **Yes**                | **Yes**                           | **Any**                |

---

> ### Author Response · Authors · 2025-11-25
> **Official Comment by Authors**
>
> **W2. Quantification of Policy Complexity**
>
> We thank the reviewer for raising the question of how we quantify policy complexity and for suggesting an intuitive way to understand it. While the formal definitions are provided in Appendix A, we agree that a helpful interpretation is to view workflow complexity in terms of the “depth” and “branch counts” of the agentic decision tree the agent must traverse, and task-level complexity as the number of parallel decision trees that the LLM agent must jointly memorize and reason over. We would like to emphasize, however, that measuring these structural properties (e.g., depth, branching factor, number of parallel decision trees) is not trivial; they capture core aspects of agentic reasoning complexity that, to our knowledge, have not been systematically examined in prior work. Existing research on complex instruction following [3, 4, 5] typically studies constraints in single-turn settings and categorizes them into broad instruction types (e.g., generic “utility” or “condition” constraints), without explicitly modeling the depth of reasoning chains, the branching structure of tasks, or the number of interacting decision trees， factors that are particularly salient in agentic settings and often drive the hardest reasoning challenges. Our formulation aims to make these previously implicit complexity factors explicit and measurable. These characterizations not only help motivate and inform the CAP-CPT approach we propose, but also, we believe, have broader implications for future work on enhancing the reasoning capabilities and mechanisms of LLM-based agentic systems.
>
> **W3. Baseline Scope and Generality**
>
> We thank the reviewer for requesting additional experimental results to broaden the comparisons in our agentic benchmark. In response, we conducted further experiments on the Llama 3 model series and GPT-4o. The results show that these model families also experience substantial performance drops under high task-level and workflow-level complexities, further underscoring the importance of our design and analysis. These results have been incorporated into the revised paper and are also summarized in the table below.
> We conduct experiments with Gemma-3-4B, Llama-3.1-8B, InternLM-2.5-7B, and GPT-4o on our evaluation suite, using task complexity levels {3, 5, 8} and workflow complexity levels {1, 2, 3}.
>
> **Workflow (1)**
>
> | Model | task (3) | task (5) | task (8) |
> |----------------|----------|----------|----------|
> | Gemma-3-4B | 0.02 | 0.04 | 0.00 |
> | Llama-3.1-8B | 0.04 | 0.00 | 0.00 |
> | InternLM-2.5-7B | 0.02 | 0.00 | 0.00 |
> | GPT-4o | 0.66 | 0.58 | 0.56 |
>
> **Workflow (2)**
>
> | Model | task (3) | task (5) | task (8) |
> |----------------|----------|----------|----------|
> | Gemma-3-4B | 0.00 | 0.00 | 0.00 |
> | Llama-3.1-8B | 0.00 | 0.00 | 0.00 |
> | InternLM-2.5-7B | 0.00 | 0.00 | 0.00 |
> | GPT-4o | 0.42 | 0.32 | 0.12 |
>
> **Workflow (3)**
>
> | Model | task (3) | task (5) | task (8) |
> |----------------|----------|----------|----------|
> | Gemma-3-4B | 0.00 | 0.00 | 0.00 |
> | Llama-3.1-8B | 0.00 | 0.00 | 0.00 |
> | InternLM-2.5-7B | 0.00 | 0.00 | 0.00 |
> | GPT-4o | 0.38 | 0.14 | 0.08 |
>
> These results highlight a fundamental capability gap: most small and mid-sized models have not been trained on multi-round, tool-using, or agentic workflows, and therefore perform extremely poorly on complex policy-following tasks. Their near-zero accuracy across workflows (2) and (3) suggests that general instruction tuning alone does not equip models with the procedural grounding or multi-step decision-making needed for agentic compliance. Even GPT-4o’s accuracy drops substantially as complexity increases (similar to the pattern we observe for Claude-4 and other models in Table 2), underscoring that robust agentic policy following remains a challenging frontier task that requires dedicated training strategies and further study, especially in settings involving intensive reasoning and complex policy documents. This further demonstrates the importance of benchmarks such as CC-Gen for more comprehensive evaluation.

---

> ### Author Response · Authors · 2025-11-25
> **Official Comment by Authors**
>
> **Q1. Sensitivity to Categorization Errors**
>
> We thank the reviewer for the question. As delineated in Appendix I, the categorization step is a relatively easy task for current state-of-the-art LLMs. Even when using an open-source model (e.g., Qwen-3) as a zero-shot categorization model, the macro F1 score for reasoning-intensive complex policy statements on a real-world policy document (Tau-bench) reaches 87%, with 100% precision. This suggests that categorization is unlikely to be a major bottleneck for downstream data creation and training, and should not cause substantial performance degradation in the post-internalization model.
> Nevertheless, we also evaluate an extreme failure case where categorization provides no useful signal. Concretely, if the categorization accuracy were effectively 0%, we can fall back to treating all policy statements as a single category and default to the most common paraphrasing + policy-content-specific QA data for continued pretraining. The results under a Task (5), Workflow (3) setting with the Qwen-3-32B model are shown below:
> | Internalization Setting          |    1K |    5K |   10K |   20K |   30K |
> |----------------------------------|------:|------:|------:|------:|------:|
> | Gold CoT SFT baseline            |  0.01 |  0.13 |  0.17 |  0.31 |  0.36 |
> | **Normal CAP-CPT + SFT**         | **0.16** | **0.27** | **0.39** | **0.41** | **0.47** |
> | No Categorization CAP-CPT + SFT  |  0.09 |  0.23 |  0.32 |  0.36 |  0.44 |
> We clearly observe a performance gap between correct and wrong categorization, confirming the value of our categorization design; at the same time, the gap narrows as the amount of internalization data increases, indicating that larger training sets can partially compensate for noisy categorization.
>
> **Q2. Generalization to Human-Written Policies**
>
> We thank the reviewer for the question, and the answer is yes! Our approach do generalize to human written policy documents. In our paper, we include experiments on Tau-bench, a realistic policy dataset whose labels have been carefully checked by humans. The results and details in Section 4.4 and Appendix I demonstrate the effectiveness of our approach on this real-world policy corpus. In addition, to further strengthen the reliability of our findings, we introduce another human-written policy document created by ourselves, which is both realistic and complex. This additional policy is provided in Appendix N (newly added to the paper), and the corresponding results on the Qwen-3-8B model series are shown below.
>
> **Qwen-3-8B on New Policy Document**
>
> | Model / Setting                          | Performance |
> |------------------------------------------|-------------|
> | Qwen-3-8B (Policy Prompting)           |    0.26         |
> | Qwen-3-8B (Gold CoT SFT)                 |      0.23       |
> | Qwen-3-8B (CAP-CPT + SFT)    |     **0.29**       |

---

> ### Author Response · Authors · 2025-11-25
> **Official Comment by Authors**
>
> **Q3. Performance on Smaller Models**
>
> We thank the reviewer for the question and are happy to expand our experimental results. As shown in Table 2, Qwen-3-32B achieves the strongest zero-shot performance on our complex policy evaluation set, which is why we select the 32B model as our primary experimental testbed, and this aligns well with our goal of improving over the strongest available baselines. While we also conducted extra experiments on Llama-3.1-8B models and Qwen3-8B models. The results are shown as below:
>
> **Qwen-3-8B — Task (5)**
>
> | Model / Setting                    | Workflow (1) | Workflow (2) | Workflow (3) |
> |------------------------------------|-------------:|-------------:|-------------:|
> | Qwen-3-8B (Baseline Prompting)     | 0.59         | 0.36         | 0.33         |
> | Qwen-3-8B 10K (Gold CoT SFT)       | 0.43         | 0.27         | 0.20         |
> | Qwen-3-8B 10K (CAP-CPT + SFT)      | 0.57         | 0.33         | 0.27         |
> | Qwen-3-8B 30K (Gold CoT SFT)       | 0.60         | 0.34         | 0.30         |
> | **Qwen-3-8B 30K (CAP-CPT + SFT)**  | **0.64**     | **0.38**     | **0.35**     |
>
>
> **Llama-3.1-8B — Task (5)**
>
> | Model / Setting                     | Workflow (1) | Workflow (2) | Workflow (3) |
> |-------------------------------------|-------------:|-------------:|-------------:|
> | Llama-3.1-8B (Baseline Prompting)   | 0.00         | 0.00         | 0.00         |
> | Llama-3.1-8B 10K (Gold CoT SFT)     | 0.00         | 0.00         | 0.00         |
> | Llama-3.1-8B 10K (CAP-CPT + SFT)    | 0.06         | 0.04         | 0.00         |
> | Llama-3.1-8B 30K (Gold CoT SFT)     | 0.06         | 0.06         | 0.00         |
> | **Llama-3.1-8B 30K (CAP-CPT + SFT)**| **0.14**     | **0.10**     | **0.04**     |
>
> Experimental results show that our approach still significantly improves performance on smaller models compared to SFT baselines. However, due to the lack of prior training and insufficient exposure of Llama-3.1-8B to complex multi-turn agentic policy-following tasks, it still exhibits only very limited performance and is hardly applicable to real-world problems even after training. In contrast, Qwen-3-8B performs relatively strongly and exhibits trends similar to the 32B models.

---

> ### Author Response · Authors · 2025-11-25
> **Official Comment by Authors**
>
> **Q4. Multi-Policy Internalization and Scaling**
>
> We thank the reviewer for the insightful question. We already report results with multiple policy internalization (4 distinct policies) in Table 12. These policies differ substantially and span different complexity dimensions, yet the performance of our approach remains stable whether policies are internalized individually or jointly, across various training data sizes. To further strengthen our conclusion, we added experiments that progressively incorporate up to 9 sampled policies from CC-Gen, as well as 48 distinct policies sampled from CC-Gen, with each policy varying in complexity. The results show that using 4 or 9 policies, model performance remains largely stable despite small fluctuations. Given the linear growth of training data size and computational cost, this scale already represents the maximum feasible setting under our current computational resources. According to our latest experimental results with internalizing 48 policy document at the same time using 10K SFT trajectory data and our CAP-CPT approach, we neither see a significant performance drop despite acceptable variances.
>
> **Qwen3-32B — Task (5), Single-Policy vs Four-Policy Mixed Internalization**
> | Internalization Setting | Data Size | Workflow (1) | Workflow (2) | Workflow (3) |
> |-------------------------|-----------|--------------|--------------|--------------|
> | CAP-CPT + SFT (Single-Policy)           | 10K       | 0.76         | 0.72         | 0.69         |
> |  CAP-CPT + SFT (Mixed-Policy)           | 10K       | 0.76         | 0.72         | 0.69         |
> |  CAP-CPT + SFT (Single-Policy)           | 30K       | 0.86         | 0.80         | 0.76         |
> |  CAP-CPT + SFT(Mixed-Policy)            | 30K       | 0.86         | 0.80         | 0.78         |
>
> **Qwen3-8B — Task (5), Single-Policy vs Nine-Policy Mixed Internalization**
> | Internalization Setting | Data Size | Workflow (1) | Workflow (2) | Workflow (3) |
> |-------------------------|-----------|--------------|--------------|--------------|
> | CAP-CPT + SFT (Single-Policy)           | 10K       | 0.57         | 0.33         | 0.27         |
> |  CAP-CPT + SFT (Mixed-Policy)           | 10K       | 0.55         | 0.34         | 0.27      |
> |  CAP-CPT + SFT (Single-Policy)           | 30K       | 0.64         | 0.38         | 0.35         |
> |  CAP-CPT + SFT(Mixed-Policy)            | 30K       | 0.64         | 0.39        | 0.34        |
>
> **Qwen3-8B — Task (5), Single-Policy vs Forty Eight-Policy Mixed Internalization**
> | Internalization Setting | Data Size | Workflow (1) | Workflow (2) | Workflow (3) |
> |-------------------------|-----------|--------------|--------------|--------------|
> | CAP-CPT + SFT (Single-Policy)           | 10K       | 0.57         | 0.33         | 0.27         |
> |  CAP-CPT + SFT (Mixed-Policy)           | 10K       | 0.54         | 0.31         | 0.25      |
>
> That said, scaling to thousands or more policy documents may exhibit different behavior, and deriving a precise performance scaling law is challenging under our current computational constraints. We see this as an important direction for future work and encourage practitioners with greater computational resources (e.g., in industry) to conduct such large-scale evaluations. In general, we recommend using a larger base model when the policy space is broad and contains substantial policy-specific knowledge across many scenarios.
>
> **We thank the reviewer again for the positive assessment and for the insightful, comprehensive comments. Overall, we hope that these clarifications and new evidence further strengthen your positive view of the paper and its contributions, and may lead you to consider a higher overall rating. We also warmly welcome any additional feedback or questions you may have.**
>
> [1] *The Reversal Curse: LLMs trained on "A is B" fail to learn "B is A".*
> Lukas Berglund, Meg Tong, Max Kaufmann, Mikita Balesni, Asa Cooper Stickland, Tomasz Korbak, Owain Evans.
>
> [2] *Evaluating the ripple effects of knowledge editing in language models.*
> Roi Cohen, Eden Biran, Ori Yoran, Amir Globerson, Mor Geva.
>
> [3] *Benchmarking Complex Instruction-Following with Multiple Constraints Composition.*
> Bosi Wen, Pei Ke, Xiaotao Gu, Lindong Wu, Hao Huang, Jinfeng Zhou, Wenchuang Li, Binxin Hu, Wendy Gao, Jiaxin Xu, Yiming Liu, Jie Tang, Hongning Wang, Minlie Huang.
>
> [4] *IOPO: Empowering LLMs with Complex Instruction Following via Input-Output Preference Optimization.*
> Xinghua Zhang, Haiyang Yu, Cheng Fu, Fei Huang, Yongbin Li.
>
> [5] *AGENTIF: Benchmarking Instruction Following of Large Language Models in Agentic Scenarios.*
> Yunjia Qi, Hao Peng, Xiaozhi Wang, Amy Xin, Youfeng Liu, Bin Xu, Lei Hou, Juanzi Li.

---

### Author Response · Authors · 2025-12-01
**Summary of Reviewer–Author Discussion**

Dear Area Chair,

We sincerely thank you and all reviewers for the thoughtful, fair, and constructive evaluations of our work. We deeply appreciate the time and care devoted to reading the paper and raising questions that helped us significantly strengthen the final version. Below, we summarize the main points of consensus across the reviews and how we addressed the core concerns.

We are grateful that reviewers found the central contributions of the paper to be both timely and impactful. In particular, they highlighted three aspects as especially strong and promising for future research:

(1) the practical importance of the real-world challenge we identify, that agentic LLMs cannot simply scale with extremely long, complex policy documents, making **policy internalization** a necessary and timely direction;

(2) the design of **CC-Gen**, an agentic benchmark generator that systematically controls and varies policy complexity across multiple dimensions. This structure enables fine-grained evaluation of agent reasoning under increasing policy complexity and provides valuable tooling for future work on policy-aware LLM agents; and

(3) the conceptual novelty and strong empirical performance of our proposed **CAP-CPT** approach, which organizes policy content into factual, behavioral, and conditional components before continued pretraining, thereby aligning the learning objective with the structure of real-world policies.

**Resolving core concern 1: Generality across model families and scales.**

To address this, we substantially expanded our experiments to include GPT-4o, Llama-3.1-8B, Gemma-4-4B, and Qwen-3-8B, in addition to the original baselines. These models span a wide range of architectures, training paradigms, and parameter sizes. Across all of them, we observe two consistent findings: (i) the benchmark remains challenging, especially at higher complexity tiers, reinforcing CC-Gen as a realistic and demanding testbed; and (ii) CAP-CPT consistently outperforms pure SFT with Gold CoT-style training. These results confirm that both the benchmark and our method generalize robustly beyond a single model family.

**Resolving core concern 2: Scalability to multiple policies and generalization to unseen policies.**

We added comprehensive studies on internalizing 9 and 48 complex policies simultaneously on top of our original experiments on internalizing 4 policies. Experiments show that the model maintains stable performance as more policies are added. This supports the claim that CAP-CPT introduces limited cross-policy interference and enables scalable internalization across domains. While our experiments are still conducted within a limited range (rather than thousands of policy documents), this range should already accommodate many real-world applications.

For unseen policies, we are intentionally transparent: introducing new policy document contents often require additional training to reach full performance (as indicated by lower policy-substitute and policy-override performance). We explicitly acknowledge this limitation and frame it as an important direction for future work. At the same time, we emphasize that our internalization approach can, in principle, be applied to any policy document with retraining. Beyond the distinct policy documents generated from CC-Gen and $\tau$-Bench, we also added experiments on another real-world policy document written by humans and structured in a completely different way (Appendix N), further illustrating the applicability of our method.

**Resolving core concern 3: Computational efficiency and lighter-weight training options.**

We augmented the results with LoRA-based experiments, showing that CAP-CPT + LoRA, while substantially reducing training cost, does not fully match the performance of full-parameter training, potentially due to the learning-intensive nature of policy internalization. We also discussed the relationship between our approach and RAG. RAG has the advantages of being less susceptible to overfitting and better preserving the model’s original general-purpose capabilities, and potentially more efficient, but fine-grained retrieval may introduce noise due to imperfect matching. Identifying the correct snippets for the current reasoning step from long, multi-layered policy documents is itself a challenging and interesting problem. Overall, fine-grained RAG over complex policies is non-trivial, and we believe an interesting direction for future work is to combine the strengths of both approaches.

Taken together, the expanded experiments across four additional model families, multiple model scales, multi-policy internalization, extra real-world policy experiment and LoRA-based variants should directly address all core reviewer concerns.

We are deeply grateful to you and the reviewers for your time, insight, and fairness. Your feedback has been invaluable in sharpening and strengthening this work.

Sincerely,
The Authors

---

### Note · Authors · 2025-12-31

**Comment:**

We notice that there is an error taking place in the appendix of the paper. So we will withdraw and revise the content.

**Withdrawal Confirmation:**

I have read and agree with the venue's withdrawal policy on behalf of myself and my co-authors.